# ADAN: ADAPTIVE NESTEROV MOMENTUM ALGORITHM FOR FASTER OPTIMIZING DEEP MODELS

## ABSTRACT

Adaptive gradient algorithms combine the moving average idea with heavy ball acceleration to estimate accurate first- and second-order moments of the gradient for accelerating convergence. However, Nesterov acceleration which converges faster than heavy ball acceleration in theory and also in many empirical cases, is much less investigated under the adaptive gradient setting. In this work, we propose the ADAptive Nesterov momentum algorithm, Adan for short, to speed up the training of deep neural networks effectively. Adan first reformulates the vanilla Nesterov acceleration to develop a new Nesterov momentum estimation (NME) method, which avoids the extra computation and memory overhead of computing gradient at the extrapolation point. Then Adan adopts NME to estimate the first- and second-order moments of the gradient in adaptive gradient algorithms for convergence acceleration. Besides, we prove that Adan finds an $\epsilon$-approximate first-order stationary point within $\mathcal{O}\left(\epsilon^{-3.5}\right)$ stochastic gradient complexity on the non-convex stochastic problems (*e.g.* deep learning problems), matching the best-known lower bound. Extensive experimental results show that Adan surpasses the corresponding SoTA optimizers on vision, language, and RL tasks and sets new SoTAs for many popular networks and frameworks, *e.g.* ResNet, ConvNext, ViT, Swin, MAE, LSTM, Transformer-XL, and BERT. More surprisingly, Adan can use half of the training cost (epochs) of SoTA optimizers to achieve higher or comparable performance on ViT, ResNet, MAE, *etc*, and also shows great tolerance to a large range of minibatch size, *e.g.* from 1k to 32k. We hope Adan can contribute to developing deep learning by reducing training costs and relieving the engineering burden of trying different optimizers on various architectures.

## 1 INTRODUCTION

Deep neural networks (DNNs) have made remarkable success in many fields, *e.g.* computer vision (Szegedy et al., 2015; He et al., 2016) and natural language processing (Sainath et al., 2013; Abdel-Hamid et al., 2014). A noticeable part of such success is contributed by the stochastic gradient-based optimizers, which find satisfactory solutions with high efficiency. Among current deep optimizers, SGD (Robbins & Monro, 1951) is the earliest and also the most representative stochastic optimizer, with dominant popularity for its simplicity and effectiveness. It adopts a single common learning rate for all gradient coordinates but often suffers unsatisfactory convergence speed on sparse data or ill-conditioned problems. In recent years, adaptive gradient algorithms, *e.g.* Adam (Kingma & Ba, 2014) and AdamW (Loshchilov & Hutter, 2018), have been proposed, which adjust the learning rate for each gradient coordinate according to the current geometry curvature of the loss objective. These adaptive algorithms, *e.g.* Adam, often offer a faster convergence speed than SGD in practice.

However, none of the above optimizers can always stay undefeated among all its competitors across different network architectures and application settings. For instance, for vanilla ResNet (He et al., 2016), SGD often achieves better generalization performance than adaptive gradient algorithms such as Adam, whereas on vision transformers (ViTs) (Touvron et al., 2021), SGD often fails, and AdamW is the dominant optimizer with higher and more stable performance. Moreover, these commonly used optimizers usually fail for large-batch training, which is a default setting of the prevalent distributed training. Although there is some performance degradation, we still tend to choose the large-batch setting for large-scale deep learning training tasks due to the unaffordable training time. For example, training the ViT-B with the batch size of 512 usually takes several days, but when the batch size comes to 32K, we may finish the training within three hours (Liu et al., 2022a). Although some

Table 1: Comparison of different adaptive gradient algorithms on nonconvex stochastic problems. "Separated Reg." refers to whether the $\ell_2$ regularizer (weight decay) can be separated from the loss objective like AdamW. "Complexity" denotes stochastic gradient complexity to find an $\epsilon$-approximate first-order stationary point. Adam-type methods (Guo et al., 2021) includes Adam, and AdaGrad (Duchi et al., 2011), *etc.* AdamW has no available convergence result. For SAM (Foret et al., 2020), A-NIGT (Cutkosky & Mehta, 2020) and Adam$^+$ (Liu et al., 2020), we compare their adaptive versions. $d$ is the variable dimension. The lower bound is proven in (Arjevani et al., 2020).

| Smooth Condition | Optimizer | Separated Reg. | Batch Size Condition | Grad Bound | Complexity | Lower Bound |
|---|---|---|---|---|---|---|
| Lipschitz Gradient | Adam-type | ✗ | ✗ | $\ell_\infty \leq c_\infty$ | $\mathcal{O}(c_\infty^2 d\epsilon^{-4})$ | $\Omega(\epsilon^{-4})$ |
| | RMSProp | ✗ | ✗ | $\ell_\infty \leq c_\infty$ | $\mathcal{O}(\sqrt{c_\infty} d\epsilon^{-4})$ | $\Omega(\epsilon^{-4})$ |
| | AdamW | ✔ | — | — | — | — |
| | Adabelief | ✗ | ✗ | $\ell_2 \leq c_2$ | $\mathcal{O}(c_2^6 \epsilon^{-4})$ | $\Omega(\epsilon^{-4})$ |
| | Padam | ✗ | ✗ | $\ell_\infty \leq c_\infty$ | $\mathcal{O}(\sqrt{c_\infty} d\epsilon^{-4})$ | $\Omega(\epsilon^{-4})$ |
| | LAMB | ✗ | $\mathcal{O}(\epsilon^{-4})$ | $\ell_2 \leq c_2$ | $\mathcal{O}(c_2^2 d\epsilon^{-4})$ | $\Omega(\epsilon^{-4})$ |
| | **Adan** (ours) | ✔ | ✗ | $\ell_\infty \leq c_\infty$ | $\mathcal{O}(c_\infty^{2.5}\epsilon^{-4})$ | $\Omega(\epsilon^{-4})$ |
| Lipschitz Hessian | A-NIGT | ✗ | ✗ | $\ell_2 \leq c_2$ | $\mathcal{O}(\epsilon^{-3.5}\log\frac{c_2}{\epsilon})$ | $\Omega(\epsilon^{-3.5})$ |
| | Adam$^+$ | ✗ | $\mathcal{O}(\epsilon^{-1.625})$ | $\ell_2 \leq c_2$ | $\mathcal{O}(\epsilon^{-3.625})$ | $\Omega(\epsilon^{-3.5})$ |
| | **Adan** (ours) | ✔ | ✗ | $\ell_\infty \leq c_\infty$ | $\mathcal{O}(c_\infty^{1.25}\epsilon^{-3.5})$ | $\Omega(\epsilon^{-3.5})$ |

methods, *e.g.* LARS (You et al., 2017) and LAMB (You et al., 2019), have been proposed to handle large batch sizes, their performance often varies significantly across batch sizes. This performance inconsistency increases the training cost and engineering burden, since one usually has to try various optimizers for different architectures or training settings.

When we rethink the current adaptive gradient algorithms, we find that they mainly combine the moving average idea with the heavy ball acceleration technique to estimate the first- and second-order moments of the gradient, *e.g.* Adam, AdamW and LAMB. However, previous studies (Nesterov, 1983; 1988; 2003) have revealed that Nesterov acceleration can theoretically achieve a faster convergence speed than heavy ball acceleration, as it uses gradient at an extrapolation point of the current solution and sees a slight "future". Moreover, recent work (Nado et al., 2021; He et al., 2021) have shown the potential of Nesterov acceleration for large-batch training. Thus we are inspired to consider efficiently integrating Nesterov acceleration with adaptive algorithms.

**Contributions: 1)** We propose an efficient DNN optimizer, named Adan. Adan develops a Nesterov momentum estimation method to estimate stable and accurate first- and second-order moments of the gradient in adaptive gradient algorithms for acceleration. **2)** Moreover, Adan enjoys a provably faster convergence speed than previous adaptive gradient algorithms such as Adam. **3)** Empirically, Adan shows superior performance over the SoTA deep optimizers across vision, language, and reinforcement learning (RL) tasks. Our *detailed* contributions are highlighted below.

Firstly, we propose an efficient Nesterov-acceleration-induced deep learning optimizer termed Adan. Given a function $f$ and the current solution $\boldsymbol{\theta}_k$, Nesterov acceleration (Nesterov, 1983; 1988; 2003) estimates the gradient $\mathbf{g}_k = \nabla f(\boldsymbol{\theta}_k')$ at the extrapolation point $\boldsymbol{\theta}_k' = \boldsymbol{\theta}_k - \eta(1 - \beta_1)\mathbf{m}_{k-1}$ with the learning rate $\eta$ and momentum coefficient $\beta_1 \in (0, 1)$, and updates the moving gradient average as $\mathbf{m}_k = (1 - \beta_1)\mathbf{m}_{k-1} + \mathbf{g}_k$. Then it runs a step by $\boldsymbol{\theta}_{k+1} = \boldsymbol{\theta}_k - \eta\mathbf{m}_k$. However, the inconsistency of the positions for parameter updating at $\boldsymbol{\theta}_k$ and gradient estimation at $\boldsymbol{\theta}_k'$ leads to the additional cost of model parameter reloading during back-propagation (BP), which is unaffordable especially for large DNNs. To avoid the model reloading during BP, we propose an alternative Nesterov momentum estimation (NME). We compute the gradient $\mathbf{g}_k = \nabla f(\boldsymbol{\theta}_k)$ at the current solution $\boldsymbol{\theta}_k$, and estimate the moving gradient average as $\mathbf{m}_k = (1 - \beta_1)\mathbf{m}_{k-1} + \mathbf{g}_k'$, where $\mathbf{g}_k' = \mathbf{g}_k + (1 - \beta_1)(\mathbf{g}_k - \mathbf{g}_{k-1})$. Our NME is provably equivalent to the vanilla one yet can avoid the extra model reloading. Then by regarding $\mathbf{g}_k'$ as the current stochastic gradient in adaptive gradient algorithms, *e.g.* Adam, we accordingly estimate the first- and second-moments as $\mathbf{m}_k = (1 - \beta_1)\mathbf{m}_{k-1} + \beta_1\mathbf{g}_k'$ and $\mathbf{n}_k = (1 - \beta_2)\mathbf{n}_{k-1} + \beta_2(\mathbf{g}_k')^2$ respectively. Finally, we update $\boldsymbol{\theta}_{k+1} = \boldsymbol{\theta}_k - \eta\mathbf{m}_k/\sqrt{\mathbf{n}_k + \varepsilon}$. In this way, Adan enjoys the merit of Nesterov acceleration, namely faster convergence speed and tolerance to large mini-batch size (Lin et al., 2020), which is verified in our experiments in Sec. 5.

Secondly, as shown in Table 1, we theoretically justify the advantages of Adan over previous SoTA adaptive gradient algorithms on nonconvex stochastic problems, *e.g.* deep learning problems.

1) Given Lipschitz gradient condition, to find an $\epsilon$-approximate first-order stationary point, Adan has the stochastic gradient complexity $\mathcal{O}(c_\infty^{2.5}\epsilon^{-4})$ which accords with the lower bound $\Omega(\epsilon^{-4})$ (up to a constant factor) (Arjevani et al., 2019). This complexity is lower than $\mathcal{O}(c_2^6\epsilon^{-4})$ of Adabelief (Zhuang et al., 2020) and $\mathcal{O}(c_2^2 d\epsilon^{-4})$ of LAMB, especially on over-parameterized networks. Specifically, for the $d$-dimensional gradient, compared with its $\ell_2$ norm $c_2$, its $\ell_\infty$ norm $c_\infty$ is usually much smaller, and can be $\sqrt{d}\times$ smaller for the best case. Moreover, different from Adam-type optimizers (*e.g.* Adam), Adan can separate the $\ell_2$ regularizer with the loss objective like AdamW whose generalization benefits have been validated in many works (Touvron et al., 2021).

2) Given the Lipschitz Hessian condition, Adan has a complexity $\mathcal{O}(c_\infty^{1.25}\epsilon^{-3.5})$ which also matches the lower bound $\Omega(\epsilon^{-3.5})$ in Arjevani et al. (2020). This complexity is superior to $\mathcal{O}(\epsilon^{-3.5}\log\frac{c_2}{\epsilon})$ of A-NIGT (Cutkosky & Mehta, 2020) and also $\mathcal{O}(\epsilon^{-3.625})$ of Adam$^+$ (Liu et al., 2020). Indeed, Adam$^+$ needs the minibatch size of order $\mathcal{O}(\epsilon^{-1.625})$ which is prohibitive in practice. For other optimizers, *e.g.* Adam, their convergence has not been provided yet under Lipschitz Hessian condition.

Finally, Adan simultaneously surpasses the corresponding SoTA optimizers across vision, language, and RL tasks, and establishes new SoTAs for many networks and settings, *e.g.* ResNet, ConvNext (Liu et al., 2022b), ViT (Touvron et al., 2021), Swin (Liu et al., 2021), MAE (He et al., 2022), LSTM (Schmidhuber et al., 1997), Transformer-XL (Dai et al., 2019) and BERT (Devlin et al., 2018). More importantly, with half of the training cost (epochs) of SoTA optimizers, Adan can achieve higher or comparable performance. Besides, Adan works well in a large range of minibatch size, *e.g.* from 1k to 32k on ViTs. The improvement of Adan for various architectures and settings can greatly relieve the engineering burden by avoiding trying different optimizers.

## 2 RELATED WORK

Current DNN optimizers can be grouped into two families: SGD and its accelerated variants, and adaptive gradient algorithms. SGD computes stochastic gradient and updates the variable along the gradient direction. Later, heavy-ball acceleration (Polyak, 1964) movingly averages stochastic gradient in SGD for faster convergence. Nesterov acceleration runs a step along the moving gradient average and then computes gradient at the new point to look ahead for correction. Typically, Nesterov acceleration converges faster both empirically and theoretically at least on convex problems, and also has superior generalization resutls on DNNs (Foret et al., 2020; Kwon et al., 2021).

Unlike SGD, adaptive gradient algorithms, *e.g.* AdaGrad, RMSProp and Adam, view the second moment of gradient as a preconditioner and also use moving gradient average to update the variable. Later, many variants have been proposed to estimate a more accurate and stable first moment of gradient or its second moment, *e.g.* AMSGrad, Adabound, and Adabelief. To improve generalization, AdamW splits the objective and trivial regularization, and its effectiveness is validated across many applications; SAM and its variants (Kwon et al., 2021) aim to find flat minima but need forward and backward twice per iteration. LARS and LAMB train DNNs with a large batch but suffer unsatisfactory performance on small batch. Xie et al. (2022) reveal the generalization and convergence gap between Adam and SGD from the perspective of diffusion theory and propose the optimizers, Adai, which accelerates the training and provably favors flat minima. Padam (Chen et al., 2021a) provides a simple but effective way to improve the generalization performance of Adam by adjusting the second-order moment in Adam. The most related work to ours is NAdam. It simplifies Nesterov acceleration to estimate the first moment of gradient in Adam. But its acceleration does not use any gradient from the extrapolation points and thus does not look ahead for correction. Moreover, there is no theoretical result to ensure its convergence. See more difference discussion in Sec. 3.2.

## 3 METHODOLOGY

In this work, we study the following regularized nonconvex optimization problem:

$$\min_{\boldsymbol{\theta}} F(\boldsymbol{\theta}) := \mathbb{E}_{\boldsymbol{\zeta}\sim\mathcal{D}}\left[f(\boldsymbol{\theta},\boldsymbol{\zeta})\right] + \frac{\lambda}{2}\|\boldsymbol{\theta}\|_2^2, \qquad (1)$$

where loss $f(\cdot,\cdot)$ is differentiable and possibly nonconvex, data $\boldsymbol{\zeta}$ is drawn from an unknown distribution $\mathcal{D}$, $\boldsymbol{\theta}$ is learnable parameters, and $\|\cdot\|$ is the classical $\ell_2$ norm. At below, we first introduce the key motivation of Adan in Sec. 3.1, and then give detailed algorithmic steps in Sec. 3.2.

### 3.1 PRELIMINARIES

Adaptive gradient algorithms, Adam and AdamW, have become the default choice to train CNNs and ViTs. Unlike SGD which uses one learning rate for all gradient coordinates, adaptive algorithms adjust the learning rate for each gradient coordinate according to the current geometry curvature of the objective function, and thus converge faster. Take RMSProp and Adam as examples. Given stochastic gradient estimator $\mathbf{g}_k := \mathbb{E}_{\zeta \sim \mathcal{D}}[\nabla f(\boldsymbol{\theta}_k, \boldsymbol{\zeta})] + \boldsymbol{\xi}_k$, $e.g.$ minibatch gradient, where $\boldsymbol{\xi}_k$ is the gradient noise, RMSProp updates the variable $\boldsymbol{\theta}$ as follows:

$$\text{RMSProp:} \begin{cases} \mathbf{n}_k = (1 - \beta)\mathbf{n}_{k-1} + \beta\mathbf{g}_k^2 \\ \boldsymbol{\theta}_{k+1} = \boldsymbol{\theta}_k - \eta/(\sqrt{\mathbf{n}_k} + \varepsilon) \circ \mathbf{g}_k, \end{cases} \Rightarrow \text{Adam:} \begin{cases} \mathbf{m}_k = (1 - \beta_1)\mathbf{m}_{k-1} + \beta_1\mathbf{g}_k \\ \mathbf{n}_k = (1 - \beta_2)\mathbf{n}_{k-1} + \beta_2\mathbf{g}_k^2 \\ \boldsymbol{\theta}_{k+1} = \boldsymbol{\theta}_k - \eta/(\sqrt{\mathbf{n}_k} + \varepsilon) \circ \mathbf{m}_k, \end{cases}$$

where $\mathbf{m}_0 = \mathbf{g}_0$, $\mathbf{n}_0 = \mathbf{g}_0^2$, the scalar $\eta$ is the base learning rate, and $\circ$ denotes the element-wise product. Based on RMSProp, Adam[1] replaces the estimated gradient $\mathbf{g}_k$ with a moving average $\mathbf{m}_k$ of all previous gradient $\mathbf{g}_k$. By inspection, one can easily observe that the moving average idea in Adam is similar to the classical (stochastic) heavy-ball acceleration (HBA) technique (Polyak, 1964):

$$\text{HBA:} \quad \mathbf{g}_k = \nabla f(\boldsymbol{\theta}_k) + \boldsymbol{\xi}_k, \qquad \mathbf{m}_k = (1 - \beta_1)\mathbf{m}_{k-1} + \mathbf{g}_k, \qquad \boldsymbol{\theta}_{k+1} = \boldsymbol{\theta}_k - \eta\mathbf{m}_k.$$

Both Adam and HBA share the spirit of moving gradient average, though HBA does not have the factor $\beta_1$ on the gradient $\mathbf{g}_k$. That is, given one gradient coordinate, if its gradient directions are more consistent along the optimization trajectory, Adam/HBA accumulates a larger gradient value in this direction and thus goes ahead for a bigger gradient step, which accelerates convergence.

In addition to HBA, Nesterov's accelerated (stochastic) gradient descent (AGD) (Nesterov, 1983; 1988; 2003) is another popular acceleration technique in the optimization community:

$$\text{AGD:} \ \mathbf{g}_k = \nabla f(\boldsymbol{\theta}_k - \eta(1 - \beta_1)\mathbf{m}_{k-1}) + \boldsymbol{\xi}_k, \ \mathbf{m}_k = (1 - \beta_1)\mathbf{m}_{k-1} + \mathbf{g}_k, \ \boldsymbol{\theta}_{k+1} = \boldsymbol{\theta}_k - \eta\mathbf{m}_k. \quad (2)$$

Unlike HBA, AGD uses the gradient at the extrapolation point $\boldsymbol{\theta}_k' = \boldsymbol{\theta}_k - \eta(1 - \beta_1)\mathbf{m}_{k-1}$. Hence when the adjacent iterates share consistent gradient directions, AGD sees a slight future to converge faster. Indeed, AGD theoretically converges faster than HBA and achieves optimal convergence rate on the general smooth convex problems (Nesterov, 2003). Meanwhile, since the over-parameterized DNNs have been observed/proved to have many convex-alike local basins (Hardt & Ma, 2016; Xie et al., 2017; Li & Yuan, 2017), AGD seems more suitable than HBA for DNNs. For large-batch training, Nado et al. (2021) showed that AGD has the potential to achieve comparable performance to some specifically designed optimizers, $e.g.$ LARS and LAMB. With its advantage in convergence and large-batch training, we consider applying AGD to improve adaptive algorithms.

### 3.2 ADAPTIVE NESTEROV MOMENTUM ALGORITHM

**Main Iteration.** We temporarily set $\lambda = 0$ in Eqn. (1). As aforementioned, AGD computes gradient at an extrapolation point $\boldsymbol{\theta}_k'$ instead of the current iterate $\boldsymbol{\theta}_k$, which however brings extra computation and memory overhead for computing $\boldsymbol{\theta}_k'$ and preserving both $\boldsymbol{\theta}_k$ and $\boldsymbol{\theta}_k'$. To solve the issue, Lemma 1 with proof in Appendix D reformulates AGD (2) into its equivalent but more DNN-efficient version.

**Lemma 1.** *Assume* $\mathbb{E}(\boldsymbol{\xi}_k) = \mathbf{0}$, $\text{Cov}(\boldsymbol{\xi}_i, \boldsymbol{\xi}_j) = 0$ *for any* $k, i, j > 0$, $\bar{\boldsymbol{\theta}}_k$ *and* $\bar{\mathbf{m}}_k$ *be the iterate and momentum of the vanilla AGD in Eqn.* (2), *respectively. Let* $\boldsymbol{\theta}_{k+1} := \bar{\boldsymbol{\theta}}_{k+1} - \eta(1 - \beta_1)\bar{\mathbf{m}}_k$ *and* $\mathbf{m}_k := (1 - \beta_1)^2\bar{\mathbf{m}}_{k-1} + (2 - \beta_1)(\nabla f(\boldsymbol{\theta}_k) + \boldsymbol{\xi}_k)$. *The vanilla AGD in Eqn.* (2) *becomes AGD-II:*

$$\mathbf{g}_k = \mathbb{E}_{\zeta \sim \mathcal{D}}[\nabla f(\boldsymbol{\theta}_k, \boldsymbol{\zeta})] + \boldsymbol{\xi}_k, \ \mathbf{m}_k = (1 - \beta_1)\mathbf{m}_{k-1} + [\mathbf{g}_k + (1 - \beta_1)(\mathbf{g}_k - \mathbf{g}_{k-1})], \ \boldsymbol{\theta}_{k+1} = \boldsymbol{\theta}_k - \eta\mathbf{m}_k.$$

*Moreover, if vanilla AGD in Eqn.* (2) *converges, so does AGD-II, and* $\mathbb{E}(\boldsymbol{\theta}_\infty) = \mathbb{E}(\bar{\boldsymbol{\theta}}_\infty)$.

The main idea in Lemma 1 is that we maintain $(\boldsymbol{\theta}_k - \eta(1 - \beta_1)\mathbf{m}_{k-1})$ rather than $\boldsymbol{\theta}_k$ in vanilla AGD at each iteration, since there is no difference between them when the algorithm converges. Like other adaptive optimizers, by regarding $\mathbf{g}_k' = \mathbf{g}_k + (1 - \beta_1)(\mathbf{g}_k - \mathbf{g}_{k-1})$ as the current stochastic gradient and movingly averaging $\mathbf{g}_k'$ to estimate the first- and second-moments of gradient, we obtain

$$\text{Vanilla Adan:} \begin{cases} \mathbf{m}_k = (1 - \beta_1)\mathbf{m}_{k-1} + \beta_1[\mathbf{g}_k + (1 - \beta_1)(\mathbf{g}_k - \mathbf{g}_{k-1})] \\ \mathbf{n}_k = (1 - \beta_3)\mathbf{n}_{k-1} + \beta_3[\mathbf{g}_k + (1 - \beta_1)(\mathbf{g}_k - \mathbf{g}_{k-1})^2] \\ \boldsymbol{\theta}_{k+1} = \boldsymbol{\theta}_k - \boldsymbol{\eta}_k \circ \mathbf{m}_k \ \text{with} \ \boldsymbol{\eta}_k = \eta/(\sqrt{\mathbf{n}_k} + \varepsilon). \end{cases}$$

---

[1]For presentation convenience, we omit the de-bias term in adaptive gradient methods.

The main difference of Adan with Adam-type methods and Nadam ([Dozat, 2016](#)) is that as compared in Eqn. (3), the momentum $\mathbf{m}_k$ of Adan is the average of $\{\mathbf{g}_t + (1 - \beta_1)(\mathbf{g}_t - \mathbf{g}_{t-1})\}_{t=1}^k$ while those of Adam-type and Nadam are the average of $\{\mathbf{g}_t\}_{t=1}^k$. So is their second-order term $\mathbf{n}_k$.

$$\mathbf{m}_k = \begin{cases} \sum_{t=0}^k c_{k,t}[\mathbf{g}_t + (1-\beta_1)(\mathbf{g}_t - \mathbf{g}_{t-1})], & \text{Adan}, \\ \sum_{t=0}^k c_{k,t}\mathbf{g}_t, & \text{Adam}, \\ \frac{\mu_{k+1}}{\mu'_{k+1}}\left(\sum_{t=0}^k c_{k,t}\mathbf{g}_t\right) + \frac{1-\mu_k}{\mu'_k}\mathbf{g}_k, & \text{Nadam}, \end{cases} \quad c_{k,t} = \begin{cases} \beta_1(1-\beta_1)^{(k-t)} & t > 0, \\ \\ (1-\beta_1)^k & t = 0, \end{cases} \quad (3)$$

where $\{\mu_t\}_{t=1}^\infty$ is a predefined exponentially decaying sequence, $\mu'_k = 1 - \prod_{t=1}^k \mu_t$. So Nadam is more like Adam than Adan, as their $\mathbf{m}_k$ movingly averages the historical gradients instead of gradient differences in Adan. For a large $k$ (i.e. small $\mu_k$), $\mathbf{m}_k$ in Nadam and Adam are almost the same.

As shown in Eqn. (3), the moment $\mathbf{m}_k$ in Adan consists of two terms, i.e. gradient term $\mathbf{g}_t$ and gradient difference term $(\mathbf{g}_t - \mathbf{g}_{t-1})$, which actually have different physic meanings. So here we decouple them for greater flexibility and also better trade-off between them. Specifically, we estimate

$$(\boldsymbol{\theta}_{k+1} - \boldsymbol{\theta}_k)/\boldsymbol{\eta}_k = \sum_{t=0}^k \left[ c_{k,t}\mathbf{g}_t + (1-\beta_2)c'_{k,t}(\mathbf{g}_t - \mathbf{g}_{t-1}) \right] = \mathbf{m}_k + (1-\beta_2)\mathbf{v}_k, \quad (4)$$

where $c'_{k,t} = \beta_2(1-\beta_2)^{(k-t)}$ for $t > 0$, $c'_{k,t} = (1-\beta_2)^k$ for $t = 0$, and $\mathbf{m}_k$ and $\mathbf{v}_k$ are defined as

$$\mathbf{m}_k = (1-\beta_1)\mathbf{m}_{k-1} + \beta_1\mathbf{g}_k, \qquad \mathbf{v}_k = (1-\beta_2)\mathbf{v}_{k-1} + \beta_2(\mathbf{g}_k - \mathbf{g}_{k-1}).$$

This change for a flexible estimation does not impair convergence speed. As we show in Theorem 1, the complexity of Adan under this change matches the lower complexity bound. We do not separate the gradients and their difference in the second-order moment $\mathbf{n}_k$, since $\mathbb{E}(\mathbf{n}_k)$ contains the correlation term $\mathrm{Cov}(\mathbf{g}_k, \mathbf{g}_{k-1}) \neq 0$ which may have statistical significance.

**Decay Weight by Proximation.** As observed in AdamW, decoupling the optimization objective and simple-type regularization (*e.g.* $\ell_2$ regularizer) can largely improve the generalization performance. Here we follow this idea but from a rigorous optimization perspective. Intuitively, at each iteration, we minimize the first-order approximation of $F(\cdot)$ at the point $\boldsymbol{\theta}_k$:

$$\boldsymbol{\theta}_{k+1} = \boldsymbol{\theta}_k - \boldsymbol{\eta}_k \circ \bar{\mathbf{m}}_k = \arg\min_{\boldsymbol{\theta}} F(\boldsymbol{\theta}_k) + \langle \bar{\mathbf{m}}_k, \boldsymbol{\theta} - \boldsymbol{\theta}_k \rangle + \frac{1}{2\eta}\|\boldsymbol{\theta} - \boldsymbol{\theta}_k\|^2_{\sqrt{\mathbf{n}_k}},$$

where $\|\mathbf{x}\|^2_{\sqrt{\mathbf{n}_k}} := \left\langle \mathbf{x}, \left(\sqrt{\mathbf{n}_k} + \varepsilon\right) \circ \mathbf{x} \right\rangle$ and $\bar{\mathbf{m}}_k := \mathbf{m}_k + (1-\beta_2)\mathbf{v}_k$ is the first-order derivative of $F(\cdot)$ in some sense. Follow the idea of proximal gradient descent ([Parikh & Boyd, 2014](#); [Zhuang et al., 2022](#)), we decouple the $\ell_2$ regularizer from $F(\cdot)$ and only linearize the loss function $f(\cdot)$:

$$\boldsymbol{\theta}_{k+1} = \arg\min_{\boldsymbol{\theta}} \left( \frac{\lambda_k}{2}\|\boldsymbol{\theta}\|^2_{\sqrt{\mathbf{n}_k}} + \langle \bar{\mathbf{m}}_k, \boldsymbol{\theta} - \boldsymbol{\theta}_k \rangle + \frac{1}{2\eta}\|\boldsymbol{\theta} - \boldsymbol{\theta}_k\|^2_{\sqrt{\mathbf{n}_k}} \right) = \frac{\boldsymbol{\theta}_k - \boldsymbol{\eta}_k \circ \bar{\mathbf{m}}_k}{1 + \lambda_k \eta}, \quad (5)$$

where $\lambda_k > 0$ is the weight decay at the $k$-th iteration. One can find that the optimization objective of at the $k$-th iteration is changed from the vanilla "static" function $F(\cdot)$ in Eqn. (1) to a "dynamic" function $F_k(\cdot)$ in Eqn. (6), which adaptively regularizes the coordinates with larger gradient more:

$$F_k(\boldsymbol{\theta}) := \mathbb{E}_{\boldsymbol{\zeta} \sim \mathcal{D}}\left[f(\boldsymbol{\theta}, \boldsymbol{\zeta})\right] + \frac{\lambda_k}{2}\|\boldsymbol{\theta}\|^2_{\sqrt{\mathbf{n}_k}}. \quad (6)$$

We summarize our Adan in Algorithm 1. We reset the momentum term properly by the restart condition, a common trick to stabilize optimization and benefit convergence ([Li & Lin, 2022](#); [Jin et al., 2018](#)). But to make Adan simple, in all experiments except Table 8, we do not use this restart strategy although it can improve performance as shown in Table 8.

## 4 CONVERGENCE ANALYSIS

For analysis, we make several mild assumptions used in many works, *e.g.* ([Guo et al., 2021](#)).

**Assumption 1** ($L$-smoothness). *The function $f(\cdot, \cdot)$ is $L$-smooth w.r.t. the parameter, if $\exists L > 0$,*

$$\|\nabla \mathbb{E}_{\boldsymbol{\zeta}}[f(\mathbf{x}, \boldsymbol{\zeta})] - \nabla \mathbb{E}_{\boldsymbol{\zeta}}[f(\mathbf{y}, \boldsymbol{\zeta})]\| \leq L\|\mathbf{x} - \mathbf{y}\|, \qquad \forall \mathbf{x}, \mathbf{y}.$$

**Assumption 2** (Unbiased and bounded gradient oracle). *The stochastic gradient oracle $\mathbf{g}_k = \mathbb{E}_{\boldsymbol{\zeta}}[\nabla f(\boldsymbol{\theta}_k, \boldsymbol{\zeta})] + \boldsymbol{\xi}_k$ is unbiased, and its magnitude and variance are bounded with probability 1:*

$$\mathbb{E}(\boldsymbol{\xi}_k) = \mathbf{0}, \quad \|\mathbf{g}_k\|_\infty \leq c_\infty/3, \quad \mathbb{E}\left(\|\boldsymbol{\xi}_k\|^2\right) = \mathbb{E}\left(\|\nabla \mathbb{E}_{\boldsymbol{\zeta}}[f(\boldsymbol{\theta}_k, \boldsymbol{\zeta})] - \mathbf{g}_k\|^2\right) \leq \sigma^2, \quad \forall k \in [T].$$

---

**Algorithm 1: Adan** (Adaptive Nesterov Momentum Algorithm)

**Input:** initialization $\boldsymbol{\theta}_0$, step size $\eta$, weight decay $\lambda_k > 0$, restart condition.
**Output:** some average of $\{\boldsymbol{\theta}_k\}_{k=1}^K$.

1 **while** $k < K$ **do**
2     compute the stochastic gradient estimator $\mathbf{g}_k$ at $\boldsymbol{\theta}_k$;
3     $\mathbf{m}_k = (1 - \beta_1)\mathbf{m}_{k-1} + \beta_1 \mathbf{g}_k$          /* set $\mathbf{m}_0 = \mathbf{g}_0$ */;
4     $\mathbf{v}_k = (1 - \beta_2)\mathbf{v}_{k-1} + \beta_2(\mathbf{g}_k - \mathbf{g}_{k-1})$      /* set $\mathbf{v}_1 = \mathbf{g}_1 - \mathbf{g}_0$ */;
5     $\mathbf{n}_k = (1 - \beta_3)\mathbf{n}_{k-1} + \beta_3[\mathbf{g}_k + (1 - \beta_2)(\mathbf{g}_k - \mathbf{g}_{k-1})]^2$;
6     $\boldsymbol{\theta}_{k+1} = (1 + \lambda_k\eta)^{-1}[\boldsymbol{\theta}_k - \boldsymbol{\eta}_k \circ (\mathbf{m}_k + (1 - \beta_2)\mathbf{v}_k)]$ with $\boldsymbol{\eta}_k = \eta/(\sqrt{\mathbf{n}_k} + \varepsilon)$;
7     **if** *restart condition holds* **then**
8        get stochastic gradient estimator $\mathbf{g}_0$ at $\boldsymbol{\theta}_{k+1}$;
9        $\mathbf{m}_0 = \mathbf{g}_0$,    $\mathbf{v}_0 = \mathbf{0}$,    $\mathbf{n}_0 = \mathbf{g}_0^2$,    update $\boldsymbol{\theta}_1$ by Line 6,    $k = 1$;
10     **end if**
11 **end while**

---

**Assumption 3** ($\rho$-Lipschitz continuous Hessian). *The function $f(\cdot, \cdot)$ has $\rho$-continuous Hessian:*

$$\left\|\nabla^2 \mathbb{E}_{\boldsymbol{\zeta}}[f(\mathbf{x}, \boldsymbol{\zeta})] - \nabla^2 \mathbb{E}_{\boldsymbol{\zeta}}[f(\mathbf{x}, \boldsymbol{\zeta})]\right\| \leq \rho\|\mathbf{x} - \mathbf{y}\|, \qquad \forall \mathbf{x}, \mathbf{y},$$

*where $\|\cdot\|$ is the spectral norm for matrix and the $\ell_2$ norm for vector.*

For a general nonconvex problem, if Assumptions 1 and 2 hold, the lower bound of the stochastic gradient complexity to find an $\epsilon$-approximate first-order stationary point ($\epsilon$-ASP) is $\Omega(\epsilon^{-4})$ (Arjevani et al., 2019; 2020). Moreover, if Assumption 3 further holds, the lower complexity bound becomes $\Omega(\epsilon^{-3.5})$ for a non-variance-reduction algorithm (Arjevani et al., 2019; 2020).

**Lipschitz Gradient.** Theorem 1 with proof in Appendix E proves the convergence of Adan on problem (6) with lipschitz gradient condition.

**Theorem 1.** *Suppose Assumptions 1 and 2 hold. Let $\max\{\beta_1, \beta_2\} = \mathcal{O}(\epsilon^2)$, $\mu := \sqrt{2}\beta_3 c_\infty/\varepsilon \ll 1$, $\eta = \mathcal{O}(\epsilon^2)$, and $\lambda_k = \lambda(1 - \mu)^k$. Algorithm 1 runs at most $K = \Omega(c_\infty^{2.5}\epsilon^{-4})$ iterations to achieve*

$$\frac{1}{K+1}\sum_{k=0}^{K}\mathbb{E}\left(\|\nabla F_k(\boldsymbol{\theta}_k)\|^2\right) \leq 4\epsilon^2.$$

*That is, to find an $\epsilon$-ASP, the stochastic gradient complexity of Adan on problem (6) is $\mathcal{O}(c_\infty^{2.5}\epsilon^{-4})$.*

Theorem 1 shows that under Assumptions 1 and 2, Adan can converge to an $\epsilon$-ASP of a nonconvex stochastic problem with stochastic gradient complexity $\mathcal{O}(c_\infty^{2.5}\epsilon^{-4})$ which accords with the lower bound $\Omega(\epsilon^{-4})$ in Arjevani et al. (2019). For this convergence, Adan has no requirement on minibatch size and only assumes gradient estimation to be unbiased and bounded. Moreover, as shown in Table 1 in Sec. 1, the complexity of Adan is superior to those of previous adaptive gradient algorithms. For Adabelief and LAMB, Adan always has lower complexity and respectively enjoys $d^3\times$ and $d^2\times$ lower complexity for the worst case. Adam-type optimizers (*e.g.* Adam and AMSGrad) enjoy the same complexity as Adan. But they cannot separate the $\ell_2$ regularizer with the objective like AdamW and our Adan. The regularizer separation can boost generalization performance (Touvron et al., 2021; Liu et al., 2021) and already helps AdamW dominate training of ViT-alike architectures. Besides, some previous analyses (Luo et al., 2018; Zaheer et al., 2018; Liu et al., 2019a; Shi et al., 2020) need the momentum coefficient (i.e. $\beta$s) to be close or increased to one, which contradicts with the practice that $\beta$s are close to zero. In contrast, Theorem 1 assumes that all $\beta$s are very small, which is more consistent with the practice. Note that when $\mu = c/T$, we have $\lambda_k/\lambda \in [(1 - c), 1]$ during training. Hence we could choose the $\lambda_k$ as a fixed constant in the experiment for convenience.

**Lipschitz Hessian.** With Assumption 3, we further need a restart condition. Consider an extension point $\mathbf{y}_{k+1} := \boldsymbol{\theta}_{k+1} + \boldsymbol{\eta}_k \circ [\mathbf{m}_k + (1 - \beta_2)\mathbf{v}_k - \beta\mathbf{g}_k]$, and a restart condition:

$$(k + 1)\sum_{t=0}^{k}\|\mathbf{y}_{t+1} - \mathbf{y}_t\|_{\sqrt{\mathbf{n}_t}}^2 > R^2, \qquad (7)$$

where the constant $R$ controls the restart frequency. Intuitively, when the parameters have accumulated enough updates, the iterate may reach a new local basin. Resetting the momentum at this moment helps Adan to better use the local geometric information. Besides, we change $\boldsymbol{\eta}_k$ from $\eta/(\sqrt{\mathbf{n}_k} + \varepsilon)$ to $\eta/(\sqrt{\mathbf{n}_{k-1}} + \varepsilon)$ to ensure $\boldsymbol{\eta}_k$ to be independent of noise $\boldsymbol{\zeta}_k$. See its proof in Appendix F.

Table 2: Top-1 accuracy (%) of ResNet and ConvNext on ImageNet under their official settings. $*$ and $\diamond$ are respectively reported in (Wightman et al., 2021; Liu et al., 2022b).

| Epoch | ResNet-50 | | | ResNet-101 | | |
|---|---|---|---|---|---|---|
| | 100 | 200 | 300 | 100 | 200 | 300 |
| SAM | 77.3 | 78.7 | 79.4 | 79.5 | 81.1 | 81.6 |
| SGD-M | 77.0 | 78.6 | 79.3 | 79.3 | 81.0 | 81.4 |
| Adam | 76.9 | 78.4 | 78.8 | 78.4 | 80.2 | 80.6 |
| AdamW | 77.0 | 78.9 | 79.3 | 78.9 | 79.9 | 80.4 |
| LAMB | 77.0 | 79.2 | 79.8* | 79.4 | 81.1 | 81.3* |
| **Adan (ours)** | **78.1** | **79.7** | **80.2** | **79.9** | **81.6** | **81.8** |

| Epoch | ConvNext Tiny | |
|---|---|---|
| | 150 | 300 |
| AdamW | 81.2 | 82.1$\diamond$ |
| **Adan (ours)** | **81.7** | **82.4** |
| Epoch | ConvNext Small | |
| | 150 | 300 |
| AdamW | 82.2 | 83.1$\diamond$ |
| **Adan (ours)** | **82.5** | **83.3** |

**Theorem 2.** *Suppose Assumptions 1-3 hold. Let $R = \mathcal{O}(\epsilon^{0.5})$, $\max\{\beta_1, \beta_2\} = \mathcal{O}(\epsilon^2)$, $\beta_3 = \mathcal{O}(\epsilon^4)$, $\eta = \mathcal{O}(\epsilon^{1.5})$, $K = \mathcal{O}(\epsilon^{-2})$, $\lambda = 0$. Then Algorithm 1 with restart condition Eqn.(7) satisfies:*

$$\mathbb{E}\left(\left\|\nabla F_k(\bar{\boldsymbol{\theta}})\right\|\right) = \mathcal{O}(c_\infty^{0.5}\epsilon),$$

*where $\bar{\boldsymbol{\theta}} := \frac{1}{K_0}\sum_{k=1}^{K_0}\boldsymbol{\theta}_k$, $\quad K_0 = \operatorname{argmin}_{\lfloor\frac{K}{2}\rfloor \leq k \leq K-1}\|\mathbf{y}_{t+1} - \mathbf{y}_t\|^2_{\sqrt{\mathbf{n}_t}}$. Moreover, to find an $\epsilon$-ASP, Algorithm 1 restarts at most $\mathcal{O}(c_\infty^{0.5}\epsilon^{-1.5})$ times in which each restarting cycle has at most $K = \mathcal{O}(\epsilon^{-2})$ iterations, and hence needs at most $\mathcal{O}(c_\infty^{1.25}\epsilon^{-3.5})$ stochastic gradient complexity.*

From Theorem 2, one can observe that with an extra smooth Hessian condition in Assumption 3 and a restart condition (7), Adan improves its vanilla stochastic gradient complexity from $\tilde{\mathcal{O}}(c_\infty^{2.5}\epsilon^{-4})$ to $\mathcal{O}(c_\infty^{1.25}\epsilon^{-3.5})$, which also matches the corresponding lower bound $\Omega(\epsilon^{-3.5})$. This complexity is lower than $\mathcal{O}(\epsilon^{-3.5}\log\frac{c_2}{\epsilon})$ of A-NIGT and $\mathcal{O}(\epsilon^{-3.625})$ of Adam$^+$. For other DNN optimizers, *e.g.* Adam, their convergence under Lipschitz Hessian condition has not been proved yet.

Moreover, Theorem 2 still holds for the large batch size. For example, by using minibatch size $b = \mathcal{O}(\epsilon^{-1.5})$, our results still hold when $R = \mathcal{O}(\epsilon^{0.5})$, $\max\{\beta_1, \beta_2\} = \mathcal{O}(\epsilon^{0.5})$, $\beta_3 = \mathcal{O}(\epsilon)$, $\eta = \mathcal{O}(1)$, $K = \mathcal{O}(\epsilon^{-0.5})$ and $\lambda = 0$. In this case, our $\eta$ is of the order $\mathcal{O}(1)$, and is much larger than $\mathcal{O}(\text{ploy}(\epsilon))$ of other optimizers (*e.g.*, LAMB and Adam$^+$) for handling large minibatch. This large step size often boosts convergence speed in practice, which is actually desired.

## 5 EXPERIMENTAL RESULTS

We evaluate Adan on vision, NLP and RL tasks. For vision tasks, we test Adan on several representative SoTA backbones under the supervised settings, including 1) CNN-type architectures (ResNets and ConvNexts (Liu et al., 2022b)) and 2) ViTs vanilla ViTs and Swins (Liu et al., 2021)). Moreover, we also investigate Adan via the self-supervised pretraining by using it to train MAE ViT (He et al., 2022). For NLP tasks, we train LSTM, Transformer-XL (Dai et al., 2019), and BERT (Devlin et al., 2018) for sequence modeling. On RL tasks, we evaluate Adan on four games in MuJoCo (Todorov et al., 2012). In all experiments, we only replace the optimizer with Adan and tune the step size, warmup epochs, and weight decay, *etc*, while fixing the optimizer-independent hyper-parameters, *e.g.* data augmentation and model architectures. Moreover, to make Adan simple, in all experiments except Table 8, we do not use the restart strategy in Algorithm 1. **Due to space limitation, we defer the RL results and the ablation study into Appendix B.3 and B.5, respectively.**

### 5.1 EXPERIMENTS FOR VISION TASKS

Besides the vanilla supervised training setting used in ResNets (He et al., 2016), we further consider two prevalent training settings on ImageNet, namely the following Training Setting I and II.

**Training Setting I.** The recently proposed "A2 training recipe" in (Wightman et al., 2021) has lifted the performance limits of many SoTA CNN-type architectures by stronger data augmentation. Specifically, for data augmentation, this setting uses random crop, horizontal flipping, Mixup (0.1)/CutMix (1.0) with probability 0.5, and RandAugment with $M = 7, N = 2$ and MSTD $= 0.5$. It sets stochastic depth (0.05), and adopts cosine learning rate decay and binary cross-entropy loss.

**Training Setting II.** For this setting, data augmentation includes random crop, horizontal flipping, Mixup (0.8), CutMix (1.0), RandAugment ($M = 9$, MSTD $= 0.5$) and Random Erasing ($p = 0.25$). It uses cross-entropy loss, cosine decay, and stochastic depth. For both settings, please refer to their details, *e.g.* data augmentation, in Appendix Sec. A.1.

Table 3: Top-1 accuracy (%) of ViT and Swin on ImageNet. We use their official Training Setting II to train them. ∗ and ⋄ are respectively reported in (Touvron et al., 2021; Liu et al., 2021).

| Epoch | ViT Small 150 | ViT Small 300 | ViT Base 150 | ViT Base 300 | Swin Tiny 150 | Swin Tiny 300 | Swin small 150 | Swin small 300 | Swin Base 150 | Swin Base 300 |
|---|---|---|---|---|---|---|---|---|---|---|
| AdamW | 78.3 | 79.9* | 79.5 | 81.8* | 79.9 | 81.2⋄ | 82.1 | 83.2⋄ | 82.6 | 83.5⋄ |
| **Adan (ours)** | **79.6** | **80.9** | **81.7** | **82.3** | **81.3** | **81.6** | **82.9** | **83.7** | **83.3** | **83.8** |

**Results on CNN-type Architectures.** To train ResNet and ConvNext, we respectively use their official Training Setting I and II. *For SoTA ResNet/ConvNext, its default official optimizer is LAMB/AdamW.* From Table 2, one can observe that on ResNet, 1) in most cases, Adan only running 200 epochs can achieve higher or comparable top-1 accuracy on ImageNet compared with the official SoTA result trained by LAMB with 300 epochs; 2) Adan gets more improvements over other optimizers, when training is insufficient, *e.g.* 100 epochs. The possible reason for observation 1) is the regularizer separation, which can dynamically adjust the weight decay for each coordinate instead of sharing a common one. For observation 2), this can be explained by the faster convergence speed of Adan than other optimizers. As shown in Figure 1, Adan converges faster than many adaptive gradient optimizers. This faster speed partially comes from its large learning rate guaranteed by Theorem 2, almost 3× larger than that of LAMB. The same as Nesterov acceleration, Adan could look ahead for possible corrections. More comparison on convergence speed and loss curve is in Appendix B.2. On ConvNext, one can observe similar comparison results on ResNets. Additional results in Appendix Sec. B.1 provide more comparison on ResNet-18 under the vanilla setting in (He et al., 2016).

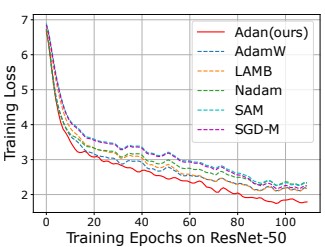

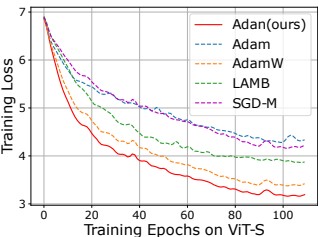

Figure 1: Training curves of various optimizers.

**Results on ViTs. 1) Supervised Training.** We train ViT and Swin under their official training setting, i.e. Training Setting II. Table 3 shows that across different model sizes of ViT and Swin, Adan outperforms the official AdamW optimizer by a large margin. For ViTs, their gradient per iteration differs much from the previous one due to the much sharper loss landscape than CNNs (Chen et al., 2021b) and the strong random augmentations for training. So it is hard to train ViTs to converge within a few epochs. Thanks to its faster convergence, as shown in Figure 1, Adan is very suitable for this situation. Moreover, the direction correction term from the gradient difference $\mathbf{v}_k$ of Adan can also better correct the first- and second-order moments. One piece of evidence is that the first-order moment decay coefficient $\beta_1 = 0.02$ of Adan is much smaller than $0.1$ used in other deep optimizers.

**2) Self-supervised MAE Training (pretraining + finetuning).** We follow the MAE training framework to pretrain and fine-tune ViT-B for 300/800 pretraining epochs and 100 fine-tuning epochs, and ViT-L for 800 pretraining epochs and 50 fine-tuning epochs, on ImageNet. Table 4 shows that 1) on ViT-B, Adan makes 0.5% improvement over AdamW under 300 pretraining epochs, and Adan pretrained 800 epochs surpasses AdamW pretrained 1,600 epochs by nontrivial 0.2%;

Table 4: Top-1 Acc. (%) of ViT-B and ViT-L trained by MAE under the official Training Setting II. ∗ and ⋄ are respectively reported in (Chen et al., 2022; He et al., 2022).

| Epoch | MAE-ViT-B 300 | MAE-ViT-B 800 | MAE-ViT-B 1600 | MAE-ViT-L 800 | MAE-ViT-L 1600 |
|---|---|---|---|---|---|
| AdamW | 82.9* | — | 83.6⋄ | 85.4⋄ | 85.9⋄ |
| **Adan** | **83.4** | **83.8** | — | **85.9** | — |

2) on ViT-L, Adan only uses 800 pretraining epochs to achieve the same performance of AdamW with 1,600 pretraining epochs. All these results show a superior performance of Adan.

**3) Large-Batch Training.** Though large batch size can increase parallelism to reduce training time and is heavily desired, optimizers often suffer performance degradation, or even fail. For instance, AdamW fails to train ViTs when batch size is beyond 4,096. How to solve the problem remains open (He et al., 2021).

Table 5: Top-1 Acc. (%) of ViT-S on ImageNet under Training Setting I.

| Batch Size | 1k | 2k | 4k | 8k | 16k | 32k |
|---|---|---|---|---|---|---|
| LAMB | 78.9 | 79.2 | 79.8 | 79.7 | 79.5 | 78.4 |
| **Adan** | **80.9** | **81.1** | **81.1** | **80.8** | **80.5** | **80.2** |

At present, LAMB is the most effective optimizer for large batch size. Table 5 reveals that Adan is robust to batch sizes from 2k to 32k, and shows higher performance and robustness than LAMB.

Table 6: Results (the higher, the better) of BERT-base model on the development set of GLUE. The first line is from (Wolf et al., 2020) while the second line is reproduced by us.

| BERT-base | MNLI | QNLI | QQP | RTE | SST-2 | CoLA | STS-B | **Average** |
|---|---|---|---|---|---|---|---|---|
| Adam (official) | 83.7/84.8 | 89.3 | 90.8 | 71.4 | 91.7 | 48.9 | 91.3 | 81.5 |
| Adam (reproduced) | 84.9/84.9 | 90.8 | 90.9 | 69.3 | 92.6 | 58.5 | 88.7 | 82.5 |
| **Adan (ours)** | **85.7/85.6** | **91.3** | **91.2** | **73.3** | **93.2** | **64.6** | **89.3** | **84.3 (+1.8)** |

## 5.2 EXPERIMENTS FOR NATURAL LANGUAGE PROCESSING TASKS

**Results on BERT.** Similar to the pretraining experiments of MAE which is also a self-supervised learning framework on vision tasks, we utilize Adan to train BERT (Devlin et al., 2018) from scratch, which is one of the most widely used pretraining models/frameworks for NLP tasks. We employ the exact BERT training setting in the widely used codebase—Fairseq (Ott et al., 2019). See more training details in Appendix A.3.

From Table 6, one can see that in the most commonly used BERT training experiment, Adan reveals much better advantage over Adam. Specifically, in all GLUE tasks, on BERT-base model, Adan achieves higher performance than Adam, and makes 1.8 average improvements on all tasks. In addition, on some tasks of Adan, BERT-base trained by Adan can outperform some large models. e.g., BERT-large which achieves 70.4% on RTE, 93.2% on SST-2 and 60.6 correlation on CoLA, and XLNet-large which has 63.6 correlation on CoLA. See (Liu et al., 2019b) for more results.

**Results on Transformer-XL.** We evaluate Adan on Transformer-XL (Dai et al., 2019) which is often used to model long sequences. We follow the exact official setting [2] to train Transformer-XL-base on the WikiText-103 dataset that is the largest available word-level language modeling benchmark with long-term dependency. We only replace the default Adam optimizer of Transformer-XL-base by our Adan, and do not make other changes for the hyper-parameter. For Adan, we set $\beta_1 = 0.1, \beta_2 = 0.1$, and $\beta_3 = 0.001$, and choose learning rate as 0.001. We test Adan and Adam with several training steps, including 50k, 100k, and 200k (official).

Table 7: Test PPL (the lower, the better) for Transformer-XL-base model on the WikiText-103 dataset.

| Transformer-XL | Training Steps | | |
|---|---|---|---|
| | 50k | 100k | 200k |
| Adam | 28.5 | 25.5 | 24.2 |
| **Adan (ours)** | **26.2** | **24.2** | **23.5** |

Table 7 shows that on Transformer-XL-base, Adan surpasses its default Adam optimizer in terms of test PPL (the lower, the better) under all training steps. Surprisingly, Adan using 100k training steps can even achieve comparable results to Adam with 200k training steps. All these results demonstrate the superiority of Adan over the default SoTA Adam optimizer in Transformer-XL.

**Results on LSTM.** In Appendix B.4, the results on LSTM shows the superiority of our Adan over several representative optimizers, *e.g.* SGD, Adam and AdamW, on the Penn TreeBank dataset.

## 5.3 DISCUSSION ON RESTART STRATEGY

Here we investigate the performance Adan with and without restart strategy on ViT and ConvNext under 300 training epochs. From the results in Table 8, one can observe that the restart strategy slightly improves test performance of Adan on both ViT and ConvNext. However, to make our Adan simple and avoid hyper-parameter tuning of the restart strategy (e.g., restart frequency), in all experiments except Table 8, we do not use this restart strategy.

Table 8: Top-1 Acc. (%) of ViT-S and ConvNext-T on ImageNet under Training Setting II trained with 300 epochs.

| | ViT Small | ConvNext Tiny |
|---|---|---|
| Adan w/o restart | 80.71 | 81.38 |
| Adan w/ restart | **80.87** | **81.62** |

## 6 CONCLUSION

In this paper, we propose a new deep optimizer, Adan. We reformulate the vanilla AGD to a more efficient version and use it to estimate the first- and second-order moments in adaptive optimization algorithms. We prove that the complexity of Adan matches the lower bounds and is superior to those of other adaptive optimizers. Finally, extensive experimental results demonstrate that Adan consistently surpasses other optimizers on many popular backbones and frameworks, including ResNet, ConvNext, ViT, Swin, MAE-ViT, LSTM, Transformer-XL and BERT.

---

[2] https://github.com/kimiyoung/transformer-xl

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

APPENDIX

The appendix contains some additional experimental results and the technical proofs of convergence results of the paper entitled "Adan: Adaptive Nesterov Momentum Algorithm for Faster Optimizing Deep Models". It is structured as follows. Sec. A provides details of the training setting and Adan's implementation. It also gives detailed steps to perform the experiment on BERT. Sec. B include the additional experimental results, which contains the results on ResNet-18 in Sec. B.1, convergence curve in Sec. B.2, experiments on RL tasks in Sec. B.3, results on LSTM in Sec. B.4, and the ablation study in Sec. B.5.

After Sec. C, which summarizes the notations throughout this document, we provide the technical proofs of convergence results. Then Sec. D provides the proof of the equivalence between AGD and reformulated AGD, i.e., the proof of Lemma 1. And then, given Lipschitz gradient condition, Sec. E provides the convergence analysis in Theorem 1. Next, we show Adan's faster convergence speed with Lipschitz Hessian condition in Sec. F, by first reformulating our Algorithm 1 and introducing some auxiliary bounds. Finally, we present some auxiliary lemmas in Sec. G.

## A  TRAINING SETTING AND IMPLEMENTATION DETAILS

### A.1  TRAINING SETTING

**Training Setting I.** The recently proposed "A2 training recipe" in (Wightman et al., 2021) has pushed the performance limits of many SoTA CNN-type architectures by using stronger data augmentation and more training iterations. For example, on ResNet50, it sets new SoTA $80.4\%$, and improves the accuracy $76.1\%$ under vanilla setting in (He et al., 2016). Specifically, for data augmentation, this setting uses random crop, horizontal flipping, Mixup (0.1) (Zhang et al., 2018)/CutMix (1.0) (Yun et al., 2019) with probability $0.5$, and RandAugment (Cubuk et al., 2020) with $M = 7, N = 2$ and MSTD $= 0.5$. It sets stochastic depth $(0.05)$ (Huang et al., 2016), and adopts cosine learning rate decay and binary cross-entropy (BCE) loss. For Adan, we use batch size 2048 for ResNet and ViT.

**Training Setting II.** We follow the same official training procedure of ViT/Swin/ConvNext. For this setting, data augmentation includes random crop, horizontal flipping, Mixup (0.8), CutMix (1.0), RandAugment ($M = 9$, MSTD $= 0.5$) and Random Erasing ($p = 0.25$). We use CE loss, the cosine decay for base learning rate, the stochastic depth (with official parameters), and weight decay. For Adan, we set batch size 2048 for Swin/ViT/ConvNext and 4096 for MAE. We follow MAE and tune $\beta_3$ as 0.1.

### A.2  IMPLEMENTATION DETAILS OF ADAN

For the large-batch training experiment, we use the sqrt rule to scale the learning rate: $\mathrm{lr} = \sqrt{\frac{\text{batch size}}{256}} \times 6.25\text{e-}3$, and respectively set warmup epochs $\{20, 40, 60, 100, 160, 200\}$ for batch size bs $= \{1k, 2k, 4k, 8k, 16k, 32k\}$. For other remaining experiments, we use the hyper-parameters: learning rate 1.5e-2 for ViT/Swin/ResNet/ConvNext and MAE fine-tuning, and 2.0e-3 for MAE pre-training according to the official settings. We set $\beta_1 = 0.02, \beta_2 = 0.08$ and $\beta_3 = 0.01$, and let weight decay be 0.02 unless noted otherwise. We clip the global gradient norm to 5 for ResNet training and do not clip the gradient for ViT, Swin, ConvNext, and MAE. In the implementation, to keep consistent with Adam-type optimizers, we utilize the de-bias strategy for Adan.

### A.3  DETAILED STEPS FOR BERT

We replace the default Adam optimizer in BERT with our Adan for both pretraining and fune-tuning. Specifically, we first pretrain BERT-base on the Bookcorpus and Wikipedia datasets, and then finetune BERT-base separately for each GLUE task on the corresponding training data. Note, GLUE is a collection of 9 tasks/datasets to evaluate natural language understanding systems, in which the tasks are organized as either single-sentence classification or sentence-pair classification.

Here we simply replace the Adam optimizer in BERT with our Adan and do not make other changes, *e.g.* random seed, warmup steps and learning rate decay strategy, dropout probability, *etc.* For

Table 9: Top-1 accuracy (%) of ResNet18 under the official setting in (He et al., 2016). ∗ are reported in (Zhuang et al., 2020).

| **Adan** | SGD | Nadam | AdaBound | Adam | Radam | Padam | LAMB | AdamW | AdaBlief | Adai |
|---|---|---|---|---|---|---|---|---|---|---|
| **70.60** | 70.23* | 68.82 | 68.13* | 63.79* | 67.62* | 70.07 | 68.46 | 67.93* | 70.08* | 69.68 |

Table 10: Top-1 accuracy (%) of different optimizers when training ViT-S on ImageNet trained under training setting II. * is reported in (Touvron et al., 2021).

| Epoch | 100 | 150 | 200 | 300 |
|---|---|---|---|---|
| AdamW (default) | 76.1 | 78.9 | 79.2 | 79.9* |
| Adam | 62.0 | 64.0 | 64.5 | 66.7 |
| Adai | 66.4 | 72.6 | 75.3 | 77.4 |
| SGD-M (AGD) | 64.3 | 68.7 | 71.4 | 73.9 |
| LAMB | 69.4 | 73.8 | 75.9 | 77.7 |
| **Adan (ours)** | **77.5** | **79.6** | **80.0** | **80.9** |

pretraining, we use Adan with its default weight decay (0.02) and $\beta$s ($\beta_1 = 0.02, \beta_2 = 0.08$, and $\beta_3 = 0.01$), and choose learning rate as 0.001. For fine-tuning, we consider a limited hyper-parameter sweep for each task, with a batch size of 16, and learning rates $\in \{2e - 5, 4e - 5\}$ and use Adan with $\beta_1 = 0.02, \beta_2 = 0.01$, and $\beta_3 = 0.01$ and weight decay 0.01. Following the conventional setting, we run each fine-tuning experiment three times and report the median performance in Table 6.

Same as the official setting, on MNLI, we report the mismatched and matched accuracy. And we report Matthew's Correlation and Person Correlation on the task of CoLA and STS-B, respectively. The performance on the other tasks is measured by classification accuracy.

The performance of our reproduced one (second row) is slightly better than the vanilla results of BERT reported in Huggingface-transformer (Wolf et al., 2020) (widely used codebase for transformers in NLP), since the vanilla Bookcorpus data in (Wolf et al., 2020) is not available and thus we train on the latest Bookcorpus data version.

# B  ADDITIONAL EXPERIMENTAL RESULTS

## B.1  RESUTLS ON RESNET-18

Since some well-known deep optimizers also test ResNet-18 for 90 epochs under the official vanilla training setting in (He et al., 2016), we also run Adan 90 epochs under this setting for more comparison. Table 9 shows that Adan consistently outperforms SGD and all compared adaptive optimizers. Note for this setting, it is not easy for adaptive optimizers to surpass SGD due to the absence of heavy-tailed noise, which is the crucial factor helping adaptive optimizers beat AGD (Zhang et al., 2020).

## B.2  DETAILED COMPARISON AND CONVERGENCE CURVE

Besides AdamW, we also compare Adan with several other popular optimizers, including Adam, SGD-M, and LAMB, on ViT-S. Table 10 shows that SGD, Adam, and LAMB perform poorly on ViT-S, which is also observed in the works (Xiao et al., 2021; Nado et al., 2021). These results demonstrate that the decoupled weight decay in Adan and AdamW is much more effective than 1) the vanilla weight decay, namely the commonly used $\ell_2$ regularization in SGD, and 2) the one without any weight decay, since as shown in Eqn. (6), the decoupled weight decay is a dynamic regularization along the training trajectory and could better regularize the loss. Compared with AdamW, the advantages of Adan mainly come from its faster convergence shown in Figure 2 (b). We will discuss this below.

In Figure 2 (a), we plot the curve of training and test loss along with the training epochs on ResNet50. One can observe that Adan converges faster than the compared baselines and enjoys the smallest training and test losses. This demonstrates its fast convergence property and good generalization ability. To sufficiently investigate the fast convergence of Adan, we further plot the curve of training and test loss on the ViT-Small in Figure 2 (b). From the results, we can see that Adan consistently

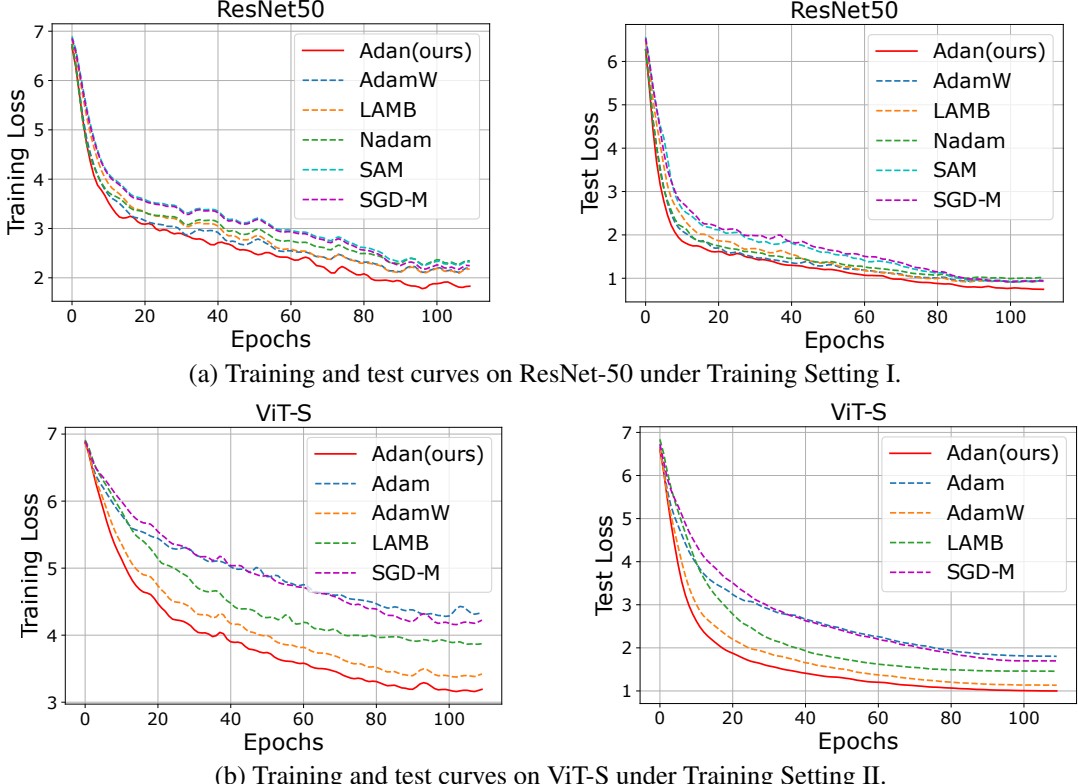

Figure 2: Training and test curves of various optimizers on ImageNet dataset. Training loss is larger due to its stronger data argumentation.

shows faster convergence behaviors than other baselines in terms of both training loss and test loss. This also partly explains the good performance of Adan over other optimizers.

**Discussion about convergence complexity** Under the corresponding assumptions, most compared optimizers already achieve the optimal complexity in terms of the dependence on optimization $\epsilon$, and their complexities only differ from their constant factors, e.g. $c_2$, $c_\infty$ and $d$. For instance, with Lipschitz gradient but without Lipschitz Hessian, most optimizers have complexity $\mathcal{O}\left(\frac{x}{\epsilon^4}\right)$ which matches the lower bound $\mathcal{O}\left(\frac{1}{\epsilon^4}\right)$ in Arjevani et al. (2019), where the constant factor $x$ varies from different optimizers, $e.g. x = c_\infty^2 d$ in Adam-type optimizer, $x = c_2^6$ in Adabelief, $x = c_2^2 d$ in LAMB, and $x = c_\infty^{2.5}$ in Adan. So under the same conditions, one cannot improve the complexity dependence on $\epsilon$ but can improve the constant factors, which are significant, especially for the network.

Actually, we empirically find $c_\infty = \mathcal{O}(8.2), c_2 = \mathcal{O}(430), d = 2.2 \times 10^7$ in the ViT-small across different optimizers, e.g., AdamW, Adam, Adan, LAMB. In the extreme case, under the widely used Lipschitz gradient assumption, the complexity bound of Adan is $7.6 \times 10^6$ smaller than the one of Adam, $3.3 \times 10^{13}$ smaller than the one of AdaBlief, $2.1 \times 10^{10}$ smaller than the one of LAMB, *etc*. For ResNet50, we also observe $c_\infty = \mathcal{O}(78), c_2 = \mathcal{O}(970), d = 2.5 \times 10^7$ which also means a large big improvement of Adan over other optimizers.

### B.3 RESULTS ON REINFORCEMENT LEARNING TASKS

Here we evaluate Adan on reinforcement learning tasks. Specifically, we replace the default Adam optimizer in PPO (Duan et al., 2016) which is one of the most popular policy gradient method, and do not many any other change in PPO. For brevity, we call this new PPO version "PPO-Adan". Then we test PPO and PPO-Adan on several games which are actually continuous control environments simulated by the standard and widely-used engine, MuJoCo (Todorov et al., 2012). For these test games, their agents receive a reward at each step. Following standard evaluation, we run each game under 10 different and independent random seeds (i.e. $1 \sim 10$), and test the performance

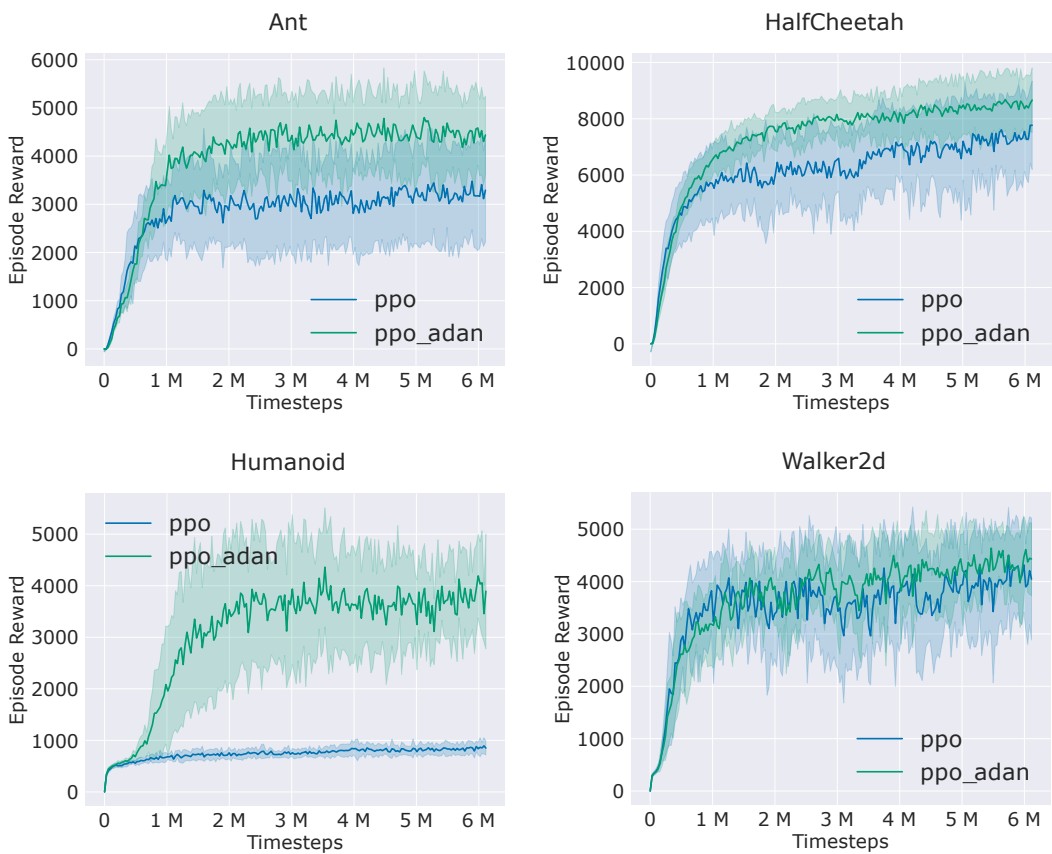

Figure 3: Comparison of PPO and our PPO-Adan on several RL games simulated by MuJoCo. Here PPO-Adan simply replaces the Adam optimizer in PPO with our Adan and does not change others.

Table 11: Test perplexity (the lower, the better) on Penn Treebank for one-, two- and three-layered LSTMs. All results except Adan and Padam in the table are reported by AdaBelief.

| LSTM | **Adan** | AdaBelief | SGD | AdaBound | Adam | AdamW | Padam | RAdam | Yogi |
|---|---|---|---|---|---|---|---|---|---|
| 1 layer | **83.6** | 84.2 | 85.0 | 84.3 | 85.9 | 84.7 | 84.2 | 86.5 | 86.5 |
| 2 layers | **65.2** | 66.3 | 67.4 | 67.5 | 67.3 | 72.8 | 67.2 | 72.3 | 71.3 |
| 3 layers | **59.8** | 61.2 | 63.7 | 63.6 | 64.3 | 69.9 | 63.2 | 70.0 | 67.5 |

for 10 episodes every 30,000 steps. All these experiments are based on the widely used codebase Tianshou[3] (Weng et al., 2021). For fairness, we use the default hyper-parameters in Tianshou, *e.g.* batch size, discount, and GAE parameter. We use Adan with its default $\beta$s ($\beta_1 = 0.02$, $\beta_2 = 0.08$, and $\beta_3 = 0.01$). Following the default setting, we do not adopt the weight decay and choose the learning rate as 3e-4.

We report the results on four test games in Figure 3, in which the solid line denotes the averaged episodes rewards in the evaluation and the shaded region is its 75% confidence intervals. From Figure 3, one can observe that on the four test games, PPO-Adan achieves much higher rewards than vanilla PPO which uses Adam as its optimizer. These results demonstrate the advantages of Adan over Adam, since PPO-Adan simply replaces the Adam optimizer in PPO with our Adan and does not make other changes.

## B.4 RESULTS ON LSTM

To begin with, we test our Adan on LSTM (Schmidhuber et al., 1997) by using the Penn TreeBank dataset (Marcinkiewicz, 1994), and report the perplexity (the lower, the better) on the test set in

---

[3] https://github.com/thu-ml/tianshou

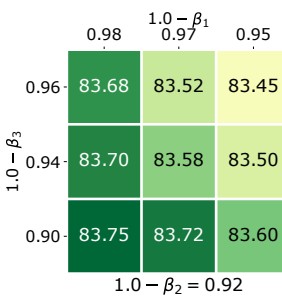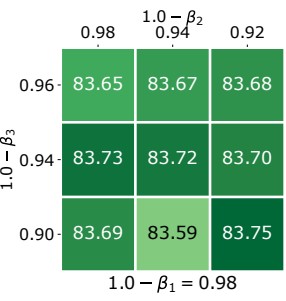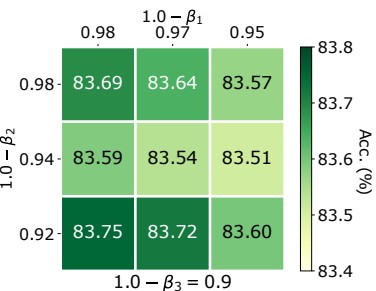

Figure 4: Effects of momentum coefficients $(\beta_1, \beta_2, \beta_3)$ to top-1 accuracy (%) of Adan on ViT-B under MAE training framework (800 pretraining and 100 fine-tuning epochs on ImageNet).

Table 12: Top-1 accuracy (%) of ViT-S on ImageNet trained under Training Setting I and II. $*$ is reported in (Touvron et al., 2021).

| Training | Training Setting I | | Training Setting II | |
|---|---|---|---|---|
| epochs | AdamW | Adan | AdamW | Adan |
| 150 | 76.4 | **80.2** | 78.3 | **79.6** |
| 300 | 77.9 | **81.1** | 79.9* | **80.7** |

Table 11. We follow the exact experimental setting in Adablief (Zhuang et al., 2020). Indeed, all our implementations are also based on the code provided by Adablief (Zhuang et al., 2020)[4]. We use the default setting for all the hyper-parameters provided by Adablief, since it provides more baselines for a fair comparison. For Adan, we utilize its default weight decay (0.02) and $\beta$s ($\beta_1 = 0.02, \beta_2 = 0.08$, and $\beta_3 = 0.01$). We choose the learning rate as 0.01 for Adan.

Table 11 shows that on the three LSTM models, Adan always achieves the lowest perplexity, making about 1.0 overall average perplexity improvement over the runner-up. Moreover, when the LSTM depth increases, the advantage of Adan becomes more remarkable.

### B.5    ABLATION STUDY

#### B.5.1    ROBUSTNESS TO IN MOMENTUM COEFFICIENTS

Here we choose MAE to investigate the effects of the momentum coefficients ($\beta$s) to Adan, since as shown in MAE, its pre-training is actually sensitive to momentum coefficients of AdamW. To this end, following MAE, we pretrain and fine tune ViT-B on ImageNet for 800 pretraining and 100 fine-tuning epochs. We also fix one of $(\beta_1, \beta_2, \beta_3)$ and tune others. Figure 4 shows that by only pretraining 800 epochs, Adan achieves 83.7%+ in most cases and outperforms the official accuracy 83.6% obtained by AdamW with 1600 pretraining epochs, indicating the robustness of Adan to $\beta$s. We also observe 1) Adan is not sensitive to $\beta_2$; 2) $\beta_1$ has a certain impact on Adan, namely the smaller the $(1.0 - \beta_1)$, the worse the accuracy; 3) similar to findings of MAE, a small second-order coefficient $(1.0 - \beta_3)$ can improve the accuracy. The smaller the $(1.0 - \beta_3)$, the more current landscape information the optimizer would utilize to adjust the coordinate-wise learning rate. Maybe the complex pre-training task of MAE is more preferred to the local geometric information.

#### B.5.2    ROBUSTNESS TO TRAINING SETTINGS

In convention, many works (Liu et al., 2021; 2022b; Touvron et al., 2022; Wightman et al., 2021; Touvron et al., 2021) often preferably chose LAMB/Adam/SGD for Training Setting I and AdamW for Training Setting II. Table 12 investigates Adan under both settings and shows consistent improvement of Adan. Moreover, one can also observe that Adan under Setting I largely improves the accuracy of Adan under Setting II. It actually surpasses the best-known accuracy 80.4% on ViT-small in (Touvron et al., 2022) trained by advanced layer scale strategy and stronger data augmentation.

---

[4]https://github.com/juntang-zhuang/Adabelief-Optimizer.    The reported results in (Zhuang et al., 2020) slightly differ from the those in (Chen et al., 2021a) because of their different settings for LSTM and training hyper-parameters.

## C NOTATION

We provide some notation that are frequently used throughout the paper. The scale $c$ is in normal font. And the vector is in bold lowercase. Give two vectors $\mathbf{x}$ and $\mathbf{y}$, $\mathbf{x} \geq \mathbf{y}$ means that $(\mathbf{x} - \mathbf{y})$ is a non-negative vector. $\mathbf{x}/\mathbf{y}$ or $\frac{\mathbf{x}}{\mathbf{y}}$ represents the element-wise vector division. $\mathbf{x} \circ \mathbf{y}$ means the element-wise multiplication, and $(\mathbf{x})^2 = \mathbf{x} \circ \mathbf{x}$. $\langle \cdot, \cdot \rangle$ is the inner product. Given a non-negative vector $\mathbf{n} \geq 0$, we let $\|\mathbf{x}\|_{\sqrt{\mathbf{n}}}^2 := \langle \mathbf{x}, \sqrt{\mathbf{n} + \varepsilon} \circ \mathbf{x} \rangle$. Unless otherwise specified, $\|\mathbf{x}\|$ is the vector $\ell_2$ norm. Note that $\mathbb{E}(\mathbf{x})$ is the expectation of random random vector $\mathbf{x}$.

## D PROOF OF LEMMA 1: EQUIVALENCE BETWEEN THE AGD AND AGD II

In this section, we show how to get AGD II from AGD. For convenience, we omit the noise term $\zeta_k$. Note that, let $\alpha := 1 - \beta_1$:

$$\text{AGD:} \begin{cases} \mathbf{g}_k = \nabla f(\boldsymbol{\theta}_k - \eta\alpha\mathbf{m}_{k-1}) \\ \mathbf{m}_k = \alpha\mathbf{m}_{k-1} + \mathbf{g}_k \\ \boldsymbol{\theta}_{k+1} = \boldsymbol{\theta}_k - \eta\mathbf{m}_k \end{cases}.$$

We can get:

$$\begin{aligned} \boldsymbol{\theta}_{k+1} - \eta\alpha\mathbf{m}_k &= \boldsymbol{\theta}_k - \eta\mathbf{m}_k - \eta\alpha\mathbf{m}_k \\ &= \boldsymbol{\theta}_k - \eta(1+\alpha)(\alpha\mathbf{m}_{k-1} + \nabla f(\boldsymbol{\theta}_k - \eta\alpha\mathbf{m}_{k-1})) \\ &= \boldsymbol{\theta}_k - \eta\alpha\mathbf{m}_{k-1} - \eta\alpha^2\mathbf{m}_{k-1} - \eta(1+\alpha)(\nabla f(\boldsymbol{\theta}_k - \eta\alpha\mathbf{m}_{k-1})). \end{aligned} \tag{8}$$

Let

$$\begin{cases} \bar{\boldsymbol{\theta}}_{k+1} := \boldsymbol{\theta}_{k+1} - \eta\alpha\mathbf{m}_k, \\ \bar{\mathbf{m}}_k := \alpha^2\mathbf{m}_{k-1} + (1+\alpha)\nabla f(\boldsymbol{\theta}_k - \eta\alpha\mathbf{m}_{k-1}) = \alpha^2\mathbf{m}_{k-1} + (1+\alpha)\nabla f(\bar{\boldsymbol{\theta}}_k) \end{cases}$$

Then, by Eq.(8), we have:

$$\bar{\boldsymbol{\theta}}_{k+1} = \bar{\boldsymbol{\theta}}_k - \eta\bar{\mathbf{m}}_k. \tag{9}$$

On the other hand, we have $\bar{\mathbf{m}}_{k-1} = \alpha^2\mathbf{m}_{k-2} + (1+\alpha)\nabla f(\bar{\boldsymbol{\theta}}_{k-1})$ and :

$$\begin{aligned} \bar{\mathbf{m}}_k - \alpha\bar{\mathbf{m}}_{k-1} &= \alpha^2\mathbf{m}_{k-1} + (1+\alpha)\nabla f(\bar{\boldsymbol{\theta}}_k) - \alpha\bar{\mathbf{m}}_{k-1} \\ &= (1+\alpha)\nabla f(\bar{\boldsymbol{\theta}}_k) + \alpha^2(\alpha\mathbf{m}_{k-2} + \nabla f(\bar{\boldsymbol{\theta}}_{k-1})) - \alpha\bar{\mathbf{m}}_{k-1} \\ &= (1+\alpha)\nabla f(\bar{\boldsymbol{\theta}}_k) + \alpha(\alpha^2\mathbf{m}_{k-2} + \alpha\nabla f(\bar{\boldsymbol{\theta}}_{k-1}) - \bar{\mathbf{m}}_{k-1}) \\ &= (1+\alpha)\nabla f(\bar{\boldsymbol{\theta}}_k) + \alpha(\alpha^2\mathbf{m}_{k-2} + \alpha\nabla f(\bar{\boldsymbol{\theta}}_{k-1})) - \alpha\bar{\mathbf{m}}_{k-1} \\ &= (1+\alpha)\nabla f(\bar{\boldsymbol{\theta}}_k) - \alpha\nabla f(\bar{\boldsymbol{\theta}}_{k-1}) \\ &= \nabla f(\bar{\boldsymbol{\theta}}_k) + \alpha(\nabla f(\bar{\boldsymbol{\theta}}_k) - \nabla f(\bar{\boldsymbol{\theta}}_{k-1})). \end{aligned} \tag{10}$$

Finally, due to Eq.(9) and Eq.10, we have:

$$\begin{cases} \bar{\mathbf{m}}_k = \alpha\bar{\mathbf{m}}_{k-1} + \left( \nabla f(\bar{\boldsymbol{\theta}}_k) + \alpha(\nabla f(\bar{\boldsymbol{\theta}}_k) - \nabla f(\bar{\boldsymbol{\theta}}_{k-1})) \right) \\ \bar{\boldsymbol{\theta}}_{k+1} = \bar{\boldsymbol{\theta}}_k - \eta\bar{\mathbf{m}}_k \end{cases}$$

## E CONVERGENCE ANALYSIS WITH LIPSCHITZ GRADIENT

Before starting the proof, we first provide several notations. Let $F_k(\boldsymbol{\theta}) := E_{\zeta}[f(\boldsymbol{\theta}, \zeta)] + \frac{\lambda_k}{2}\|\boldsymbol{\theta}\|_{\sqrt{\mathbf{n}_k}}^2$ and $\mu := \sqrt{2}\beta_3 c_\infty/\varepsilon$,

$$\|\mathbf{x}\|_{\sqrt{\mathbf{n}_k}}^2 := \langle \mathbf{x}, (\sqrt{\mathbf{n}_k} + \varepsilon) \circ \mathbf{x} \rangle, \quad \lambda_k = \lambda(1-\mu)^k.$$

Moreover, we let

$$\tilde{\boldsymbol{\theta}}_k := (\sqrt{\mathbf{n}_k} + \varepsilon) \circ \boldsymbol{\theta}_k.$$

**Lemma 2.** *Assume $f(\cdot)$ is L-smooth. For*

$$\boldsymbol{\theta}_{k+1} = \operatorname*{argmin}_{\boldsymbol{\theta}} \left( \frac{\lambda_k}{2} \|\boldsymbol{\theta}\|^2_{\sqrt{\mathbf{n}_k}} + f(\boldsymbol{\theta}_k) + \langle \mathbf{u}_k, \boldsymbol{\theta} - \boldsymbol{\theta}_k \rangle + \frac{1}{2\eta} \|(\boldsymbol{\theta} - \boldsymbol{\theta}_k)\|^2_{\sqrt{\mathbf{n}_k}} \right).$$

*With $\eta \leq \min\{\frac{\varepsilon}{3L}, \frac{1}{10\lambda}\}$, then we have:*

$$F_{k+1}(\boldsymbol{\theta}_{k+1}) \leq F_k(\boldsymbol{\theta}_k) - \frac{\eta}{4c_\infty} \left\| \mathbf{u}_k + \lambda_k \tilde{\boldsymbol{\theta}}_k \right\|^2 + \frac{\eta}{2\varepsilon} \|\mathbf{g}_k - \mathbf{u}_k\|^2,$$

*where $\mathbf{g}_k := \nabla f(\boldsymbol{\theta}_k)$.*

*Proof.* We denote $\mathbf{p}_k := \mathbf{u}_k / (\sqrt{\mathbf{n}_k} + \varepsilon)$. By the optimality condition of $\boldsymbol{\theta}_{k+1}$, we have

$$\lambda_k \boldsymbol{\theta}_k + \mathbf{p}_k = \frac{\lambda_k \tilde{\boldsymbol{\theta}}_k + \mathbf{u}_k}{\sqrt{\mathbf{n}_k} + \varepsilon} = \frac{1 + \eta \lambda_k}{\eta} (\boldsymbol{\theta}_k - \boldsymbol{\theta}_{k+1}). \tag{11}$$

Then for $\eta \leq \frac{\varepsilon}{3L}$, we have:

$$F_{k+1}(\boldsymbol{\theta}_{k+1}) \leq f(\boldsymbol{\theta}_k) + \langle \nabla f(\boldsymbol{\theta}_k), \boldsymbol{\theta}_{k+1} - \boldsymbol{\theta}_k \rangle + \frac{L}{2} \|\boldsymbol{\theta}_{k+1} - \boldsymbol{\theta}_k\|^2 + \frac{\lambda_{k+1}}{2} \|\boldsymbol{\theta}_{k+1}\|^2_{\sqrt{\mathbf{n}_{k+1}}}$$

$$\overset{(a)}{\leq} f(\boldsymbol{\theta}_k) + \langle \nabla f(\boldsymbol{\theta}_k), \boldsymbol{\theta}_{k+1} - \boldsymbol{\theta}_k \rangle + \frac{L}{2} \|\boldsymbol{\theta}_{k+1} - \boldsymbol{\theta}_k\|^2 + \frac{\lambda_k}{2} \|\boldsymbol{\theta}_{k+1}\|^2_{\sqrt{\mathbf{n}_k}}$$

$$\overset{(b)}{\leq} F_k(\boldsymbol{\theta}_k) + \left\langle \boldsymbol{\theta}_{k+1} - \boldsymbol{\theta}_k, \lambda_k \boldsymbol{\theta}_k + \frac{\mathbf{g}_k}{\sqrt{\mathbf{n}_k} + \varepsilon} \right\rangle_{\sqrt{\mathbf{n}_k}} + \frac{L/\varepsilon + \lambda_k}{2} \|\boldsymbol{\theta}_{k+1} - \boldsymbol{\theta}_k\|^2_{\sqrt{\mathbf{n}_k}}$$

$$= F_k(\boldsymbol{\theta}_k) + \frac{L/\varepsilon + \lambda_k}{2} \|\boldsymbol{\theta}_{k+1} - \boldsymbol{\theta}_k\|^2_{\sqrt{\mathbf{n}_k}} + \left\langle \boldsymbol{\theta}_{k+1} - \boldsymbol{\theta}_k, \lambda_k \boldsymbol{\theta}_k + \mathbf{p}_k + \frac{\mathbf{g}_k - \mathbf{u}_k}{\sqrt{\mathbf{n}_k} + \varepsilon} \right\rangle_{\sqrt{\mathbf{n}_k}}$$

$$\overset{(c)}{=} F_k(\boldsymbol{\theta}_k) + \left( \frac{L/\varepsilon + \lambda_k}{2} - \frac{1 + \eta \lambda_k}{\eta} \right) \|\boldsymbol{\theta}_{k+1} - \boldsymbol{\theta}_k\|^2_{\sqrt{\mathbf{n}_k}} + \left\langle \boldsymbol{\theta}_{k+1} - \boldsymbol{\theta}_k, \frac{\mathbf{g}_k - \mathbf{u}_k}{\sqrt{\mathbf{n}_k} + \varepsilon} \right\rangle_{\sqrt{\mathbf{n}_k}}$$

$$\overset{(d)}{\leq} F_k(\boldsymbol{\theta}_k) + \left( \frac{L/\varepsilon}{2} - \frac{1}{\eta} \right) \|\boldsymbol{\theta}_{k+1} - \boldsymbol{\theta}_k\|^2_{\sqrt{\mathbf{n}_k}} + \frac{1}{2\eta} \|\boldsymbol{\theta}_{k+1} - \boldsymbol{\theta}_k\|^2_{\sqrt{\mathbf{n}_k}} + \frac{\eta}{2\varepsilon} \|\mathbf{g}_k - \mathbf{u}_k\|^2$$

$$\leq F_k(\boldsymbol{\theta}_k) - \frac{1}{3\eta} \|\boldsymbol{\theta}_{k+1} - \boldsymbol{\theta}_k\|^2_{\sqrt{\mathbf{n}_k}} + \frac{\eta}{2\varepsilon} \|\mathbf{g}_k - \mathbf{u}_k\|^2$$

$$\leq F_k(\boldsymbol{\theta}_k) - \frac{\eta}{4c_\infty} \left\| \mathbf{u}_k + \lambda_k \tilde{\boldsymbol{\theta}}_k \right\|^2 + \frac{\eta}{2\varepsilon} \|\mathbf{g}_k - \mathbf{u}_k\|^2,$$

where (a) comes from the fact $\lambda_{k+1}(1 - \mu)^{-1} = \lambda_k$ and Proposition 3:

$$\left( \frac{\sqrt{\mathbf{n}_k} + \varepsilon}{\sqrt{\mathbf{n}_{k+1}} + \varepsilon} \right)_i \geq 1 - \mu,$$

which implies:

$$\lambda_{k+1} \|\boldsymbol{\theta}_{k+1}\|^2_{\sqrt{\mathbf{n}_{k+1}}} \leq \frac{\lambda_{k+1}}{1 - \mu} \|\boldsymbol{\theta}_{k+1}\|^2_{\sqrt{\mathbf{n}_k}} = \lambda_k \|\boldsymbol{\theta}_{k+1}\|^2_{\sqrt{\mathbf{n}_k}},$$

and (b) is from:

$$\|\boldsymbol{\theta}_{k+1}\|^2_{\sqrt{\mathbf{n}_k}} = \left( \|\boldsymbol{\theta}_k\|^2_{\sqrt{\mathbf{n}_k}} + 2 \langle \boldsymbol{\theta}_{k+1} - \boldsymbol{\theta}_k, \boldsymbol{\theta}_k \rangle_{\sqrt{\mathbf{n}_k}} + \|\boldsymbol{\theta}_{k+1} - \boldsymbol{\theta}_k\|^2_{\sqrt{\mathbf{n}_k}} \right),$$

(c) is due to Eqn. (11), and for (d), we utilize:

$$\left\langle \boldsymbol{\theta}_{k+1} - \boldsymbol{\theta}_k, \frac{\mathbf{g}_k - \mathbf{u}_k}{\sqrt{\mathbf{n}_k} + \varepsilon} \right\rangle_{\sqrt{\mathbf{n}_k}} \leq \frac{1}{2\eta} \|\boldsymbol{\theta}_{k+1} - \boldsymbol{\theta}_k\|^2_{\sqrt{\mathbf{n}_k}} + \frac{\eta}{2\varepsilon} \|\mathbf{g}_k - \mathbf{u}_k\|^2,$$

the last inequality comes from the fact in Eqn. (11) and $\eta \leq \frac{1}{10\lambda}$, such that:

$$\frac{1}{3\eta} \|(\boldsymbol{\theta}_{k+1} - \boldsymbol{\theta}_k)\|^2_{\sqrt{\mathbf{n}_k}} = \frac{\eta}{3(\sqrt{\mathbf{n}_k} + \varepsilon)(1 + \eta \lambda_k)} \left\| \mathbf{u}_k + \lambda_k \tilde{\boldsymbol{\theta}}_k \right\|^2 \geq \frac{\eta}{4c_\infty} \left\| \mathbf{u}_k + \lambda_k \tilde{\boldsymbol{\theta}}_k \right\|^2.$$

$\square$

**Theorem 1.** *Suppose Assumptions 1 and 2 hold. Let $c_l := \frac{1}{c_\infty}$ and $c_u := \frac{1}{\varepsilon}$. With $\beta_3 c_\infty / \varepsilon \ll 1$,*

$$\eta^2 \leq \frac{c_l \beta_1^2}{8 c_u^3 L^2}, \quad \max\{\beta_1, \beta_2\} \leq \frac{c_l \epsilon^2}{96 c_u \sigma^2}, \quad T \geq \max\left\{\frac{24\Delta_0}{\eta c_l \epsilon^2}, \frac{24 c_u \sigma^2}{\beta_1 c_l \epsilon^2}\right\},$$

*where $\Delta_0 := F(\boldsymbol{\theta}_0) - f^*$ and $f^* := \min_{\boldsymbol{\theta}} \mathbb{E}_{\boldsymbol{\zeta}}[\nabla f(\boldsymbol{\theta}_k, \boldsymbol{\zeta})]$, then we let $\mathbf{u}_k := \mathbf{m}_k + (1 - \beta_1)\mathbf{v}_k$ and have:*

$$\frac{1}{T+1} \sum_{k=0}^{T} \mathbb{E}\left(\left\|\mathbf{u}_k + \lambda_k \tilde{\boldsymbol{\theta}}_k\right\|^2\right) \leq \epsilon^2,$$

*and*

$$\frac{1}{T+1} \sum_{k=0}^{T} \mathbb{E}\left(\left\|\mathbf{m}_k - \mathbf{g}_k^{full}\right\|^2\right) \leq \frac{\epsilon^2}{4}, \quad \frac{1}{T+1} \sum_{k=0}^{T} \mathbb{E}\left(\|\mathbf{v}_k\|^2\right) \leq \frac{\epsilon^2}{4}.$$

*Hence, we have:*

$$\frac{1}{T+1} \sum_{k=0}^{T} \mathbb{E}\left(\left\|\nabla_{\boldsymbol{\theta}_k}\left(\frac{\lambda_k}{2}\|\boldsymbol{\theta}\|^2_{\sqrt{\mathbf{n}_k}} + \mathbb{E}_{\boldsymbol{\zeta}}[\nabla f(\boldsymbol{\theta}_k, \boldsymbol{\zeta})]\right)\right\|^2\right) \leq 4\epsilon^2.$$

*Proof.* For convince, we let $\mathbf{u}_k := \mathbf{m}_k + (1 - \beta_1)\mathbf{v}_k$ and $\mathbf{g}_k^{full} := \mathbb{E}_{\boldsymbol{\zeta}}[\nabla f(\boldsymbol{\theta}_k, \boldsymbol{\zeta})]$. We have:

$$\left\|\mathbf{u}_k - \mathbf{g}_k^{full}\right\|^2 \leq 2\left\|\mathbf{m}_k - \mathbf{g}_k^{full}\right\|^2 + 2(1 - \beta_1)^2 \|\mathbf{v}_k\|^2.$$

By Lemma 2, Lemma 5, and Lemma 6, we already have:

$$F_{k+1}(\boldsymbol{\theta}_{k+1}) \leq F_k(\boldsymbol{\theta}_k) - \frac{\eta c_l}{4}\left\|\mathbf{u}_k + \lambda_k \tilde{\boldsymbol{\theta}}_k\right\|^2 + \eta c_u \left\|\mathbf{g}_k^{full} - \mathbf{m}_k\right\|^2 + \eta c_u (1 - \beta_1)^2 \|\mathbf{v}_k\|^2, \quad (12)$$

$$\mathbb{E}\left(\left\|\mathbf{m}_{k+1} - \mathbf{g}_{k+1}^{full}\right\|^2\right) \leq (1 - \beta_1)\mathbb{E}\left(\left\|\mathbf{m}_k - \mathbf{g}_k^{full}\right\|^2\right) + \frac{(1 - \beta_1)^2 L^2}{\beta_1}\mathbb{E}\left(\|\boldsymbol{\theta}_{k+1} - \boldsymbol{\theta}_k\|^2\right) + \beta_1^2 \sigma^2 \tag{13}$$

$$\mathbb{E}\left(\|\mathbf{v}_{k+1}\|^2\right) \leq (1 - \beta_2)\mathbb{E}\left(\|\mathbf{v}_k\|^2\right) + 2\beta_2 \mathbb{E}\left(\left\|\mathbf{g}_{k+1}^{full} - \mathbf{g}_k^{full}\right\|^2\right) + 3\beta_2^2 \sigma^2 \tag{14}$$

Then by adding Eq.(12) with $\frac{\eta c_u}{\beta_1} \times$ Eq.(13) and $\frac{\eta c_u (1 - \beta_1)^2}{\beta_2} \times$ Eq.(14), we can get:

$$\mathbb{E}(\Phi_{k+1}) \leq \mathbb{E}\left(\Phi_k - \frac{\eta c_l}{4}\left\|\mathbf{u}_k + \lambda_k \tilde{\boldsymbol{\theta}}_k\right\|^2 + \frac{\eta c_u}{\beta_1}\left(\frac{(1 - \beta_1)^2 L^2}{\beta_1}\|\boldsymbol{\theta}_{k+1} - \boldsymbol{\theta}_k\|^2 + \beta_1^2 \sigma^2\right)\right)$$

$$+ \frac{\eta c_u (1 - \beta_1)^2}{\beta_2}\mathbb{E}\left(2\beta_2 L^2 \|\boldsymbol{\theta}_{k+1} - \boldsymbol{\theta}_k\|^2 + 3\beta_2^2 \sigma^2\right)$$

$$\leq \mathbb{E}\left(\Phi_k - \frac{\eta c_l}{4}\left\|\mathbf{u}_k + \lambda_k \tilde{\boldsymbol{\theta}}_k\right\|^2 + \eta c_u L^2 \left(\frac{(1 - \beta_1)^2}{\beta_1^2} + 2(1 - \beta_1)^2\right)\|\boldsymbol{\theta}_{k+1} - \boldsymbol{\theta}_k\|^2\right)$$

$$+ (\beta_1 + 3\beta_2)\eta c_u \sigma^2$$

$$\overset{(a)}{\leq} \mathbb{E}\left(\Phi_k - \frac{\eta c_l}{4}\left\|\mathbf{u}_k + \lambda_k \tilde{\boldsymbol{\theta}}_k\right\|^2 + \frac{\eta c_u L^2}{\beta_1^2}\|\boldsymbol{\theta}_{k+1} - \boldsymbol{\theta}_k\|^2\right) + 4\beta_m \eta c_u \sigma^2$$

$$\overset{(b)}{\leq} \mathbb{E}\left(\Phi_k + \left(\frac{(\eta c_u)^3 L^2}{\beta_1^2} - \frac{\eta c_l}{4}\right)\left\|\mathbf{u}_k + \lambda_k \tilde{\boldsymbol{\theta}}_k\right\|^2\right) + 4\beta_m \eta c_u \sigma^2$$

$$\leq \mathbb{E}\left(\Phi_k - \frac{\eta c_l}{8}\left\|\mathbf{u}_k + \lambda_k \tilde{\boldsymbol{\theta}}_k\right\|^2\right) + 4\beta_m \eta c_u \sigma^2,$$

where we let:

$$\Phi_k := F_k(\boldsymbol{\theta}_k) - f^* + \frac{\eta c_u}{\beta_1}\left\|\mathbf{m}_k - \mathbf{g}_k^{full}\right\|^2 + \frac{\eta c_u (1 - \beta_1)^2}{\beta_2}\|\mathbf{v}_k\|^2,$$

$$\beta_m = \max\{\beta_1, \beta_2\} \leq \frac{2}{3}, \quad \eta \leq \frac{c_l \beta_1^2}{8 c_u^3 L^2},$$

and for (a), when $\beta_1 \leq \frac{2}{3}$, we have:

$$\frac{(1-\beta_1)^2}{\beta_1^2} + 2(1-\beta_1)^2 < \frac{1}{\beta_1^2},$$

and (b) is due to Eq.(11) from Lemma 2. And hence, we have:

$$\sum_{k=0}^{T} \mathbb{E}(\Phi_{k+1}) \leq \sum_{k=0}^{T} \mathbb{E}(\Phi_k) - \frac{\eta c_l}{8} \sum_{k=0}^{T} \left\| \mathbf{u}_k + \lambda_k \tilde{\boldsymbol{\theta}}_k \right\|^2 + (T+1)4\eta c_u \beta_m \sigma^2.$$

Hence, we can get:

$$\frac{1}{T+1} \sum_{k=0}^{T} \mathbb{E}\left( \left\| \mathbf{u}_k + \lambda_k \tilde{\boldsymbol{\theta}}_k \right\|^2 \right) \leq \frac{8\Phi_0}{\eta c_l T} + \frac{32 c_u \beta \sigma^2}{c_l} = \frac{8\Delta_0}{\eta c_l T} + \frac{8 c_u \sigma^2}{\beta_1 c_l T} + \frac{32 c_u \beta_m \sigma^2}{c_l} \leq \epsilon^2,$$

where

$$\Delta_0 := F(\boldsymbol{\theta}_0) - f^*, \quad \beta_m \leq \frac{c_l \epsilon^2}{96 c_u \sigma^2}, \quad T \geq \max\left\{ \frac{24\Delta_0}{\eta c_l \epsilon^2}, \frac{24 c_u \sigma^2}{\beta_1 c_l \epsilon^2} \right\}.$$

We finish the first part of the theorem. From Eq.(13), we can conclude that:

$$\frac{1}{T+1} \sum_{k=0}^{T} \mathbb{E}\left( \left\| \mathbf{m}_k - \mathbf{g}_k^{full} \right\|^2 \right) \leq \frac{\sigma^2}{\beta T} + \frac{L^2 \eta^2 c_u^2 \epsilon^2}{\beta_1^2} + \beta_1 \sigma^2 < \frac{\epsilon^2}{4}.$$

From Eq.(14), we can conclude that:

$$\frac{1}{T+1} \sum_{k=0}^{T} \mathbb{E}\left( \|\mathbf{v}_k\|^2 \right) \leq 2L^2 \eta^2 c_u^2 \epsilon^2 + 3\beta_2 \sigma^2 < \frac{\epsilon^2}{4}.$$

Finally we have:

$$\frac{1}{T+1} \sum_{k=0}^{T} \mathbb{E}\left( \left\| \nabla_{\boldsymbol{\theta}_k} \left( \frac{\lambda_k}{2} \|\boldsymbol{\theta}\|_{\sqrt{\mathbf{n}_k}}^2 + \mathbb{E}_{\boldsymbol{\zeta}}[f(\boldsymbol{\theta}_k, \boldsymbol{\zeta})] \right) \right\|^2 \right)$$

$$\leq \frac{1}{T+1} \left( \sum_{k=0}^{T} \mathbb{E}\left( 2\left\| \mathbf{u}_k + \lambda_k \tilde{\boldsymbol{\theta}}_k \right\|^2 + 4\left\| \mathbf{m}_k - \mathbf{g}_k^{full} \right\|^2 + 4\|\mathbf{v}_k\|^2 \right) \right) \leq 4\epsilon^2.$$

Now, we have finished the proof. □

# F  FASTER CONVERGENCE WITH LIPSCHITZ HESSIAN

For convince, we let $\lambda = 0$, $\beta_1 = \beta_2 = \beta$ and $\beta_3 = \beta^2$ in the following proof. To consider the weight decay term in the proof, we refer to the previous section for more details. For the ease of notation, we denote $\mathbf{x}$ instead of $\boldsymbol{\theta}$ the variable needed to be optimized in the proof, and abbreviate $E_{\boldsymbol{\zeta}}[f(\boldsymbol{\theta}_k, \boldsymbol{\zeta})]$ as $f(\boldsymbol{\theta}_k)$.

## F.1  REFORMULATION

---
**Algorithm 2:** Nesterov Adaptive Momentum Estimation Reformulation

**Input:** initial point $\boldsymbol{\theta}_0$, stepsize $\eta$, average coefficients $\beta$, and $\varepsilon$.

1 **begin**
2    **while** $k < K$ **do**
3       get stochastic gradient estimator $\mathbf{g}_k$ at $\mathbf{x}_k$;
4       $\hat{\mathbf{m}}_k = (1 - \beta)\hat{\mathbf{m}}_{k-1} + \beta(\mathbf{g}_k + (1 - \beta)(\mathbf{g}_k - \mathbf{g}_{k-1}))$;
5       $\mathbf{n}_k = (1 - \beta^2)\mathbf{n}_{k-1} + \beta^2(\mathbf{g}_{k-1} + (1 - \beta)(\mathbf{g}_{k-1} - \mathbf{g}_{k-2}))^2$;
6       $\boldsymbol{\eta}_k = \eta/(\sqrt{\mathbf{n}_k} + \varepsilon)$;
7       $\mathbf{y}_{k+1} = \mathbf{x}_k - \boldsymbol{\eta}_k \beta \mathbf{g}_k$;
8       $\mathbf{x}_{k+1} = \mathbf{y}_{k+1} + (1 - \beta)[(\mathbf{y}_{k+1} - \mathbf{y}_k) + (\boldsymbol{\eta}_{k-1} - \boldsymbol{\eta}_k)(\hat{\mathbf{m}}_{k-1} - \beta \mathbf{g}_{k-1})]$;
9       **if** $(k + 1)\sum_{t=0}^{k} \left\| (\sqrt{\mathbf{n}_t} + \varepsilon)^{1/2} \circ (\mathbf{y}_{t+1} - \mathbf{y}_t) \right\|^2 \geq R^2$ **then**
10          get stochastic gradient estimator $\mathbf{g}_0$ at $\mathbf{x}_{k+1}$;
11          $\hat{\mathbf{m}}_0 = \mathbf{g}_0, \quad \mathbf{n}_0 = \mathbf{g}_0^2, \quad \mathbf{x}_0 = \mathbf{y}_0 = \mathbf{x}_{k+1}, \quad \mathbf{x}_1 = \mathbf{y}_1 = \mathbf{x}_0 - \eta \frac{\hat{\mathbf{m}}_0}{\sqrt{\mathbf{n}_0} + \varepsilon}, \quad k = 1$;
12       **end if**
13    **end while**
14    $K_0 = \operatorname{argmin}_{\lfloor \frac{K}{2} \rfloor \leq k \leq K-1} \left\| (\sqrt{\mathbf{n}_k} + \varepsilon)^{1/2} \circ (\mathbf{y}_{k+1} - \mathbf{y}_k) \right\|$;
15 **end**

**Output:** $\bar{\mathbf{x}} := \frac{1}{K_0} \sum_{k=1}^{K_0} \mathbf{x}_k$

---

We first prove the equivalent form between Algorithm 1 and Algorithm 2. The main iteration in Algorithm 1 is:

$$\begin{cases} \mathbf{m}_k = (1 - \beta)\mathbf{m}_{k-1} + \beta \mathbf{g}_k, \\ \mathbf{v}_k = (1 - \beta)\mathbf{v}_{k-1} + \beta((\mathbf{g}_k - \mathbf{g}_{k-1})), \\ \mathbf{x}_{k+1} = \mathbf{x}_k - \boldsymbol{\eta}_k \circ (\mathbf{m}_k + (1 - \beta)\mathbf{v}_k). \end{cases}$$

Let $\hat{\mathbf{m}}_k := (\mathbf{m}_k + (1 - \beta)\mathbf{v}_k)$, we can simplify the variable:

$$\begin{cases} \hat{\mathbf{m}}_k = (1 - \beta)\hat{\mathbf{m}}_{k-1} + \beta(\mathbf{g}_k + (1 - \beta)(\mathbf{g}_k - \mathbf{g}_{k-1})), \\ \mathbf{x}_{k+1} = \mathbf{x}_k - \boldsymbol{\eta}_k \circ \hat{\mathbf{m}}_k. \end{cases}$$

We let $\mathbf{y}_{k+1} := \mathbf{x}_{k+1} + \boldsymbol{\eta}_k(\hat{\mathbf{m}}_k - \beta \mathbf{g}_k)$, then we can get:

$$\mathbf{y}_{k+1} = \mathbf{x}_{k+1} + \boldsymbol{\eta}_k \hat{\mathbf{m}}_k - \beta \boldsymbol{\eta}_k \mathbf{g}_k = \mathbf{x}_{k+1} + \mathbf{x}_k - \mathbf{x}_{k+1} - \beta \boldsymbol{\eta}_k \mathbf{g}_k = \mathbf{x}_k - \beta \boldsymbol{\eta}_k \mathbf{g}_k.$$

On one hand, we have:

$$\mathbf{x}_{k+1} = \mathbf{x}_k - \boldsymbol{\eta}_k \hat{\mathbf{m}}_k = \mathbf{y}_{k+1} - \boldsymbol{\eta}_k(\hat{\mathbf{m}}_k - \beta \mathbf{g}_k).$$

On the other hand:

$$\boldsymbol{\eta}_k(\hat{\mathbf{m}}_k - \beta\mathbf{g}_k) = (1-\beta)\boldsymbol{\eta}_k(\hat{\mathbf{m}}_{k-1} + \beta(\mathbf{g}_k - \mathbf{g}_{k-1}))$$
$$=(1-\beta)\boldsymbol{\eta}_k(\hat{\mathbf{m}}_{k-1} + \beta(\mathbf{g}_k - \mathbf{g}_{k-1}))$$
$$=(1-\beta)\boldsymbol{\eta}_k\left(\frac{\mathbf{x}_{k-1} - \mathbf{x}_k}{\boldsymbol{\eta}_{k-1}} + \beta(\mathbf{g}_k - \mathbf{g}_{k-1})\right)$$
$$=(1-\beta)\frac{\boldsymbol{\eta}_k}{\boldsymbol{\eta}_{k-1}}(\mathbf{x}_{k-1} - \mathbf{x}_k + \beta\boldsymbol{\eta}_{k-1}(\mathbf{g}_k - \mathbf{g}_{k-1}))$$
$$=(1-\beta)\frac{\boldsymbol{\eta}_k}{\boldsymbol{\eta}_{k-1}}(\mathbf{y}_k - \mathbf{x}_k + \beta\boldsymbol{\eta}_{k-1}\mathbf{g}_k)$$
$$=(1-\beta)\left[\frac{\boldsymbol{\eta}_k}{\boldsymbol{\eta}_{k-1}}(\mathbf{y}_k - \mathbf{y}_{k+1} - \beta(\boldsymbol{\eta}_k - \boldsymbol{\eta}_{k-1})\mathbf{g}_k)\right]$$
$$=(1-\beta)\left[(\mathbf{y}_k - \mathbf{y}_{k+1}) + \frac{\boldsymbol{\eta}_k - \boldsymbol{\eta}_{k-1}}{\boldsymbol{\eta}_{k-1}}(\mathbf{y}_k - \mathbf{y}_{k+1} - \beta\boldsymbol{\eta}_k\mathbf{g}_k)\right]$$
$$=(1-\beta)\left[(\mathbf{y}_k - \mathbf{y}_{k+1}) + \frac{\boldsymbol{\eta}_k - \boldsymbol{\eta}_{k-1}}{\boldsymbol{\eta}_{k-1}}(\mathbf{y}_k - \mathbf{x}_k)\right]$$
$$=(1-\beta)[(\mathbf{y}_k - \mathbf{y}_{k+1}) + (\boldsymbol{\eta}_k - \boldsymbol{\eta}_{k-1})(\mathbf{m}_{k-1} - \beta\mathbf{g}_{k-1})].$$

Hence, we can conclude that:

$$\mathbf{x}_{k+1} = \mathbf{y}_{k+1} + (1-\beta)[(\mathbf{y}_{k+1} - \mathbf{y}_k) + (\boldsymbol{\eta}_{k-1} - \boldsymbol{\eta}_k)(\hat{\mathbf{m}}_{k-1} - \beta\mathbf{g}_{k-1})].$$

The main iteration in Algorithm 1 becomes:

$$\begin{cases} \mathbf{y}_{k+1} = \mathbf{x}_k - \beta\boldsymbol{\eta}_k\mathbf{g}_k, \\ \mathbf{x}_{k+1} = \mathbf{y}_{k+1} + (1-\beta)\left[(\mathbf{y}_{k+1} - \mathbf{y}_k) + \frac{\boldsymbol{\eta}_{k-1} - \boldsymbol{\eta}_k}{\boldsymbol{\eta}_{k-1}}(\mathbf{y}_k - \mathbf{x}_k)\right]. \end{cases} \tag{15}$$

### F.2 AUXILIARY BOUNDS

We first show some interesting property. Define $\mathcal{K}$ to be the iteration number when the 'if condition' triggers, that is,

$$\mathcal{K} := \min_k\left\{k \,\middle|\, k\sum_{t=0}^{k-1}\left\|(\sqrt{\mathbf{n}_t} + \varepsilon)^{1/2} \circ (\mathbf{y}_{t+1} - \mathbf{y}_t)\right\|^2 > R^2\right\}.$$

**Proposition 1.** *Given $k \le \mathcal{K}$ and $\beta \le \varepsilon/(\sqrt{2}c_\infty + \varepsilon)$, we have:*

$$\left\|(\sqrt{\mathbf{n}_k} + \varepsilon)^{1/2} \circ (\mathbf{x}_k - \mathbf{y}_k)\right\| \le R.$$

*Proof.* First of all, we let $\hat{\mathbf{n}}_k := (\sqrt{\mathbf{n}_k} + \varepsilon)^{1/2}$. Due to Proposition 3, we have:

$$\left(\frac{\sqrt{\mathbf{n}_{k-1}} + \varepsilon}{\sqrt{\mathbf{n}_k} + \varepsilon}\right)_i \in \left[1 - \frac{\sqrt{2}\beta^2 c_\infty}{\varepsilon}, 1 + \frac{\sqrt{2}\beta^2 c_\infty}{\varepsilon}\right],$$

then, we get:

$$\hat{\mathbf{n}}_k \le \left(1 - \frac{\sqrt{2}\beta^2 c_\infty}{\varepsilon}\right)^{-1/2}\hat{\mathbf{n}}_{k-1} \le (1-\beta)^{-1/4}\hat{\mathbf{n}}_{k-1},$$

where we use the fact $\beta \le \varepsilon/(2\sqrt{2}c_\infty + \varepsilon)$. For any $1 \le k \le \mathcal{K}$, we have:

$$\|\hat{\mathbf{n}}_k \circ (\mathbf{y}_k - \mathbf{y}_{k-1})\|^2 \le (1-\beta)^{-1/2}\|\hat{\mathbf{n}}_{k-1} \circ (\mathbf{y}_k - \mathbf{y}_{k-1})\|^2$$
$$\le (1-\beta)^{-1}\sum_{t=1}^{k-1}\|\hat{\mathbf{n}}_t \circ (\mathbf{y}_{t+1} - \mathbf{y}_t)\|^2 \le \frac{R^2}{k(1-\beta)},$$

hence, we can conclude that:

$$\|\hat{\mathbf{n}}_k \circ (\mathbf{y}_k - \mathbf{y}_{k-1})\|^2 \leq \frac{R^2}{k(1-\beta)}. \tag{16}$$

On the other hand, by Eq.(15), we have:

$$\mathbf{x}_{k+1} - \mathbf{y}_{k+1} = (1-\beta)\left[(\mathbf{y}_{k+1} - \mathbf{y}_k) + \frac{\boldsymbol{\eta}_k - \boldsymbol{\eta}_{k-1}}{\boldsymbol{\eta}_{k-1}}(\mathbf{x}_k - \mathbf{y}_k)\right],$$

and hence,

$$\|\hat{\mathbf{n}}_k \circ (\mathbf{x}_k - \mathbf{y}_k)\| \leq (1-\beta)\left[\|\hat{\mathbf{n}}_k \circ (\mathbf{y}_k - \mathbf{y}_{k-1})\| + \left\|\frac{\boldsymbol{\eta}_{k-1} - \boldsymbol{\eta}_{k-2}}{\boldsymbol{\eta}_{k-2}}\right\|_\infty \|\hat{\mathbf{n}}_k \circ (\mathbf{x}_{k-1} - \mathbf{y}_{k-1})\|\right]$$

$$\overset{(a)}{\leq} \sqrt{1-\beta}\frac{R}{\sqrt{k}} + (1-\beta)\frac{\sqrt{2}\beta^2 c_\infty}{\varepsilon}\left(1 - \frac{\sqrt{2}\beta^2 c_\infty}{\varepsilon}\right)^{-1/2}\|\hat{\mathbf{n}}_{k-1} \circ (\mathbf{x}_{k-1} - \mathbf{y}_{k-1})\|$$

$$\leq \sqrt{1-\beta}\frac{R}{\sqrt{k}} + \beta(1-\beta)^{3/4}\|\hat{\mathbf{n}}_{k-1} \circ (\mathbf{x}_{k-1} - \mathbf{y}_{k-1})\|$$

$$\leq \sqrt{1-\beta}R\left(\frac{1}{\sqrt{k}} + \frac{\beta(1-\beta)^{3/4}}{\sqrt{k-1}} + \cdots + \left(\beta(1-\beta)^{3/4}\right)^{k-1}\right)$$

$$\overset{(b)}{\leq} \sqrt{1-\beta}R\left(\sum_{t=1}^{k-1}\frac{1}{t^2}\right)^{1/4}\left(\sum_{t=0}^{k}\left(\beta(1-\beta)^{3/4}\right)^{4t/3}\right)^{3/4} \overset{(c)}{<} R,$$

where (a) comes from Eq.(16) and the proposition 3, (b) is the application of Hölder's inequality and (c) comes from the facts when $\beta \leq 1/2$:

$$\sum_{t=1}^{\infty}\frac{1}{t^2} = \frac{\pi^2}{6}, \quad \sqrt{1-\beta}\left(\sum_{t=0}^{k}\left(\beta(1-\beta)^{3/4}\right)^{4t/3}\right)^{3/4} \leq \left(\frac{(1-\beta)^{2/3}}{1-\beta^{4/3}(1-\beta)}\right)^{3/4}.$$

$\square$

### F.3 DECREASE OF ONE RESTART CYCLE

**Lemma 3.** *Suppose that Assumptions 1-2 hold. Let $R = \mathcal{O}(\epsilon^{0.5})$, $\beta = \mathcal{O}(\epsilon^2)$, $\eta = \mathcal{O}(\epsilon^{1.5})$, $\mathcal{K} \leq K = \mathcal{O}(\epsilon^{-2})$. Then we have:*

$$\mathbb{E}\left(f(\mathbf{y}_\mathcal{K}) - f(\mathbf{x}_0)\right) = -\mathcal{O}(\epsilon^{1.5}). \tag{17}$$

*Proof.* Recall Eq.(15) and denote $\mathbf{g}_k^{full} := \nabla f(\boldsymbol{\theta}_k)$ for convenience:

$$\begin{cases} \mathbf{y}_{k+1} = \mathbf{x}_k - \beta\boldsymbol{\eta}_k \circ \left(\mathbf{g}_k^{full} + \boldsymbol{\xi}_k\right) \\ \mathbf{x}_{k+1} - \mathbf{y}_{k+1} = (1-\beta)\left[(\mathbf{y}_{k+1} - \mathbf{y}_k) + \left(\frac{\boldsymbol{\eta}_k - \boldsymbol{\eta}_{k-1}}{\boldsymbol{\eta}_{k-1}} \circ (\mathbf{x}_k - \mathbf{y}_k)\right)\right], \end{cases} \tag{18}$$

In this proof, we let $\hat{\mathbf{n}}_k := \left(\sqrt{\mathbf{n}_k} + \varepsilon\right)^{1/2}$, and hence $\boldsymbol{\eta}_k = \eta/\hat{\mathbf{n}}_k^2$. By the $L$-smoothness condition, for $1 \leq k \leq \mathcal{K}$, we have:

$$\mathbb{E}\left(f(\mathbf{y}_{k+1}) - f(\mathbf{x}_k)\right) \leq \mathbb{E}\left(\langle \mathbf{g}_k, \mathbf{y}_{k+1} - \mathbf{x}_k\rangle + \frac{L}{2}\|\mathbf{y}_{k+1} - \mathbf{x}_k\|^2\right)$$

$$= \mathbb{E}\left(-\left\langle\frac{\mathbf{y}_{k+1} - \mathbf{x}_k}{\beta\boldsymbol{\eta}_k} + \boldsymbol{\xi}_k, \mathbf{y}_{k+1} - \mathbf{x}_k\right\rangle + \frac{L}{2}\|\mathbf{y}_{k+1} - \mathbf{x}_k\|^2\right)$$

$$\overset{(a)}{\leq} \mathbb{E}\left(-\frac{1}{\eta\beta}\|\hat{\mathbf{n}}_k \circ (\mathbf{y}_{k+1} - \mathbf{x}_k)\|^2 + \frac{L}{2}\|\mathbf{y}_{k+1} - \mathbf{x}_k\|^2\right) + \frac{\eta\beta\sigma^2}{\varepsilon} \tag{19}$$

$$\leq \mathbb{E}\left(-\frac{1}{\eta\beta}\|\hat{\mathbf{n}}_k \circ (\mathbf{y}_{k+1} - \mathbf{x}_k)\|^2 + \frac{L}{2\varepsilon}\|\hat{\mathbf{n}}_k \circ (\mathbf{y}_{k+1} - \mathbf{x}_k)\|^2\right) + \frac{\eta\beta\sigma^2}{\varepsilon}$$

$$\leq \mathbb{E}\left(-\frac{1}{2\eta\beta}\|\hat{\mathbf{n}}_k \circ (\mathbf{y}_{k+1} - \mathbf{x}_k)\|^2\right) + \frac{\eta\beta\sigma^2}{\varepsilon},$$

where (a) comes from the facts:

$$\mathbb{E}\left(\langle\boldsymbol{\xi}_k, \mathbf{y}_{k+1} - \mathbf{x}_k\rangle\right) = \mathbb{E}\left(\langle\boldsymbol{\xi}_k, \mathbf{x}_k - \beta\boldsymbol{\eta}_k \circ (\mathbf{g}_k + \boldsymbol{\xi}_k)\rangle\right) = \mathbb{E}\left(\langle\boldsymbol{\xi}_k, \beta\boldsymbol{\eta}_k \circ \boldsymbol{\xi}_k\rangle\right) \leq \frac{\eta\beta\sigma^2}{\varepsilon}.$$

and the last inequality is due to $L\eta \leq \varepsilon$. On the other hand, we have:

$$\mathbb{E}(f(\mathbf{x}_k) - f(\mathbf{y}_k)) \leq \mathbb{E}\left(\langle\nabla f(\mathbf{y}_k), \mathbf{x}_k - \mathbf{y}_k\rangle + \frac{L}{2}\|\mathbf{x}_k - \mathbf{y}_k\|^2\right)$$

$$= \mathbb{E}\left(\langle\mathbf{g}_k, \mathbf{x}_k - \mathbf{y}_k\rangle + \langle\nabla f(\mathbf{y}_k) - \nabla f(\mathbf{x}_k), \mathbf{x}_k - \mathbf{y}_k\rangle + \frac{L}{2}\|\mathbf{x}_k - \mathbf{y}_k\|^2\right)$$

$$\leq \mathbb{E}\left(\langle\mathbf{g}_k, \mathbf{x}_k - \mathbf{y}_k\rangle + \frac{1}{2L}\|\nabla f(\mathbf{y}_k) - \nabla f(\mathbf{x}_k)\|^2 + \frac{L}{2}\|\mathbf{x}_k - \mathbf{y}_k\|^2 + \frac{L}{2}\|\mathbf{x}_k - \mathbf{y}_k\|^2\right)$$

$$\leq \mathbb{E}\left(\langle\mathbf{g}_k, \mathbf{x}_k - \mathbf{y}_k\rangle + \frac{3L}{2}\|\mathbf{x}_k - \mathbf{y}_k\|^2\right)$$

$$= \mathbb{E}\left(-\left\langle\frac{\mathbf{y}_{k+1} - \mathbf{x}_k}{\beta\boldsymbol{\eta}_k} + \boldsymbol{\xi}_k, \mathbf{x}_k - \mathbf{y}_k\right\rangle + \frac{3L}{2}\|\mathbf{x}_k - \mathbf{y}_k\|^2\right)$$

$$= \mathbb{E}\left(\frac{1}{\eta\beta}\langle\hat{\mathbf{n}}_k^2 \circ (\mathbf{y}_{k+1} - \mathbf{x}_k), \mathbf{y}_k - \mathbf{x}_k\rangle + \frac{3L}{2}\|\mathbf{x}_k - \mathbf{y}_k\|^2\right)$$

$$\overset{(a)}{\leq} \mathbb{E}\left(\frac{1}{2\eta\beta}\left(\|\hat{\mathbf{n}}_k \circ (\mathbf{y}_{k+1} - \mathbf{x}_k)\|^2 + \|\hat{\mathbf{n}}_k \circ (\mathbf{y}_k - \mathbf{x}_k)\|^2 - \|\hat{\mathbf{n}}_k \circ (\mathbf{y}_{k+1} - \mathbf{y}_k)\|^2\right) + \frac{3L}{2}\|\mathbf{x}_k - \mathbf{y}_k\|^2\right)$$

$$\overset{(b)}{\leq} \mathbb{E}\left(\frac{1}{2\eta\beta}\left(\|\hat{\mathbf{n}}_k \circ (\mathbf{y}_{k+1} - \mathbf{x}_k)\|^2 - \|\hat{\mathbf{n}}_k \circ (\mathbf{y}_{k+1} - \mathbf{y}_k)\|^2\right) + \frac{1 + \beta/2}{2\eta\beta}\|\hat{\mathbf{n}}_k \circ (\mathbf{y}_k - \mathbf{x}_k)\|^2\right)$$

(20)

where (a) comes from the following facts, and in (b), we use $3L\eta \leq \frac{\varepsilon}{2}$:

$$2\langle\hat{\mathbf{n}}_k^2 \circ (\mathbf{y}_{k+1} - \mathbf{x}_k), \mathbf{y}_k - \mathbf{x}_k\rangle = \|\hat{\mathbf{n}}_k \circ (\mathbf{y}_{k+1} - \mathbf{x}_k)\|^2 + \|\hat{\mathbf{n}}_k \circ (\mathbf{y}_k - \mathbf{x}_k)\|^2 - \|\hat{\mathbf{n}}_k \circ (\mathbf{y}_{k+1} - \mathbf{y}_k)\|^2.$$

By combing Eq.(19) and Eq.(20), we have:

$$\mathbb{E}(f(\mathbf{y}_{k+1}) - f(\mathbf{y}_k)) \leq \mathbb{E}\left(-\frac{1}{2\eta\beta}\|\hat{\mathbf{n}}_k \circ (\mathbf{y}_{k+1} - \mathbf{y}_k)\|^2 + \frac{1 + \beta/2}{2\eta\beta}\|\hat{\mathbf{n}}_k \circ (\mathbf{y}_k - \mathbf{x}_k)\|^2\right) + \frac{\eta\beta\sigma^2}{\varepsilon}$$

$$\overset{(a)}{\leq} \mathbb{E}\left(-\frac{1}{2\eta\beta}\|\hat{\mathbf{n}}_k \circ (\mathbf{y}_{k+1} - \mathbf{y}_k)\|^2 + \frac{1 - \beta/2 - \beta^2/2}{2\eta\beta}\|\hat{\mathbf{n}}_{k-1} \circ (\mathbf{y}_k - \mathbf{y}_{k-1})\|^2\right) + \frac{4\beta^2 R^2 c_\infty^2}{\eta\varepsilon^2} + \frac{\eta\beta\sigma^2}{\varepsilon},$$

where (a) comes from the following fact, and note that by Proposition 1 we already have $\hat{\mathbf{n}}_k \leq (1 - \beta)^{-1/4}\hat{\mathbf{n}}_{k-1}$:

$$\|\hat{\mathbf{n}}_k \circ (\mathbf{x}_k - \mathbf{y}_k)\|^2$$

$$\leq (1 - \beta)^2\left[(1 + \alpha)\|\hat{\mathbf{n}}_k \circ (\mathbf{y}_k - \mathbf{y}_{k-1})\|^2 + (1 + \frac{1}{\alpha})\hat{\beta}^2\|\hat{\mathbf{n}}_k \circ (\mathbf{x}_{k-1} - \mathbf{y}_{k-1})\|^2\right]$$

$$\leq (1 - \beta)^{3/2}\left[(1 + \alpha)\|\hat{\mathbf{n}}_{k-1} \circ (\mathbf{y}_k - \mathbf{y}_{k-1})\|^2 + (1 + \frac{1}{\alpha})\hat{\beta}^2\|\hat{\mathbf{n}}_{k-1} \circ (\mathbf{x}_{k-1} - \mathbf{y}_{k-1})\|^2\right]$$

(21)

$$\leq (1 - \beta)\|\hat{\mathbf{n}}_{k-1} \circ (\mathbf{y}_k - \mathbf{y}_{k-1})\|^2 + \frac{\hat{\beta}^2(1 - \beta)^{3/2}}{1 - (1 - \beta)^{1/2}}\|\hat{\mathbf{n}}_{k-1} \circ (\mathbf{x}_{k-1} - \mathbf{y}_{k-1})\|^2$$

$$\leq (1 - \beta)\|\hat{\mathbf{n}}_{k-1} \circ (\mathbf{y}_k - \mathbf{y}_{k-1})\|^2 + \frac{2\hat{\beta}^2}{\beta}\|\hat{\mathbf{n}}_{k-1} \circ (\mathbf{x}_{k-1} - \mathbf{y}_{k-1})\|^2$$

$$\leq (1 - \beta)\|\hat{\mathbf{n}}_{k-1} \circ (\mathbf{y}_k - \mathbf{y}_{k-1})\|^2 + 4\beta^3 R^2 c_\infty^2/\varepsilon^2,$$

where we let $\hat{\beta} := \sqrt{2}\beta^2 c_\infty/\varepsilon$, $\alpha = (1 - \beta)^{-1/2} - 1$, and the last inequality we use the results in Proposition 1. Summing over $k = 2, \cdots, \mathcal{K} - 1$, and note that $\mathbf{y}_1 = \mathbf{x}_1$, and hence we have $\mathbb{E}(f(\mathbf{y}_2) - f(\mathbf{x}_1)) = \mathbb{E}(f(\mathbf{y}_2) - f(\mathbf{y}_1)) \leq \eta\beta\sigma c_\infty/\sqrt{\varepsilon}$ due to Eq. (19), then we get:

$$\mathbb{E}(f(\mathbf{y}_\mathcal{K}) - f(\mathbf{y}_1)) \leq \mathbb{E}\left(-\frac{1}{4\eta}\sum_{t=1}^{\mathcal{K}-1}\|\hat{\mathbf{n}}_k \circ (\mathbf{y}_{t+1} - \mathbf{y}_t)\|^2\right) + \frac{4\mathcal{K}\beta^2 R^2 c_\infty^2}{\eta\varepsilon^2} + \frac{\mathcal{K}\eta\beta\sigma^2}{\varepsilon}.$$

On the other hand, similar to the results given in Eq.(19), we have:

$$\mathbb{E}\left(f(\mathbf{y}_1) - f(\mathbf{y}_0)\right) = \mathbb{E}\left(f(\mathbf{x}_1) - f(\mathbf{x}_0)\right) \le \mathbb{E}\left(-\frac{1}{2\eta}\|\hat{\mathbf{n}}_k \circ (\mathbf{y}_1 - \mathbf{y}_0)\|^2\right) + \frac{\eta\sigma^2}{\varepsilon}.$$

Therefore, using $\beta\mathcal{K} = \mathcal{O}(1)$ and the restart condition

$$\mathcal{K}\sum_{t=0}^{\mathcal{K}-1}\left\|(\sqrt{\mathbf{n}_t} + \varepsilon)^{1/2} \circ (\mathbf{y}_{t+1} - \mathbf{y}_t)\right\|^2 \ge R^2,$$

we can get:

$$\mathbb{E}\left(f(\mathbf{y}_\mathcal{K}) - f(\mathbf{y}_0)\right) \le \mathbb{E}\left(-\frac{1}{4\eta}\sum_{t=0}^{\mathcal{K}-1}\|\hat{\mathbf{n}}_k \circ (\mathbf{y}_{k+1} - \mathbf{y}_k)\|^2\right) + \frac{4\mathcal{K}\beta^2 R^2 c_\infty^2}{\eta\varepsilon^2} + \frac{(\mathcal{K}\beta + 1)\eta\sigma^2}{\varepsilon}$$

$$\le -\frac{R^2}{4\mathcal{K}\eta} + \frac{4\mathcal{K}\beta^2 R^2 c_\infty^2}{\eta\varepsilon^2} + \frac{(\mathcal{K}\beta + 1)\eta\sigma^2}{\varepsilon} = -\mathcal{O}\left(\frac{R^2}{\mathcal{K}\eta} - \frac{\beta R^2}{\eta} - \eta\right) = -\mathcal{O}(\epsilon^{1.5}).$$

Now, we finish the proof of this claim. $\qquad\square$

### F.4 GRADIENT IN THE LAST RESTART CYCLE

Before showing the main results, we first provide several definitions for the convenience of proof. Note that, for any $k < \mathcal{K}$ we already have:

$$(\varepsilon)^{1/2}\|\mathbf{y}_k - \mathbf{y}_0\| \le (\varepsilon)^{1/2}\sqrt{k\sum_{t=0}^{k-1}\|\mathbf{y}_{t+1} - \mathbf{y}_t\|^2} \le R.$$

and we have:

$$\mathbb{E}\left(\|\mathbf{x}_k - \mathbf{x}_0\|\right) \le \mathbb{E}\left(\|\mathbf{y}_k - \mathbf{x}_k\| + \|\mathbf{y}_k - \mathbf{x}_0\|\right) \le \frac{2R}{\varepsilon^{1/2}}, \tag{22}$$

where we utilize the results from Proposition 1. For each epoch, denote $\mathbf{H} := \nabla^2 f(\mathbf{x}_0)$. We then define:

$$h(\mathbf{y}) := \left\langle \mathbf{g}_0^{full}, \mathbf{y} - \mathbf{x}_0\right\rangle + \frac{1}{2}(\mathbf{y} - \mathbf{x}_0)^\top \mathbf{H}(\mathbf{y} - \mathbf{x}_0).$$

Recall the Eq. (15):

$$\begin{cases} \mathbf{y}_{k+1} = \mathbf{x}_k - \beta\boldsymbol{\eta}_k \circ \left(\mathbf{g}_k^{full} + \boldsymbol{\xi}_k\right) = \mathbf{x}_k - \beta\boldsymbol{\eta}_k \circ (\nabla h(\mathbf{x}_k) + \boldsymbol{\delta}_k + \boldsymbol{\xi}_k) \\ \mathbf{x}_{k+1} - \mathbf{y}_{k+1} = (1 - \beta)\left[(\mathbf{y}_{k+1} - \mathbf{y}_k) + \left(\frac{\boldsymbol{\eta}_k - \boldsymbol{\eta}_{k-1}}{\boldsymbol{\eta}_{k-1}} \circ (\mathbf{x}_k - \mathbf{y}_k)\right)\right], \end{cases} \tag{23}$$

where we let $\boldsymbol{\delta}_k := \mathbf{g}_k^{full} - \nabla h(\mathbf{x}_k)$, and we can get that:

$$\mathbb{E}\left(\|\boldsymbol{\delta}_k\|\right) = \mathbb{E}\left(\left\|\mathbf{g}_k^{full} - \mathbf{g}_0^{full} - \mathbf{H}(\mathbf{x}_k - \mathbf{x}_0)\right\|\right)$$

$$= \mathbb{E}\left(\left\|\left(\int_0^1 \nabla^2 h(\mathbf{x}_0 + t(\mathbf{x}_k - \mathbf{x}_0)) - \mathbf{H}\right)(\mathbf{x}_k - \mathbf{x}_0)dt\right\|\right) \tag{24}$$

$$\le \frac{\rho}{2}\mathbb{E}\left(\|\mathbf{x}_k - \mathbf{x}_0\|^2\right) \le \frac{2\rho R^2}{\varepsilon}.$$

Iterations in Eq.(23) can be viewed as applying the proposed optimizer to the quadratic approximation $h(\mathbf{x})$ with the gradient error $\delta_k$, which is in the order of $\mathcal{O}\left(\rho R^2/\varepsilon\right)$.

**Lemma 4.** *Suppose that Assumptions 1-3 hold. Let $B = \mathcal{O}(\epsilon^{0.5})$, $\beta = \mathcal{O}(\epsilon^2)$, $\eta = \mathcal{O}(\epsilon^{1.5})$, $\mathcal{K} \le K = \mathcal{O}(\epsilon^{-2})$. Then we have:*

$$\mathbb{E}\left(\|\nabla f(\bar{\mathbf{x}})\|\right) = \mathcal{O}(\epsilon),$$

*where $\bar{\mathbf{x}} := \frac{1}{K_0 - 1}\sum_{k=1}^{K_0}\mathbf{x}_k$.*

*Proof.* Since $h(\cdot)$ is quadratic, then we have:

$$\mathbb{E}\left(\|\nabla h(\bar{\mathbf{x}})\|\right) = \mathbb{E}\left(\left\|\frac{1}{K_0-1}\sum_{k=1}^{K_0}\nabla h(\mathbf{x}_k)\right\|\right)$$

$$=\frac{1}{K_0-1}\mathbb{E}\left\|\sum_{k=1}^{K_0}(\beta\boldsymbol{\eta}_k)^{-1}\circ(\mathbf{y}_{k+1}-\mathbf{x}_k)+\boldsymbol{\xi}_k+\boldsymbol{\delta}_k\right\|$$

$$\leq\frac{1}{(K_0-1)\beta}\mathbb{E}\left\|\sum_{k=1}^{K_0}(\beta\boldsymbol{\eta}_k)^{-1}\circ(\mathbf{y}_{k+1}-\mathbf{x}_k)\right\|+\frac{1}{(K_0-1)}\mathbb{E}\left\|\sum_{k=1}^{K_0}\boldsymbol{\xi}_k\right\|+\frac{1}{(K_0-1)}\mathbb{E}\left\|\sum_{k=1}^{K_0}\boldsymbol{\delta}_k\right\|$$

$$\overset{(a)}{\leq}\frac{1}{(K_0-1)\beta}\mathbb{E}\left\|\sum_{k=1}^{K_0}(\boldsymbol{\eta}_k)^{-1}\circ(\mathbf{y}_{k+1}-\mathbf{x}_k)\right\|+\frac{\sigma}{\sqrt{K_0-1}}+\frac{2\rho R^2}{\varepsilon}$$

$$=\frac{1}{(K_0-1)\beta}\mathbb{E}\left\|\sum_{k=1}^{K_0}\frac{\mathbf{y}_{k+1}-\mathbf{y}_k-(1-\beta)(\mathbf{y}_k-\mathbf{y}_{k-1})}{\boldsymbol{\eta}_k}-(1-\beta)\frac{\boldsymbol{\eta}_{k-1}-\boldsymbol{\eta}_{k-2}}{\boldsymbol{\eta}_{k-2}\boldsymbol{\eta}_k}(\mathbf{x}_{k-1}-\mathbf{y}_{k-1})\right\|$$

$$+\frac{\sigma}{\sqrt{K_0-1}}+\frac{2\rho R^2}{\varepsilon}$$

$$\overset{(b)}{\leq}\frac{1}{(K_0-1)\beta}\mathbb{E}\left\|\sum_{k=1}^{K_0}\frac{\mathbf{y}_{k+1}-\mathbf{y}_k-(1-\beta)(\mathbf{y}_k-\mathbf{y}_{k-1})}{\boldsymbol{\eta}_k}\right\|+\frac{2\beta c_\infty^{1.5}R}{\eta\varepsilon}+\frac{\sigma}{\sqrt{K_0-1}}+\frac{2\rho R^2}{\varepsilon}$$

$$\overset{(c)}{\leq}\frac{1}{(K_0-1)\beta}\mathbb{E}\left\|\sum_{k=1}^{K_0}\left(\frac{\mathbf{y}_{k+1}-\mathbf{y}_k}{\boldsymbol{\eta}_k}-\frac{(1-\beta)(\mathbf{y}_k-\mathbf{y}_{k-1})}{\boldsymbol{\eta}_{k-1}}\right)\right\|+\frac{4\beta c_\infty^{1.5}R}{\eta\varepsilon}+\frac{\sigma}{\sqrt{K_0-1}}+\frac{2\rho R^2}{\varepsilon}$$

$$\leq\frac{1}{(K_0-1)\beta}\mathbb{E}\left\|\frac{\mathbf{y}_{K_0}-\mathbf{y}_{K_0-1}}{\boldsymbol{\eta}_{K_0}}\right\|+\frac{1}{(K_0-1)}\mathbb{E}\left\|\sum_{k=1}^{K_0-1}\frac{\mathbf{y}_{k+1}-\mathbf{y}_k}{\boldsymbol{\eta}_k}\right\|+\frac{4\beta c_\infty^{1.5}R}{\eta\varepsilon}+\frac{\sigma}{\sqrt{K_0-1}}+\frac{2\rho R^2}{\varepsilon}$$

$$\overset{(d)}{\leq}\frac{1}{(K_0-1)}\mathbb{E}\left\|\sum_{k=1}^{K_0}\frac{\mathbf{y}_{k+1}-\mathbf{y}_k}{\boldsymbol{\eta}_k}\right\|+\frac{4R\sqrt{c_\infty}}{\beta\eta K^2}+\frac{4\beta c_\infty^{1.5}R}{\eta\varepsilon}+\frac{\sigma}{\sqrt{K_0-1}}+\frac{2\rho R^2}{\varepsilon}$$

$$\leq\frac{\sqrt{2c_\infty}}{\eta K}\mathbb{E}\left\|\sum_{k=1}^{K_0}(\sqrt{\mathbf{n}_k}+\varepsilon)^{1/2}\circ(\mathbf{y}_{k+1}-\mathbf{y}_k)\right\|+\frac{4R\sqrt{c_\infty}}{\beta\eta K^2}+\frac{4\beta c_\infty^{1.5}B}{\eta\varepsilon}+\frac{\sigma}{\sqrt{K_0-1}}+\frac{2\rho R^2}{\varepsilon}$$

$$\leq\frac{\sqrt{2c_\infty}R}{\eta K}+\frac{4R\sqrt{c_\infty}}{\beta\eta K^2}+\frac{4\beta c_\infty^{1.5}R}{\eta\varepsilon}+\frac{\sigma}{\sqrt{K_0-1}}+\frac{2\rho R^2}{\varepsilon}$$

$$=\mathcal{O}\left(\frac{R}{\eta K}+\frac{\beta R}{\eta}+\frac{1}{\sqrt{K}}+R^2\right)=\mathcal{O}(\epsilon),$$

where (a) is due to the independence of $\boldsymbol{\xi}_k$'s and Eq.(24), (b) comes from Propositions 1 and 2:

$$\left\|\frac{\boldsymbol{\eta}_{k-1}-\boldsymbol{\eta}_{k-2}}{\boldsymbol{\eta}_{k-2}\boldsymbol{\eta}_k}(\mathbf{x}_{k-1}-\mathbf{y}_{k-1})\right\|\leq\frac{\sqrt{\mathbf{n}_k}+\varepsilon}{\eta(\sqrt{\mathbf{n}_{k-1}}+\varepsilon)^{1/2}}\left\|\frac{\boldsymbol{\eta}_{k-1}-\boldsymbol{\eta}_{k-2}}{\boldsymbol{\eta}_{k-2}}\right\|_\infty\|\hat{\mathbf{n}}_{k-1}\circ(\mathbf{x}_{k-1}-\mathbf{y}_{k-1})\|$$

$$\leq\frac{(\sqrt{\mathbf{n}_k}+\varepsilon)^{1/2}}{\eta}\frac{\sqrt{2}\beta^2 c_\infty}{\varepsilon}\left(1-\frac{\sqrt{2}\beta^2 c_\infty}{\varepsilon}\right)^{-1/2}R$$

$$\leq\frac{(c_\infty+\varepsilon)^{1/2}}{\eta}\frac{\sqrt{2}\beta^2 c_\infty}{\varepsilon}\frac{R}{(1-\beta)^{1/4}}\leq\left(\frac{1}{1-\beta}\right)^{1/4}\frac{2\beta^2 c_\infty^{1.5}R}{\eta\varepsilon},$$

we use the following bounds in (c):

$$\left\| \frac{(\mathbf{y}_k - \mathbf{y}_{k-1})}{\boldsymbol{\eta}_{k-1}} - \frac{(\mathbf{y}_k - \mathbf{y}_{k-1})}{\boldsymbol{\eta}_k} \right\| = \left\| \frac{\boldsymbol{\eta}_k - \boldsymbol{\eta}_{k-1}}{\boldsymbol{\eta}_{k-1}\boldsymbol{\eta}_k}(\mathbf{y}_k - \mathbf{y}_{k-1}) \right\|$$

$$\leq \frac{\left(\sqrt{\mathbf{n}_{k-1}} + \varepsilon\right)^{1/2}}{\eta} \left\| \frac{\boldsymbol{\eta}_k - \boldsymbol{\eta}_{k-1}}{\boldsymbol{\eta}_k} \right\|_\infty \left\| \left(\sqrt{\mathbf{n}_{k-1}} + \varepsilon\right)^{1/2} \circ (\mathbf{y}_k - \mathbf{y}_{k-1}) \right\|$$

$$\leq \frac{\left(\sqrt{\mathbf{n}_{k-1}} + \varepsilon\right)^{1/2}}{\eta} \frac{\sqrt{2}\beta^2 c_\infty}{\varepsilon} \frac{R}{k} \leq \frac{(c_\infty + \varepsilon)^{1/2}}{\eta} \frac{\sqrt{2}\beta^2 c_\infty}{\varepsilon} \frac{R}{k} \leq \frac{2\beta^2 c_\infty^{1.5} R}{\eta \varepsilon k},$$

(d) is implied by $K_0 = \operatorname{argmin}_{\lfloor \frac{K}{2} \rfloor \leq k \leq K-1} \left\| \left(\sqrt{\mathbf{n}_k} + \varepsilon\right)^{1/2} \circ (\mathbf{y}_{k+1} - \mathbf{y}_k) \right\|$ and restart condition:

$$\left\| \frac{\mathbf{y}_{K_0} - \mathbf{y}_{K_0-1}}{\boldsymbol{\eta}_{K_0}} \right\|^2 \leq \frac{\sqrt{\mathbf{n}_{K_0}} + \varepsilon}{\eta^2} \left\| \left(\sqrt{\mathbf{n}_{K_0}} + \varepsilon\right)^{1/2} \circ (\mathbf{y}_{K_0} - \mathbf{y}_{K_0-1}) \right\|^2$$

$$\left\| \left(\sqrt{\mathbf{n}_{K_0}} + \varepsilon\right)^{1/2} \circ (\mathbf{y}_{K_0} - \mathbf{y}_{K_0-1}) \right\|^2 \leq \frac{1}{K - \lfloor K/2 \rfloor} \sum_{k=\lfloor K/2 \rfloor}^{K-1} \left\| \left(\sqrt{\mathbf{n}_k} + \varepsilon\right)^{1/2} \circ (\mathbf{y}_{k+1} - \mathbf{y}_k) \right\|^2$$

$$\leq \frac{1}{K - \lfloor K/2 \rfloor} \sum_{k=1}^{K} \left\| \left(\sqrt{\mathbf{n}_k} + \varepsilon\right)^{1/2} \circ (\mathbf{y}_{k+1} - \mathbf{y}_k) \right\|^2 \leq \frac{1}{K - \lfloor K/2 \rfloor} \frac{R^2}{K} \leq \frac{2R^2}{K^2}.$$

Finally, we have:

$$\mathbb{E}\left(\|\nabla f(\bar{\mathbf{x}})\|\right) = \mathbb{E}\left(\|\nabla h(\bar{\mathbf{x}})\|\right) + \mathbb{E}\left(\|\nabla f(\bar{\mathbf{x}}) - \nabla h(\bar{\mathbf{x}})\|\right) = \mathcal{O}(\epsilon) + \frac{2\rho R^2}{\varepsilon} = \mathcal{O}(\epsilon),$$

where we use the results from Eq.(24), namely:

$$\mathbb{E}\left(\|\nabla f(\bar{\mathbf{x}}) - \nabla h(\bar{\mathbf{x}})\|\right) = \mathbb{E}\left(\left\| \nabla f(\bar{\mathbf{x}}) - \mathbf{g}_0^{full} - \mathbf{H}(\bar{\mathbf{x}} - \mathbf{x}_0) \right\|\right) \leq \frac{\rho}{2} \mathbb{E}\left(\|\bar{\mathbf{x}} - \mathbf{x}_0\|^2\right),$$

and we also note that, by Eq.(22):

$$\mathbb{E}\|\bar{\mathbf{x}} - \mathbf{x}_0\| \leq \frac{1}{K_0 - 1} \sum_{k=1}^{K_0} \mathbb{E}\|\mathbf{x}_k - \mathbf{x}_0\| \leq \frac{2R}{\varepsilon^{1/2}}.$$

$\square$

## F.5 PROOF FOR MAIN THEOREM

**Theorem 2.** *Suppose that Assumptions 1-3 hold. Let $B = \mathcal{O}(\epsilon^{0.5})$, $\beta = \mathcal{O}(\epsilon^2)$, $\eta = \mathcal{O}(\epsilon^{1.5})$, $\mathcal{K} \leq K = \mathcal{O}(\epsilon^{-2})$. Then Algorithm 1 find an $\epsilon$-approximate first-order stationary point within at most $\mathcal{O}(\epsilon^{-3.5})$ iterations. Namely, we have:*

$$\mathbb{E}\left(f(\mathbf{y}_\mathcal{K}) - f(\mathbf{x}_0)\right) = -\mathcal{O}(\epsilon^{1.5}), \quad \mathbb{E}\left(\|\nabla f(\bar{\mathbf{x}})\|\right) = \mathcal{O}(\epsilon).$$

*Proof.* Note that at the beginning of each restart cycle in Algorithm 2, we set $\mathbf{x}_0$ to be the last iterate $\mathbf{x}_\mathcal{K}$ in the previous restart cycle. Due to Lemma 3, we already have:

$$\mathbb{E}\left(f(\mathbf{y}_\mathcal{K}) - f(\mathbf{x}_0)\right) = -\mathcal{O}(\epsilon^{1.5}).$$

Summing this inequality over all cycles, say $N$ total restart cycles, we have:

$$\min_{\mathbf{x}} f(\mathbf{x}) - f(\mathbf{x}_{\text{init}}) = -\mathcal{O}(N\epsilon^{1.5}),$$

Hence, the Algorithm 2 terminates within at most $\mathcal{O}(\epsilon^{-1.5}\Delta_f)$ restart cycles, where $\Delta_f := f(\mathbf{x}_{\text{init}}) - \min_{\mathbf{x}} f(\mathbf{x})$. Note that each cycle contain at most $K = \mathcal{O}(\epsilon^{-2})$ iteration step, therefore, the total iteration number must be less than $\mathcal{O}(\epsilon^{-3.5}\Delta_f)$.

On the other hand, by Lemma 4, in the last restart cycle, we have:

$$\mathbb{E}\left(\|\nabla f(\bar{\mathbf{x}})\|\right) = \mathcal{O}(\epsilon).$$

Now, we obtain the final conclusion for the theorem. $\square$

## G    AUXILIARY LEMMAS

**Proposition 2.** *If Assumption 2 holds. We have:*
$$\|\mathbf{m}_k\|_\infty \leq c_\infty, \quad \|\mathbf{n}_k\|_\infty \leq c_\infty^2.$$

*Proof.* By the definition of $\mathbf{m}_k$, we can have that:
$$\mathbf{m}_k = \sum_{t=0}^{k} c_{k,t}\mathbf{g}_t,$$

where
$$c_{k,t} = \begin{cases} \beta_1(1-\beta_1)^{(k-t)} & \text{when } t > 0, \\ \\ (1-\beta_1)^k & \text{when } t = 0. \end{cases}$$

Similar, we also have:
$$\mathbf{n}_k = \sum_{t=0}^{k} c'_{k,t}(\mathbf{g}_t + (1-\beta_2)(\mathbf{g}_t - \mathbf{g}_{t-1}))^2,$$

where
$$c'_{k,t} = \begin{cases} \beta_3(1-\beta_3)^{(k-t)} & \text{when } t > 0, \\ \\ (1-\beta_3)^k & \text{when } t = 0. \end{cases}$$

If is obvious that:
$$\sum_{t=0}^{k} c_{k,t} = 1, \quad \sum_{t=0}^{k} c'_{k,t} = 1,$$

hence, we get:
$$\|\mathbf{m}_k\|_\infty \leq \sum_{t=0}^{k} c_{k,t}\|\mathbf{g}_t\|_\infty,$$
$$\|\mathbf{n}_k\|_\infty \leq \sum_{t=0}^{k} c'_{k,t}\|\mathbf{g}_t + (1-\beta_2)(\mathbf{g}_t - \mathbf{g}_{t-1})\|_\infty^2 \leq c_\infty^2.$$

$\square$

**Proposition 3.** *If Assumption 2 holds, we have:*
$$\left\|\frac{\boldsymbol{\eta}_k - \boldsymbol{\eta}_{k-1}}{\boldsymbol{\eta}_{k-1}}\right\|_\infty \leq \frac{\sqrt{2}\beta_3 c_\infty}{\varepsilon}.$$

*Proof.* Give any index $i \in [d]$ and the definitions of $\boldsymbol{\eta}_k$, we have:
$$\left|\left(\frac{\boldsymbol{\eta}_k - \boldsymbol{\eta}_{k-1}}{\boldsymbol{\eta}_{k-1}}\right)_i\right| = \left|\left(\frac{\sqrt{\mathbf{n}_{k-1}} + \varepsilon}{\sqrt{\mathbf{n}_k} + \varepsilon}\right)_i - 1\right| = \left|\left(\frac{\sqrt{\mathbf{n}_{k-1}} - \sqrt{\mathbf{n}_k}}{\sqrt{\mathbf{n}_k} + \varepsilon}\right)_i\right|.$$

Note that, by the definition of $\mathbf{n}_k$, we have:
$$\left|\left(\frac{\sqrt{\mathbf{n}_{k-1}} - \sqrt{\mathbf{n}_k}}{\sqrt{\mathbf{n}_k} + \varepsilon}\right)_i\right| \leq \left|\left(\frac{\sqrt{|\mathbf{n}_{k-1} - \mathbf{n}_k|}}{\sqrt{\mathbf{n}_k} + \varepsilon}\right)_i\right|$$
$$= \beta_3\left(\frac{\sqrt{\left|\mathbf{n}_{k-1} - (\mathbf{g}_k + (1-\beta_2)(\mathbf{g}_k - \mathbf{g}_{k-1}))^2\right|}}{\sqrt{\mathbf{n}_k} + \varepsilon}\right)_i \leq \frac{\sqrt{2}\beta_3 c_\infty}{\varepsilon},$$

hence, we have:
$$\left|\left(\frac{\boldsymbol{\eta}_k - \boldsymbol{\eta}_{k-1}}{\boldsymbol{\eta}_{k-1}}\right)_i\right| \in \left[0, \frac{\sqrt{2}\beta_3 c_\infty}{\varepsilon}\right].$$

We finish the proof. $\square$

**Lemma 5.** *Consider a moving average sequence:*

$$\mathbf{m}_k = (1 - \beta)\mathbf{m}_{k-1} + \beta\mathbf{g}_k,$$

*where we note that:*

$$\mathbf{g}_k = \mathbb{E}_{\boldsymbol{\zeta}}[\nabla f(\boldsymbol{\theta}_k, \boldsymbol{\zeta})] + \boldsymbol{\xi}_k,$$

*and we denote* $\mathbf{g}_k^{full} := E_{\boldsymbol{\zeta}}[\nabla f(\boldsymbol{\theta}_k, \boldsymbol{\zeta})]$ *for convenience. Then we have:*

$$\mathbb{E}\left(\left\|\mathbf{m}_k - \mathbf{g}_k^{full}\right\|^2\right) \leq (1-\beta)\mathbb{E}\left(\left\|\mathbf{m}_{k-1} - \mathbf{g}_{k-1}^{full}\right\|^2\right) + \frac{(1-\beta)^2 L^2}{\beta}\mathbb{E}\left(\|\boldsymbol{\theta}_{k-1} - \boldsymbol{\theta}_k\|^2\right) + \beta^2\sigma^2.$$

*Proof.* Note that, we have:

$$\mathbf{m}_k - \mathbf{g}_k^{full} = (1-\beta)\left(\mathbf{m}_{k-1} - \mathbf{g}_{k-1}^{full}\right) + (1-\beta)\mathbf{g}_{k-1}^{full} - \mathbf{g}_k^{full} + \beta\mathbf{g}_k$$

$$= (1-\beta)\left(\mathbf{m}_{k-1} - \mathbf{g}_{k-1}^{full}\right) + (1-\beta)\left(\mathbf{g}_{k-1}^{full} - \mathbf{g}_k^{full}\right) + \beta\left(\mathbf{g}_k - \mathbf{g}_k^{full}\right).$$

Then, take expectation on both sides:

$$\mathbb{E}\left(\left\|\mathbf{m}_k - \mathbf{g}_k^{full}\right\|^2\right)$$

$$= (1-\beta)^2\mathbb{E}\left(\left\|\mathbf{m}_{k-1} - \mathbf{g}_{k-1}^{full}\right\|^2\right) + (1-\beta)^2\mathbb{E}\left(\left\|\mathbf{g}_{k-1}^{full} - \mathbf{g}_k^{full}\right\|^2\right) + \beta^2\sigma^2 +$$

$$2(1-\beta)^2\mathbb{E}\left(\left\langle\mathbf{m}_{k-1} - \mathbf{g}_{k-1}^{full}, \mathbf{g}_{k-1}^{full} - \mathbf{g}_k^{full}\right\rangle\right)$$

$$\leq \left((1-\beta)^2 + (1-\beta)^2 a\right)\mathbb{E}\left(\left\|\mathbf{m}_{k-1} - \mathbf{g}_{k-1}^{full}\right\|^2\right) +$$

$$\left(1 + \frac{1}{a}\right)(1-\beta)^2\mathbb{E}\left(\left\|\mathbf{g}_{k-1}^{full} - \mathbf{g}_k^{full}\right\|^2\right) + \beta^2\sigma^2$$

$$\overset{(a)}{\leq} (1-\beta)\mathbb{E}\left(\left\|\mathbf{m}_{k-1} - \mathbf{g}_{k-1}^{full}\right\|^2\right) + \frac{(1-\beta)^2}{\beta}\mathbb{E}\left(\left\|\mathbf{g}_{k-1}^{full} - \mathbf{g}_k^{full}\right\|^2\right) + \beta^2\sigma^2$$

$$\leq (1-\beta)\mathbb{E}\left(\left\|\mathbf{m}_{k-1} - \mathbf{g}_{k-1}^{full}\right\|^2\right) + \frac{(1-\beta)^2 L^2}{\beta}\mathbb{E}\left(\|\boldsymbol{\theta}_{k-1} - \boldsymbol{\theta}_k\|^2\right) + \beta^2\sigma^2,$$

where for (a), we set $a = \frac{\beta}{1-\beta}$. $\qquad\square$

**Lemma 6.** *Consider a moving average sequence:*

$$\mathbf{v}_k = (1 - \beta)\mathbf{v}_{k-1} + \beta(\mathbf{g}_k - \mathbf{g}_{k-1}),$$

*where we note that:*

$$\mathbf{g}_k = \mathbb{E}_{\boldsymbol{\zeta}}[\nabla f(\boldsymbol{\theta}_k, \boldsymbol{\zeta})] + \boldsymbol{\xi}_k,$$

*and we denote* $\mathbf{g}_k^{full} := E_{\boldsymbol{\zeta}}[f(\boldsymbol{\theta}_k, \boldsymbol{\zeta})]$ *for convenience. Then we have:*

$$\mathbb{E}\left(\|\mathbf{v}_k\|^2\right) \leq (1-\beta)\mathbb{E}\left(\|\mathbf{v}_{k-1}\|^2\right) + 2\beta\mathbb{E}\left(\left\|\mathbf{g}_k^{full} - \mathbf{g}_{k-1}^{full}\right\|^2\right) + 3\beta^2\sigma^2.$$

*Proof.* Take expectation on both sides:

$$\mathbb{E}\left(\|\mathbf{v}_k\|^2\right) = (1-\beta)^2\mathbb{E}\left(\|\mathbf{v}_{k-1}\|^2\right) + \beta^2\mathbb{E}\left(\|\mathbf{g}_k - \mathbf{g}_{k-1}\|^2\right) + 2\beta(1-\beta)\mathbb{E}(\langle\mathbf{v}_{k-1}, \mathbf{g}_k - \mathbf{g}_{k-1}\rangle)$$

$$\overset{(a)}{=} (1-\beta)^2\mathbb{E}\left(\|\mathbf{v}_{k-1}\|^2\right) + \beta^2\mathbb{E}\left(\|\mathbf{g}_k - \mathbf{g}_{k-1}\|^2\right) + 2\beta(1-\beta)\mathbb{E}\left(\left\langle\mathbf{v}_{k-1}, \mathbf{g}_k^{full} - \mathbf{g}_{k-1}\right\rangle\right)$$

$$\overset{(b)}{\leq} (1-\beta)^2\mathbb{E}\left(\|\mathbf{v}_{k-1}\|^2\right) + 2\beta^2\mathbb{E}\left(\left\|\mathbf{g}_k^{full} - \mathbf{g}_{k-1}^{full}\right\|^2\right) + 2\beta(1-\beta)\mathbb{E}\left(\left\langle\mathbf{v}_{k-1}, \mathbf{g}_k^{full} - \mathbf{g}_{k-1}\right\rangle\right) + 3\beta^2\sigma^2$$

$$\overset{(c)}{\leq} (1-\beta)^2\mathbb{E}\left(\|\mathbf{v}_{k-1}\|^2\right) + 2\beta^2\mathbb{E}\left(\left\|\mathbf{g}_k^{full} - \mathbf{g}_{k-1}^{full}\right\|^2\right) + 2\beta(1-\beta)\mathbb{E}\left(\left\langle\mathbf{v}_{k-1}, \mathbf{g}_k^{full} - \mathbf{g}_{k-1}^{full}\right\rangle\right) + 3\beta^2\sigma^2$$

$$\overset{(d)}{\leq} (1-\beta)\mathbb{E}\left(\|\mathbf{v}_{k-1}\|^2\right) + 2\beta\mathbb{E}\left(\left\|\mathbf{g}_k^{full} - \mathbf{g}_{k-1}^{full}\right\|^2\right) + 3\beta^2\sigma^2,$$

where for (a), we utilize the independence between $\mathbf{g}_k$ and $\mathbf{v}_{k-1}$, while for (b):

$$\mathbb{E}\left(\|\mathbf{g}_k - \mathbf{g}_{k-1}\|^2\right) \leq \mathbb{E}\left(\left\|\mathbf{g}_k - \mathbf{g}_k^{full}\right\|^2\right) + 2\mathbb{E}\left(\left\|\mathbf{g}_{k-1}^{full} - \mathbf{g}_{k-1}\right\|^2\right) + 2\mathbb{E}\left(\left\|\mathbf{g}_k^{full} - \mathbf{g}_{k-1}^{full}\right\|^2\right),$$

for (c), we know:

$$\mathbb{E}\left(\left\langle \mathbf{v}_{k-1}, \mathbf{g}_{k-1}^{full} - \mathbf{g}_{k-1}\right\rangle\right) = \mathbb{E}\left(\left\langle (1-\beta)\mathbf{v}_{k-2} + \beta(\mathbf{g}_{k-1} - \mathbf{g}_{k-2}), \mathbf{g}_{k-1}^{full} - \mathbf{g}_{k-1}\right\rangle\right)$$

$$= \mathbb{E}\left(\left\langle (1-\beta)\mathbf{v}_{k-2} - \beta\mathbf{g}_{k-2}, \mathbf{g}_{k-1}^{full} - \mathbf{g}_{k-1}\right\rangle\right) + \beta\mathbb{E}\left(\left\langle \mathbf{g}_{k-1} - \mathbf{g}_{k-1}^{full} + \mathbf{g}_{k-1}^{full}, \mathbf{g}_{k-1}^{full} - \mathbf{g}_{k-1}\right\rangle\right)$$

$$= -\beta\mathbb{E}\left(\left\|\mathbf{g}_{k-1}^{full} - \mathbf{g}_{k-1}\right\|^2\right),$$

and thus $\mathbb{E}\left(\left\langle \mathbf{v}_{k-1}, \mathbf{g}_k^{full} - \mathbf{g}_{k-1}\right\rangle\right) = \mathbb{E}\left(\left\langle \mathbf{v}_{k-1}, \mathbf{g}_k^{full} - \mathbf{g}_{k-1}^{full}\right\rangle\right) - \beta\mathbb{E}\left(\left\|\mathbf{g}_{k-1}^{full} - \mathbf{g}_{k-1}\right\|^2\right)$. Finally, for (d), we use:

$$2\mathbb{E}\left(\left\langle \mathbf{v}_{k-1}, \mathbf{g}_k^{full} - \mathbf{g}_{k-1}^{full}\right\rangle\right) \leq \mathbb{E}\left(\|\mathbf{v}_{k-1}\|^2\right) + \mathbb{E}\left(\left\|\mathbf{g}_k^{full} - \mathbf{g}_{k-1}^{full}\right\|^2\right).$$

$$\square$$

