# OpenReview forum: "Adan: Adaptive Nesterov Momentum Algorithm for Faster Optimizing Deep Models"
_ICLR.cc/2023/Conference — Submitted to ICLR 2023_

### Official Review · Reviewer_SpFC · 2022-10-24

**Confidence:** 3
**Correctness:** 3
**Technical Novelty And Significance:** 2
**Empirical Novelty And Significance:** 3
**Recommendation:** 6

**Clarity, Quality, Novelty And Reproducibility:**

The paper is well-written, with some novelty and good reproducibility given the code references.

**Strength And Weaknesses:**

Strength:
  1. The paper is well-written and easy-to-follow
  2. This is by far the first paper that have shown theoretical benefits over existing algorithms, as far as I am aware of, which is highly appreciated.
  3. The experimental results are on both small-scale and large-scale datasets, which are quite strong and convincing.

Weaknesses:
  1. The idea of combining Nesterov momentum with Adam is, as far as I am concerned, a very straightforward idea. I am actually very surprised that this hasn't been put into practice in common deep learning tools such as PyTorch or Tensorflow. Therefore, I find the idea of the paper not novel enough.
  2. The theoretical benefit looks marginal to me. Isn't the difference between all the optimizers only a multiplicative constant-level difference? When is the convergence of the algorithm better than the others (i.e., under what assumptions of $d$, $c_2$, $c_\infty$? Can the authors elaborate more?

**Summary Of The Paper:**

This paper proposes another variant of Adam by combining the idea of Nesterov momentum with the adaptive algorithms. The authors prove that in the nonconvex optimization setting, the proposed algorithm converges faster than the other adaptive algorithms. Experimental results show that the proposed algorithm performs better than the other algorithms in multiple tasks and different settings.

**Summary Of The Review:**

Overall, I find the idea of introducing Nesterov momentum to adaptive optimization algorithm not novel enough, but I am impressed by the theoretical results and the experiments the authors have provided. The performance of the proposed algorithm does look significantly better than the existing ones. Therefore, I vote for acceptance.

---

> ### Author Response · Authors · 2022-11-16
> **Response to Reviewer SpFC (Part II)**
>
> **Q2: The theoretical benefit looks marginal to me. Isn't the difference between all the optimizers only a multiplicative constant-level difference? When is the convergence of the algorithm better than the others (i.e., under what assumptions of d, $c_2$, $c_\infty$? Can the authors elaborate more?**
>
> **A2**: The constant-level difference among the complexities of compared optimizers is not incremental because of the following two reasons.
>
> Firstly, under the corresponding assumptions,  most compared optimizers already achieve the optimal complexity in terms of the dependence on optimization accuracy  $\epsilon$, and their complexities only differ from their constant factors, e.g. $c_2$, $c_\infty$ and $d$.  For instance,  with Lipschitz gradient but without Lipschitz Hessian,  most optimizers have complexity $\mathcal{O}(\frac{x}{\epsilon^{4}})$ which matches the lower bound $\mathcal{O}(\frac{1}{\epsilon^{4}})$ in [Ref. 3-4], where the  constant factor $x$  varies from different optimizers, e.g. $x=c_{\infty}^2 d $ in Adam-type optimizer, $x=c_2^6$ in Adabelief,   $x = c_2^2d$  in LAMB, and $x=c_{\infty}^{2.5}$ in Adan.  So under the same conditions, one cannot improve the complexity dependence on $\epsilon$ but can improve the constant factors, which, as discussed below, is still significant, especially for  DNNs.
>
> Secondly, the constant-level difference may cause very different complexity whose magnitudes vary by several orders on networks.  This is because 1) the modern network is often large, e.g., 11 M parameters in the small ResNet-18, leading a very large $d$; 2) for network gradient, its $\ell_2$-norm upper bound  $c_2$ is often much larger than its $\ell_\infty$-norm upper bound $c_\infty$ as observed and proved in many works, e.g. [Ref. 5] because the stochastic algorithms can probably adaptively adjust the parameter magnitude at different layers so that these parameter magnitudes are balanced.
>
> Actually, we also empirically find $c_\infty = \mathcal{O}(8.2), c_2 = \mathcal{O}(430), d = 2.2\times 10^{7}$ in the  ViT-small across different optimizers, e.g., AdamW, Adam, Adan, LAMB.  In the extreme case, under the widely used  Lipschitz gradient assumption,  the complexity bound of Adan is $7.6\times 10^{6}$ smaller than the one of Adam, $3.3\times 10^{13}$ smaller than the one of AdaBlief, $2.1\times 10^{10}$ smaller than the one of LAMB, etc.  For ResNet50, we also observe $c_\infty = \mathcal{O}(78), c_2 = \mathcal{O}(970), d = 2.5\times 10^{7}$ which also means a large big improvement of Adan over other optimizers. **We have included this in Appendix Sec. B.2**.
>
> 	[Ref. 3] Arjevani, Yossi, Yair Carmon, John C. Duchi, Dylan J. Foster, Ayush Sekhari, and Karthik Sridharan. "Second-order information in non-convex stochastic optimization: Power and limitations.". COLT, 2020.
>
> 	[Ref. 4] Arjevani, Yossi, Yair Carmon, John C. Duchi, Dylan J. Foster, Nathan Srebro, and Blake Woodworth. "Lower bounds for non-convex stochastic optimization." Mathematical Programming 2022.
>
> 	[Ref. 5] Du, Simon S., Wei Hu, and Jason D. Lee. "Algorithmic regularization in learning deep homogeneous models: Layers are automatically balanced." Neurips, 2018.

---

> > ### Comment · Reviewer_SpFC · 2022-11-19
> > **Reply to rebuttal**
> >
> > I thank the authors for the rebuttal. After reading other reviews and rebuttals, I have decided to keep my rating unchanged.

---

> ### Author Response · Authors · 2022-11-16
> **Response to Reviewer SpFC (Part I)**
>
> Thank you for the insightful comments.  We hope your concerns can be well addressed by our clarification.
>
> **Q1: The idea of combining Nesterov momentum with Adam is, as far as I am concerned, a very straightforward idea. I am actually very surprised that this hasn't been put into practice in common deep-learning tools such as PyTorch or Tensorflow.**
>
> **A1**: Nadam (Ref. 1) is the first one that tries to combine  Nesterov momentum with Adam and achieves good performance on small datasets and small networks. But for the widely used networks (ResNets, ViTs, LSTM, Bert, etc) and datasets (ImageNet, Cifar10/100, etc), one almost does not use Nadam to train DNNs due to its extra computation overload but comparable performance with Adam.
>
> Recall the Nesterov acceleration: 1) $m_k = (1-\beta) m_{k-1} +  \nabla f(\theta_k')$,   2) $\theta_{k+1} = \theta_k - \eta m_k$, where $\theta_k' = \theta_k - \eta (1-\beta) m_{k-1}$ is an extrapolation point. To exactly implement the Nesterov acceleration technique, one needs to preserve both $\theta_k$ and $\theta_k'$ in memory during back-propagation since we need to evaluate the gradient at the extrapolation point at $\theta_k'$ and then reload model back to  $\theta_k$ for parameter updating.  This brings extra memory cost, and also makes the updating step more complex since one needs to compute the gradient at $\theta_k'$  but reload $\theta_k$ to update the parameter.
>
> To alleviate these issues, Nadam estimates the stochastic gradient at $\theta_k$,  then updates the first-order gradient moment $\hat{m} _ k = \mu_k  \hat{m} _ {k-1} + \eta_{k} \nabla f(\theta_k)$ as Adam, and finally updates parameter as $\theta _ {k+1} = \theta_k -   (\mu _ {k+1} \hat{m} _ k + \eta_k \nabla f(\theta_k) )$, where $\\{\mu_k\\}$ are pre-defined constants.  By comparing vanilla Nesterov acceleration with Nadam, one can find that Nadam aims at using $$h_k := \mu_{k+1} \hat{m} _ k + \eta_k \nabla f(\theta_k) = \mu _ {k+1}  \mu_k  \hat{m} _ {k-1} + \eta_k (1+\mu _ {k+1}) \nabla f(\theta_k),$$ to approximate $$\eta m_{k} = \eta  (1-\beta) m_{k-1} +  \eta\nabla f(\theta_k'),$$ in vanilla Nesterov acceleration. Unfortunately, this approximation may not be good enough since $h_{k} $ in Nadam does not use any gradient at the extrapolation point $\theta_k'$ while $\eta m_{k} $ indeed uses.  In this way, Nadam may not fully inherit the merits of vanilla Nesterov acceleration.
>
> In contrast, as shown in Lemma 1, our modification of  Nesterov acceleration is provably equivalent to vanilla Nesterov acceleration, which differs from Nadam.   Moreover, our Nesterov modification also avoids the issue of extra memory cost and enjoys very simple updating steps.  Note, in the submission, we also discuss the difference between Adan and Nadam on the 5-th page. As shown in Eqn. (3) on the 5-th page, Nadam is more like Adam than Nesterov momentum methods and differs from  Adan.
>
>
> PyTorch and Tensorflow also modify   Nesterov acceleration to avoid reloading model parameters during back-propagation and integrate this version into some optimizers, e.g, SGD.  However, their implementation is equivalent to the vanilla Nesterov acceleration only when the optimization objective is quadratic and may not fully inherit the merits of vanilla Nesterov acceleration (see [Ref. 2]). Moreover, in the experiments, we also compared with the Nesterov-accelerated optimizers, e.g.,  SGD-M.
>
> 	[Ref. 1] Dozat, Timothy. "Incorporating Nesterov momentum into adam." 2016.
>
> 	[Ref. 2] Sutskever, Ilya, James Martens, George Dahl, and Geoffrey Hinton. "On the importance of initialization and momentum in deep learning." ICML, 2013.

---

### Official Review · Reviewer_5SGt · 2022-10-25

**Confidence:** 3
**Correctness:** 3
**Technical Novelty And Significance:** 3
**Empirical Novelty And Significance:** 3
**Recommendation:** 5

**Clarity, Quality, Novelty And Reproducibility:**

(1) One page 4, the parameter epsilon appears inside the sqrt operation in both RMSProp and Adam. While they are implemented differently on Pytorch platform where epsilon is outside of the sqrt operation. My experience is that the placement of epsilon makes a difference in the validation performance for training DNN models.  Do you use default implementation of RMSProp and Adam on Pytorch in your experiment or you use the update expressions in your paper? The presentation for the algorithms need to be consistent in the whole paper.
(2) Source code is not available for reproducibility.
(3) There is no de-bias term in their new method Adan. Is it ignored for presentation convenience like RMSProp and Adam, or the de-bias term is removed in their implementation for better performance?
(4) I think the overhead of Nesterov acceleration technique is negligible compared to NME. There is no need for Nesterov acceleration technique to maintain both theta_k and theta_k'. NME is just a reformulation of Nesterov acceleration technique. I suggest the authors rewrite the contribution part on page 2. If the authors have different opinions, please verify their argument via experiments by implementing the Nesterov acceleration technique properly.
(5) The authors made quite a few assumptions to be able to analyze the convergence of Adan. I wonder if the assumptions are the same for other optimizers. If they are not the same, then Table-1 is an unfair comparison.  For example, why 3 is introduced in ||g_k||_{infty}\leq c_{infty}/3? Do other optimizers use the same assumption?
(6) The statement "update theta_1 by Line 7 in Algorithm 1" is not correct.


**Strength And Weaknesses:**

Strengths:
1. The new method Adan is developed by incorporating the gradient difference g_k-g_{k-1} in Adam based on a reformulation of the Nesterov acceleration technique.
2.  Both convergence analysis and experimental evaluation are conducted for Adan, showing its advantage over other optimizers.

Weaknesses:
1.  The authors claim that  Adan has lower complexity than a number of existing optimizers including AdaBelief and LAMB without an explanation. Is it because of the separation of the l2 regularizer with the objective, or because of the introduction of g_k- g_{k-1} in Adan, or because of improved mathematical deviation. This is very important. It helps the readers to understand what is the reason of the the low complexity of the new method without reading all the proofs. If it is because of the separation of the l2 regularizer with the objective or  improved mathematical deviation, it suggests that the convergence of other optimizers can also be improved.
2. It seems that the authors are aware of AdaBelief and Adam+ but didn't evaluate them in the experiments. AdaBelief is empirically found to have better performance than quite a number of opimizers including AdamW and Adam for training different types of DNN models including Transformer. Recently, a new method named Aida in the paper "On exploiting layerwise gradient statistics for effective training of deep neural networks" is found to perform better than AdaBelief.  I highly suggest the authors to also evaluate AdamBelief, Adam+ and Aida in their experiments. Furthermore, the parameter epsilon needs to be searched for each method to achieve best performance. It seems from page 7 that the authors only tune the hyper-parameters of Adan.
3. Compared to Adam, AdamW, AdaBelief, Adan needs to tune an additional parameter beta_2.

**Summary Of The Paper:**

The paper first reformulate the update expression of the Nesterov acceleration technique and then modify Adam accordingly to develop a new optimizer named Adan, which is different from NAdam. The main difference between Adan and Adam is that the gradient difference g_k-g_{k-1} is incorporated in computation of both the 1st and 2nd momentum. Both convergence analysis and experimental evaluation are conducted for Adan.

**Summary Of The Review:**

The authors develop a new method Adan by incorporating the gradient difference g_k-g_{k-1} in Adam based on a reformulation of the Nesterov acceleration technique. (1) Explanation of why Adan has lower theoretical complexity is needed for the readers to better understand the method. (2) Paper presentation needs further improvement (see my comments above). (3) I don't find anywhere that the hyper-parameters of the reference methods in the experiments are manually tuned while the hyper-parameters of Adan are tuned. Also it is suggested to evaluate AdaBelied, Adam+, Aida in the experiments.

---

> ### Author Response · Authors · 2022-11-16
> **Response to Reviewer 5SGt (Part IV)**
>
> **Q5: Compared to Adam, AdamW, AdaBelief, Adan needs to tune an additional parameter $\beta_2$.**
>
> **A5**: In vanilla Sec. B.5.1 of the appendix,   we have investigated the robustness of  Adan to the hyper-parameters $\beta=\{\beta_1, \beta_2, \beta_3\}$.  The experimental results show that Adan is very robust to $\beta_2$. So we always set $\beta_2=0.08$ for all our experiments in the submission. This means that $\beta_2$ does not increase the hyper-parameter tuning cost.
>
> **Q6: Source code is not available for reproducibility; There is no de-bias term in their new method Adan. Is it ignored for presentation convenience like RMSProp and Adam, or the de-bias term is removed in their implementation for better performance?**
>
> **A6**: We kindly remind that in the vanilla submission, we have submitted the source code in the supp file, which also includes the log and configure files for our experiments.
>
> For the de-bias term, we use it in practice but ignore it for ease of presentation, which is mentioned on the 4-th page. Please refer to the 4-th page.
>
> **Q7: I think the overhead of Nesterov acceleration technique is negligible compared to NME. There is no need for Nesterov acceleration technique to maintain both $\theta_k$ and $\theta_k'$. NME is just a reformulation of Nesterov acceleration technique. I suggest the authors rewrite the contribution part on page 2.**
>
> **A7**: Thanks. We do not claim that NME is superior to the Nesterov acceleration technique on computation.
>
> Recall the Nesterov acceleration: 1) $m_k = (1-\beta_1) m_{k-1} +  \nabla f(\theta_k')$,   2) $\theta_{k+1} = \theta_k - \eta m_k$, where $\theta_k' = \theta_k - \eta (1-\beta_1) m_{k-1}$ is an extrapolation point. To exactly implement the Nesterov acceleration technique, one needs to preserve both $\theta_k$ and $\theta_k'$ in memory during back-propagation since we need to evaluate the gradient at the extrapolation point $\theta_k'$ and then reload model back to  $\theta_k$ for parameter updating.
>
> As you suggested, we have revised our motivation to use NME rather than the vanilla Nesterov acceleration.  In the revision, we emphasize more about the above extra memory cost brought by Nesterov acceleration and thus introduce our memory-efficient NME to solve this issue. Please refer to the introduction part.
>
> Note,  the PyTorch implementation of Nesterov acceleration does not reload model parameters during back-propagation. However, their implementation is equivalent to the vanilla Nesterov acceleration only when the optimization objective is quadratic, and may not fully inherit the merits of vanilla Nesterov acceleration (see [Ref. 8]).
>
> 	[Ref. 8] Sutskever, Ilya, James Martens, George Dahl, and Geoffrey Hinton. "On the importance of initialization and momentum in deep learning." ICML, 2013.
>
>
>
>
> **Q8: The authors made quite a few assumptions to be able to analyze the convergence of Adan. I wonder if the assumptions are the same for other optimizers. If they are not the same, then Table 1 is an unfair comparison. For example, why 3 is introduced in $\|g_k\|_{\infty}\leq c_\infty/3$? Do other optimizers use the same assumption?**
>
> **A8:** Except for the gradient boundness,  all the other assumptions: 1)  Lipschitz gradient/Hessian (Assumption 1 and 3), 2) unbiased and variance-bounded stochastic gradient (Assumption 2), are necessary conditions to achieve the two complexity lower bounds on the non-convex problems (see [Ref. 9-10]). This holds not only for DNN optimizers but also for other stochastic optimization algorithms.
>
> For all the adaptive optimizers in Table 1, they all need gradient boundness, Lipschitz gradient/Hessian, and unbiased and variance-bounded stochastic gradient to prove the convergence. Indeed, some optimizers need extra assumptions. For instance, Adablief additionally assumes its second-order moment is coordinate-wisely non-increasing;  LAMB and Adam$^+$ additionally require a huge batch size.   So to achieve the two lower complexity bounds, Adan only adopts the necessary conditions and does not use more extra conditions.
>
> For 3 in gradient boundness $\|g_k\|_\infty \leq c_\infty/3$, we only introduce the denominator 3 for easy proof. We also can remove the factor $1/3$, and still prove the same complexity of Adan.
>
> 	[Ref. 9] Arjevani, Yossi, Yair Carmon, John C. Duchi, Dylan J. Foster, Ayush Sekhari, and Karthik Sridharan. "Second-order information in non-convex stochastic optimization: Power and limitations.". COLT, 2020.
>
> 	[Ref. 10] Arjevani, Yossi, Yair Carmon, John C. Duchi, Dylan J. Foster, Nathan Srebro, and Blake Woodworth. "Lower bounds for non-convex stochastic optimization." Mathematical Programming 2022.
>
>
>
>
>
>
> ---------
>
> If our response satisfies the reviewer, please consider raising your score. Thank you!

---

> > ### Comment · Reviewer_5SGt · 2022-11-18
> > **comment on Q7**
> >
> > I think that there is no need to keep theta_k in the memory. Because theta_k can be recovered from theta_k' and m_{k-1} in the backward process. Please correct  me if I am wrong.

---

> > > ### Author Response · Authors · 2022-11-18
> > > **Further response to Reviewer 5SGt on Q7**
> > >
> > > **We really thank you for your insightful comment again. We would like to clarify that your idea of  recovering $\theta_k$  from $\theta_k'$ and $m_{k-1}$, namely, only preserving $\theta_k'$ and $m_{k-1}$ in the memory at the training phase, is what we exactly do  in NME.**
> > >
> > > For convenience, we let $\alpha:=(1-\beta)$, and recall the Nesterov acceleration: 1) $m_k = \alpha m_{k-1} +  \nabla f(\theta_k')$,   2) $\theta_{k+1} = \theta_k - \eta m_k$, where $\theta_k' = \theta_k - \eta \alpha m_{k-1}$ is an extrapolation point.
> > >
> > > **Firstly**, we cancel $ \theta_k$ out by only using $\theta_k'$ and $m_{k-1}$  as you pointed out:
> > >
> > > $
> > > \theta_{k+1} = \theta_k - \eta m_k = \theta_k' + \eta \alpha m_{k-1} - \eta m_k = \theta_k' -  \eta \nabla f(\theta_k'),
> > > $
> > >
> > > then
> > >
> > > $
> > > \theta _ {k+1}' = \theta _ {k+1} - \eta \alpha m_k = \theta_k' -  \eta \nabla f(\theta_k') - \eta \alpha m_k = \theta_k' - \eta \hat{m} _ k,
> > > $
> > >
> > > where we define $\hat{m} _ k:=\nabla f(\theta_k') + \alpha m_k$. Hence, the Nesterov acceleration becomes:
> > >
> > > $
> > > \theta_{k+1}' = \theta_k' - \eta \hat{m} _ k, \quad \text{where}\quad \hat{m} _ k:=\nabla f(\theta_k') + \alpha m_k.
> > > $
> > >
> > > **Secondly**, we need to consider how to  iteratively update  $\hat{m} _ k$.
> > > To begin with, we have
> > >
> > > $
> > > \hat{m} _ k - \alpha \hat{m} _ {k-1} = \nabla f(\theta_k') + \alpha m_k - \alpha \nabla f(\theta_{k-1}') - \alpha^2 m_{k-1}
> > > = \nabla f(\theta_k') - \alpha \nabla f(\theta_{k-1}') + \alpha(m_k - \alpha m_{k-1})
> > > $
> > >
> > > Since we already have $m_k - \alpha m_{k-1} = \nabla f(\theta_k')$,  we can get:
> > >
> > > $
> > > \hat{m} _ k = \alpha \hat{m} _ {k-1} + \nabla f(\theta_k') + \alpha(\nabla f(\theta_k') - \nabla f(\theta_{k-1}')).
> > > $
> > >
> > > **Finally**, Nesterov acceleration becomes:
> > >
> > > $\hat{m} _ k = \alpha \hat{m} _ {k-1} + \nabla f(\theta_k') + \alpha(\nabla f(\theta_k') - \nabla f(\theta_{k-1}')),\qquad
> > > \theta_{k+1}' = \theta_k' - \eta \hat{m}_k, \qquad (1)$
> > >
> > > which is precisely the NME that is reformulated from the vanilla Nesterov acceleration.
> > >
> > > In summary, your idea of implementing Nesterov acceleration in a memory-efficient way by only preserving $\theta_k'$ and $m_{k-1}$ is what we have done in Lemma 1. Moreover, we utilize $\hat{m}_k$ instead of $m_k$ to  ease the  combining of Nesterov acceleration with  adaptive  methods.
> > >
> > > **If you have any other concerns, please also let us know. Many thanks!!**

---

> ### Author Response · Authors · 2022-11-16
> **Response to Reviewer 5SGt (Part III)**
>
> **Q3: The parameter epsilon needs to be searched for each method to achieve the best performance. It seems from page 7 that the authors only tune the hyper-parameters of Adan.**
>
> **A3**: We would like to clarify that the training settings (I and II) mentioned on the 7-th page are **NOT** the hyper-parameters that are tuned specifically for Adan. The settings are the standard training settings which are the most widely used for the current vision model training to achieve SOTA performance (see [Ref. 3-7]).  Under these settings, many models/optimizers have reported their official optimizers and their hyper-parameters, e.g., the well-tuned stabilizing constant $\varepsilon$, the official AdamW optimizer on Deit/Swin/ConvNext, and the official LMAB optimizer on ResNet. We exactly follow their settings for a fair comparison.
>
> For the optimizers run by us, we also try our best to tune their hyper-parameters, e.g., $\varepsilon$, learning rate and weight decay of SGD,  Adam, and  LAMB on ViTs,  learning rate and $\varepsilon$ of Padam/Nadam/Adai on ResNet-18.
>
> In most cases, the official setting and hyper-parameters provide better results since they have already tried several optimizers and tuned their hyper-parameters. For example, ConvNext chooses AdamW instead of SGD to train their CNN [Ref. 3]; the current SoTAs on ViTs and ResNets are trained by LAMB [Ref. 4-5], and AdaBlief chooses $\varepsilon = 1e-8$ instead of its default $\varepsilon = 1e-16$ on ResNet-18.
>
> It is worth mentioning that many optimizers achieve the best performance without tuning their $\varepsilon$, e.g., AdamW on Deit/Swin/ConvNext, LAMB on ResNet/ViT, and Adam on Bert. They all use the same default $\varepsilon=1e-8$, so we do this for our Adan and do not tune $\varepsilon$.
>
> We know that some optimizers may choose different  $\varepsilon$, such as AdaBlief and Adai. For a fair comparison, we reproduce the experiments AdaBlief performed and present the results with its training settings and $\varepsilon$. For Adai, we search its $\varepsilon$ from $\\{1e-3,1e-4,1e-5,1e-6,1e-8\\}$ to achieve the best performance.  Please see the comparison among AdaBlief, Adai, and Adan in the response to your second question (Q2) or Sec. B.
>
>
> 	[Red. 3] Liu, Zhuang, Hanzi Mao, Chao-Yuan Wu, Christoph Feichtenhofer, Trevor Darrell, and Saining Xie. "A convnet for the 2020s." CVPR. 2022.
>
> 	[Ref. 4] Touvron, Hugo, Matthieu Cord, and Hervé Jégou. "Deit iii: Revenge of the vit." arXiv preprint, 2022.
>
> 	[Ref. 5] Wightman, Ross, Hugo Touvron, and Hervé Jégou. "Resnet strikes back: An improved training procedure in timm." arXiv preprint, 2021.
>
> 	[Ref. 6] Liu, Ze, Yutong Lin, Yue Cao, Han Hu, Yixuan Wei, Zheng Zhang, Stephen Lin, and Baining Guo. "Swin transformer: Hierarchical vision transformer using shifted windows." ICCV, 2021.
>
> 	[Ref. 7] Liu, Ze, Han Hu, Yutong Lin, Zhuliang Yao, Zhenda Xie, Yixuan Wei, Jia Ning et al. "Swin transformer v2: Scaling up capacity and resolution." CVPR, 2022.
>
>
>
> **Q4: The parameter epsilon appears inside the sqrt operation in RMSProp and Adam. While they are implemented differently on Pytorch platform where epsilon is outside of the sqrt operation. The placement of epsilon makes a difference in the validation performance. Do you use default implementation of RMSProp and Adam on Pytorch or you use the update expressions in your paper? The presentation for the algorithms need to be consistent in the whole paper.**
>
> **A4**: Thanks.  In all our experiments,  for all compared optimizers,  we use their officially implemented code or adopt their code provided by the widely used framework (e.g., Pytorch, Timm, or Fairseq).  For Adam and RMSProp, we use their Pytorch implementation. So Adam, RMSProp, and Adan all put the $\varepsilon$   out of the sqrt root operation in their implementations.  Note, we put $\varepsilon$ in the sqrt root operation in the submission only for the ease of consistent presentation and theoretical analysis.
>
> As you suggested, in the manuscript, we have moved $\varepsilon$ out of the sqrt root operation to accord with its practical implementation,  and also have revised the proof. This modification does not affect our experimental results since, in our vanilla implementation,  $\varepsilon$ is out of the sqrt root.  As for the theoretical results, the complexity's dependence on $\varepsilon$ changes from $\mathcal{O}(c_{\infty}^{2.5}/(\varepsilon \epsilon^4))$ to $\mathcal{O}(c_{\infty}^{2.5}/(\varepsilon^{2} \epsilon^4))$ for the Lipschitz gradient case. And the complexity for the Lipschitz Hessian case becomes $\mathcal{O}(c_{\infty}^{1.25}/(\varepsilon^{1.25} \epsilon^{3.5}))$ from the original $\mathcal{O}(c_{\infty}^{2.5}/(\varepsilon\epsilon^{3.5}))$. Hence, all the theoretical refinements do not affect our conclusion.

---

> ### Author Response · Authors · 2022-11-16
> **Response to Reviewer 5SGt (Part II)**
>
> **Q2: It seems that the authors are aware of AdaBelief and Adam+ but didn't evaluate them in the experiments. I highly suggest the authors to also evaluate AdamBelief, Adam+ and Aida in their experiments.**
>
> **A2**: Thanks.  In Sec. B   of the vanilla submission, we have already compared  AdaBlief and many other optimizers (e.g., Nadam,   AdaBound,  Radam, Padam, etc.) on ResNet-18 and LSTM, which is mentioned at the beginning of Sec. 5 and at the end of Sec. 5.2.  We do not further compare these optimizers on other networks, e.g. ViTs, since these optimizers only evaluate on ResNet-18 and LSTM and do not report official results for other networks on large-scale datasets, e.g., ImageNet.  Indeed,  due to space limitations, we defer many extra experiments into  Sec. B, including the comparison of convergence curves on ResNet and ViT, the evaluation results on RL tasks,  and the ablation study.  Please see the more detailed comparison in  Sec. B.
>
> Per your suggestion, we have compared with Adai on ResNet-18 and ViT, and respectively reported the results in Tab. 9 and Tab. 10 in Sec. B of the revision.  For Adai, we search its learning rate, weight decay, and $\varepsilon$ to achieve the best performance. Please refer to the following results. Since  Adam$^+$ code is not released,   it is hard to implement and reproduce the results in a few days. So we do not compare with it now but will try our best to implement and reproduce their results.
>
> This table summarizes the results on ViT-Small trained on ImageNet with 100-300 epochs.
>
> |      Epoch      | 100  | 150  | 200  | 300  |
> | :-------------: | :--: | :--: | :--: | :--: |
> | AdamW (default) | 76.1 | 78.9 | 79.2 | 79.9 |
> |      Adam       | 62.0 | 64.0 | 64.5 | 66.7 |
> |    **Adai**     | 66.4 | 72.5 | 75.3 | 77.4 |
> |      SGD-M      | 64.3 | 68.7 | 71.4 | 73.9 |
> |      LAMB       | 69.4 | 73.8 | 75.9 | 77.7 |
> |    **Adan**     | 77.5 | 79.6 | 80.0 | 80.7 |
>
> This table reports the results on ResNet-18 trained on ImageNet with 90 epochs. The results with the notation $*$ are from the AdaBlief paper.
> |   Adan    |    SGD    | Nadam | AdaBound  |   Adam    |   Radam   | Padam | LAMB  |   AdamW   | AdaBlief  | **Adai** |
> | :-------: | :-------: | :---: | :-------: | :-------: | :-------: | ----- | :---: | :-------: | :-------: | :------: |
> | **70.60** | 70.23$^*$ | 68.82 | 68.13$^*$ | 63.79$^*$ | 67.62$^*$ | 70.07 | 68.46 | 67.93$^*$ | 70.08$^*$ |  69.68   |
>
> The results show that Adan consistently outperforms all compared adaptive optimizers. Moreover, we also have discussed  Adai in the related work. Please refer to the revision.

---

> ### Author Response · Authors · 2022-11-16
> **Response to Reviewer 5SGt (Part I)**
>
> Thank you for the insightful review! We hope our response is helpful in addressing your concerns.
>
> **Q1: The authors claim that Adan has lower complexity than a number of existing optimizers including AdaBelief and LAMB without an explanation. Is it because of the separation of the l2 regularizer with the objective, or because of the introduction of $g_k- g_{k-1}$ in Adan, or because of improved mathematical deviation? This is very important. It helps the readers to understand what is the reason for the low complexity of the new method without reading all the proofs. If it is because of the separation of the l2 regularizer with the objective or improved mathematical deviation, it suggests that the convergence of other optimizers can also be improved.**
>
> **A1**: Thanks. The improvement mainly comes from the term $(g_k- g_{k-1})$. This is because 1)  it allows Adan to use a larger learning rate which can be up to the order of $\mathcal{O}(1)$,  while the learning rate for other optimizers, e.g. Adam, AdamW, LAMB, and Adam$^+$,  is at the order of $\mathcal{O}(\operatorname{poly}(\epsilon))$ and is much smaller; 2) large learning rate can accelerate convergence in most cases, and indeed also benefits generalization by helping find the flat minima (see [Ref. 2]). The term $(g_k- g_{k-1})$ allows us to use a larger learning rate,  since, intuitively, it can help estimate more stable and accurate first- and second-order moments. Specifically, $(g_k- g_{k-1})$ contains certain second-order information by approximating it as $(g_k- g_{k-1}) \approx H_k (x_{k} - x_{k-1})$, where $H_k$ denotes a certain Hessian at a point between $x_{k}$ and $x_{k-1}$. In this way, adding the term $(g_k- g_{k-1})$ or its moving average $v_k$  into first- and second-order moments would add the above extra Hessian information and thus helps estimate more accurate and stable first- and second-order moments.
>
> For the mathematical deviation, even sharing a very similar proof roadmap with Adan, the very recent work [Ref. 1]  proves the complexity $\mathcal{O}(c_{\infty}^{2.5}/\varepsilon^{2.5} \epsilon^4)$  of AdamW  which is still inferior to the complexity $\mathcal{O}(c_{\infty}^{2.5}/\varepsilon^{2} \epsilon^4)$ of Adan, where $c_{\infty}$ is the $\ell_\infty$-norm upper bound of the stochastic gradient, and $\varepsilon$ is the small constant in $\eta_k = \eta/(\sqrt{n_k}+\varepsilon)$. Consider $\varepsilon$ is often very small, e.g., $\varepsilon=10^{-8}$ in practice, the improvement $\mathcal{O}(\varepsilon^{-0.5})$ of Adan over AdamW is actually significant, and is not from the mathematical deviation.
>
> Note, we do not borrow the proof of [Ref. 1] since our paper is released earlier.  The shared proof roadmap has two steps. First, both works upper bound the noise variance of the first-order moment. Second, both works establish the relations among the current loss, the previous loss, and the full gradient norm at the current iteration and then accumulate these relations for all iterations to obtain the desired full gradient norm.  But for each step, both works need to elaborately consider and handle each step differently because they analyze different algorithms.
>
> For the separation of the $\ell_2$ regularizer, it indeed does not boost the convergence since 1) it is used in  LAMB, AdamW, and Adan; 2) Adan still enjoys lower complexity than LAMB and AdamW.
>
> 	[Ref. 1] Anonymous. "Towards Understanding Convergence and Generalization of AdamW." ICLR Submission, 2022, https://openreview.net/forum?id=EfTN2tSGlF.
>
> 	[Ref. 2] Cohen, Jeremy, Simran Kaur, Yuanzhi Li, J. Zico Kolter, and Ameet Talwalkar. "Gradient Descent on Neural Networks Typically Occurs at the Edge of Stability." ICLR. 2020.

---

### Official Review · Reviewer_wTd4 · 2022-10-26

**Confidence:** 3
**Correctness:** 2
**Technical Novelty And Significance:** 3
**Empirical Novelty And Significance:** 3
**Recommendation:** 6

**Clarity, Quality, Novelty And Reproducibility:**

Clarity: I think the paper is easy to read. I also suggest to add reference for Table 1, Nesterov acceleration in the last paragraph of page 2 and lower bound $\epsilon^{-4}$ in the first paragraph of page 3.

Novelty: The algorithm seems new to me and the construction is interesting.

**Strength And Weaknesses:**

### Strength
1. The idea is clear and straightforward
2. The experiments are conducted for multiple tasks

### Weaknesses
1. The claim that $O(\epsilon^{-3.5})$ is the lower bound is not correct. In Assumption 1, it assumes the function is smooth with respect to each value of $\zeta$, which will already lead to a lower bound of $O(\epsilon^{-3})$ even without Lipschitz Hessian. I also found that Assumption 2 is stronger than usual, as it assume $\mathbb{E}||\xi|| \leq \sigma$ rather than $\mathbb{E}||\xi||^2 \leq \sigma^2$. So I think lower bound could be much better than $O(\epsilon^{-4})$ and $O(\epsilon^{-3.5})$ for each case.
2. From the change in equation (4), the algorithm introduce a new hyperparameter $\beta_2$. In the new algorithm, there are three momentum hyperparameters to pick: $\beta_1, \beta_2$ and $\beta_3$. When $\beta_2 = 0$, the algorithm basically reduces to the vanilla Adam. In the experiment, it performs much better than Adam, so I am not sure whether it is simply a result of tuning $\beta_2$. However, the ablation study in B.5 indicate that Adan is not sensitive to the value of $\beta_2$, which contradicts that it is much better than Adam.




**Summary Of The Paper:**

The paper introduce a new adaptive method that built upon ADAM and Nesterov momentum. First, it reformulate Nesterov momentum update and combine it with ADAM; Second, it provides $O(\epsilon^{-4})$ complexity for Lipschitz-smooth case; Third, it provides $O(\epsilon^{-3.5})$ complexity for Hessian Lipschitz case.

**Summary Of The Review:**

Overall, the algorithm looks promising, but does not match the lower bounds in theory and I am not fully convinced by the experiments.

---

> ### Author Response · Authors · 2022-11-16
> **Response to Reviewer wTd4 (Part II)**
>
> **Q2: When $\beta_2 = 0$, the algorithm basically reduces to the vanilla Adam. In the experiment, it performs much better than Adam, so I am not sure whether it is simply a result of tuning $\beta_2$. However, the ablation study in B.5 indicates that Adan is not sensitive to the value of $\beta_2$, which contradicts that it is much better than Adam.**
>
> **A2**: There is some misunderstanding, since Adan with $\beta_2$ indeed does not become Adam. Specifically, although   $\beta_2=0$ removes the effects of gradient difference term $(g_k - g_{k-1})$ in the first-order moment,   as shown in the following equation,  the second-order moment  $n_k$ in Adan indeed contains $(g_k - g_{k-1})$ because of  $\beta_3 \neq 0$:
> 	$$
> 	n_k = (1-\beta_3)g_{k-1} + \beta_3[g_k + (1-\beta_2) (g_k - g_{k-1})]^2,$$
> So the second-order moment  $n_k$ still can use the information of $(g_k - g_{k-1})$ and thus still benefits Adan as investigated below.
> 	In contrast, for Adam, its second-order moment is $n_k = (1-\beta_3)g_{k-1} + \beta_3 g_k^2$ which distinguishes it from Adan.
>
> To investigate the effectiveness of gradient difference in the second-moment term $n_k$, we perform an additional ablation study on ViT-Small trained on ImageNet for 150 epochs.
> 	We compare the official optimizer AdamW, Adan without gradient difference term in first-order moment $m_k':=m_k + (1-\beta_2) v_k$ ($\beta_2 = 0$ giving $v_k = 0$), Adan using Adam's $n_k$ (i.e. setting  $n_k = (1-\beta_3)g_{k-1} + \beta_3 g_k^2$ in Adan), and Adan itself. The results are summarized  in the following table:
> |    Optimizer     | AdamW | Adan with Adam's $n_k$ | Adan ($\beta_2 = 0$) | Adan |
> | :--------------: | :---: | :--------------------: | :------------------: | :-----: |
> | Acc. on ImageNet | 78.3  |          79.2          |         79.4         | 79.6 |
>
> One can observe that 1) for Adan itself,  the gradient difference term plays a key role in both first-order moment  $m_k'$ and second-order moment $n_k$, since independently removing gradient difference term in $m_k'$ and  $n_k$ leads to performance degradation compared with vanilla Adan; 2) even without the gradient difference term in  $m_k'$ or in $n_k$, Adan still outperforms the official AdamW optimizer on ViT-small.  Intuitively, the term $(g_k- g_{k-1})$ helps to estimate the more stable and accurate first- and second-order moments. Specifically, $(g_k- g_{k-1})$ contains certain second-order information by approximating it as $(g_k- g_{k-1}) \approx H_k (x_{k} - x_{k-1})$, where $H_k$ denotes a certain Hessian at a point between $x_{k}$ and $x_{k-1}$. In this way, adding the term $(g_k- g_{k-1})$ or its moving average $v_k$ into first- and second-order moments would add the above extra Hessian information and thus helps estimate more accurate and stable first- and second-order moments.
>
>
> **Q3: I also suggest adding a reference for Table 1, Nesterov acceleration in the last paragraph of page 2, and lower bound $\Omega(\epsilon^{-4})$ in the first paragraph of page 3.**
>
> **A3**: Thanks for the good suggestion. We have added the reference and marked the modification with blue color. If we add references into Table 1, it increases the number of pages and violates the page limit. We will refine it in the final version.

---

> > ### Comment · Reviewer_wTd4 · 2022-12-07
> > **update**
> >
> > After reading the response, I increased my score from 5 to 6.

---

> > > ### Author Response · Authors · 2022-12-07
> > > **Thanks to Reviewer wTd4**
> > >
> > > Many thanks. We are glad our response can help you better understand our work, and address your concerns. Thank you for your insightful and positive comments again!!

---

> ### Author Response · Authors · 2022-11-16
> **Response to Reviewer wTd4 (Part I)**
>
> Thank you for the insightful review and positive feedback!  We hope your concerns can be addressed by the following clarification.
>
> **Q1: The claim that $\Omega(\epsilon^{-3.5})$ is the lower bound is not correct. In Assumption 1, it assumes the function is smooth with respect to each value of $\zeta$, which will already lead to a lower bound of $\Omega(\epsilon^{-3.0})$ even without Lipschitz Hessian. I also found that Assumption 2 is stronger than usual.**
>
> **A1**: For the lower bound, as proven in [Ref. 1], on the nonconvex problems with Lipschitz gradient and Hessian,  for stochastic gradient-based methods with 1) unbiased and variance-bounded stochastic gradient and 2) stochastic gradient queried on the same point per iteration, their complexity lower bound is  $\Omega(\epsilon^{-3.5})$ to find an $\epsilon$-accurate first-order stationary point.  For condition 2), it means that per iteration, the algorithm only queries the stochastic gradient at one point (e.g. SGD, Adam, Adan) instead of multiple points (variance-reduced algorithms, e.g. SVRG).  For the nonconvex problems with Lipschitz gradient *but without* Lipschitz Hessian, the complexity lower bound is $\Theta(\epsilon^{-4})$ as shown in  [Ref. 2].
>
> Note, the above  Lipschitz gradient and Hessian assumption are defined on the training loss w.r.t. the variable/parameter instead of w.r.t. each datum/input $\zeta$. **We would like to clarify that our proofs are only based on the above Lipschitz gradient and Hessian assumptions and do not require the Lipschitz gradient and Hessian w.r.t. the datum/input $\zeta$. In the revision, we already refine the Lipschitz gradient and Hessian assumption for more clarity. Please refer to Assumption 1 and Assumption 2 in Sec. 4**.
>
> To obtain $\Omega(\epsilon^{-3})$ complexity, one often needs 1) (quasi) second-order information, e.g.,  stochastic Hessian-vector product or exact Hessian matrix,  or  2) variance reduction technique which at each iteration queries the stochastic gradient at multiple points for the same datum $\zeta$ to reduce the variance (see SVRG algorithm).  However, for practical network training, both second-order methods and variance reduction methods are almost not used because both methods largely increase the training cost or are even computationally prohibitive.
>
> 	[Ref. 1] Arjevani, Yossi, Yair Carmon, John C. Duchi, Dylan J. Foster, Ayush Sekhari, and Karthik Sridharan. "Second-order information in non-convex stochastic optimization: Power and limitations.". COLT, 2020.
>
> 	[Ref. 2] Arjevani, Yossi, Yair Carmon, John C. Duchi, Dylan J. Foster, Nathan Srebro, and Blake Woodworth. "Lower bounds for non-convex stochastic optimization." Mathematical Programming, 2022.

---

### Decision · Program_Chairs · 2023-01-20

**Decision:**

Reject

**Justification For Why Not Higher Score:**

The conceptual and algorithmic contribution of the paper seems to be limited, and the main strength is primarily in the promising experimental results. However, some of the improvement in the experiments may be due to the additional tuning that the algorithm performs compared to prior works.

**Justification For Why Not Lower Score:**

N/A

**Metareview: Summary, Strengths And Weaknesses:**

The paper proposes a variant of Adam that is based on Nesterov's accelerated gradient descent algorithm instead of the heavy-ball momentum. The paper reformulates the Nesterov update scheme so that the extrapolated point is not kept in memory, and thus avoids an unnecessary reload of the parameters during backpropagation. The paper empirically evaluates the resulting algorithm on a wide range of tasks and show that it consistently improves upon state of the art optimizers.

The reviewers agreed that the main conceptual contribution, namely the incorporation of the Nesterov AGD scheme, is a straightforward one and the theoretical contribution is limited. The reformulation is a small contribution. The reviewers appreciated the strength of the experimental results and the consistent improvements over existing algorithms across tasks. However, there were some concerns regarding the algorithm and the experiments:

- Compared to Adam-type algorithms that use only two parameters, the algorithm introduces a third parameter that needs to be tuned. Some of the practical performance could be attributed to the tuning of this additional parameter. However, the authors conducted experimental studies that indicate that the algorithm is not too sensitive to the choice of the parameter. Nevertheless, the parameter tuning overhead introduced by a third parameter is a consideration when comparing the algorithm to existing methods.

- Some existing algorithms such as Adabelief were used as comparisons in certain tasks but not others.

- The paper performed tuning of the epsilon parameter only for the proposed algorithm and not the other methods.

The main strength of the paper is that it consistently outperforms existing methods in all of the experiments, although this improvement does come with additional tuning overhead. The main weaknesses are that the conceptual and algorithmic contributions seem limited, and it is not clear how much of the practical benefit is attributable to the algorithm design versus tuning of additional parameters. Overall, the weaknesses seem to outweigh the strengths.